# Comprehensive molecular comparison of *BRCA1* hypermethylated and *BRCA1* mutated triple negative breast cancers

Dominik Glodzik[1,2,3,12], Ana Bosch[1,4,12], Johan Hartman [5], Mattias Aine [1,6], Johan Vallon-Christersson[1], Christel Reuterswärd[1], Anna Karlsson[1], Shamik Mitra [1], Emma Niméus[1,7], Karolina Holm[1], Jari Häkkinen [1], Cecilia Hegardt[1], Lao H. Saal [1], Christer Larsson[8], Martin Malmberg [4], Lisa Rydén[7], Anna Ehinger [1,9], Niklas Loman[1,4], Anders Kvist [1], Hans Ehrencrona [9,10], Serena Nik-Zainal [11,12], Åke Borg[1,12] & Johan Staaf [1,12✉]

Homologous recombination deficiency (HRD) is a defining characteristic in *BRCA*-deficient breast tumors caused by genetic or epigenetic alterations in key pathway genes. We investigated the frequency of *BRCA1* promoter hypermethylation in 237 triple-negative breast cancers (TNBCs) from a population-based study using reported whole genome and RNA sequencing data, complemented with analyses of genetic, epigenetic, transcriptomic and immune infiltration phenotypes. We demonstrate that *BRCA1* promoter hypermethylation is twice as frequent as *BRCA1* pathogenic variants in early-stage TNBC and that hypermethylated and mutated cases have similarly improved prognosis after adjuvant chemotherapy. *BRCA1* hypermethylation confers an HRD, immune cell type, genome-wide DNA methylation, and transcriptional phenotype similar to TNBC tumors with *BRCA1*-inactivating variants, and it can be observed in matched peripheral blood of patients with tumor hypermethylation. Hypermethylation may be an early event in tumor development that progress along a common pathway with *BRCA1*-mutated disease, representing a promising DNA-based biomarker for early-stage TNBC.

---

[1] Division of Oncology, Department of Clinical Sciences Lund, Lund University, Medicon Village, SE-22381 Lund, Sweden. [2] Wellcome Sanger Institute, Wellcome Genome Campus, CB10 1SA Cambridge, UK. [3] Department of Epidemiology and Biostatistics, Memorial Sloan Kettering Cancer Center, New York, NY, USA. [4] Department of Oncology, Skåne University Hospital, SE-22184 Lund, Sweden. [5] Department of Oncology and Pathology, Karolinska Institute, SE-17177 Stockholm, Sweden. [6] Division of Molecular Hematology, Department of Laboratory Medicine, Lund University, SE-22184 Lund, Sweden. [7] Division of Surgery, Department of Clinical Sciences, Lund University, SE-22184 Lund, Sweden. [8] Division of Translational Cancer Research, Department of Laboratory Medicine, Lund University, Medicon Village, SE-22381 Lund, Sweden. [9] Department of Genetics and Pathology, Laboratory Medicine, Region Skåne, SE-22184 Lund, Sweden. [10] Division of Clinical Genetics, Department of Laboratory Medicine, Lund University, SE-22184 Lund, Sweden. [11] Academic Department of Medical Genetics, The Clinical School University of Cambridge, Cambridge Biomedical Research Campus, CB2 0QQ Cambridge, UK. [12] These authors contributed equally: Dominik Glodzik, Ana Bosch, Serena Nik-Zainal, Åke Borg, Johan Staaf. ✉email: johan.staaf@med.lu.se

Triple-negative breast cancer (TNBC) encompasses a subgroup of tumors defined by absence of estrogen and progesterone receptor expression and lack of amplification of the human epidermal receptor growth factor 2/erythroblastic oncogene B (HER2/ERBB2) gene. TNBCs comprise ~10% of all breast cancers and are clinically aggressive with an often poor prognosis, partly due to the lack of targeted therapeutics. Although being classified as a clinical tumor entity, TNBC tumors display a high degree of molecular heterogeneity. TNBC tumors are associated with pathogenic variants in the BRCA1 breast cancer susceptibility gene, with 7–20% of diagnosed patients harboring pathogenic germline or somatic variants[1–4]. The BRCA1 protein has multiple distinct roles in maintaining genome integrity, particularly, through homologous recombination (HR)-mediated double strand break repair[5]. Tumor cells deficient for BRCA1 (or BRCA2) are considered HR-deficient (HRD) and sensitive to cytotoxic agents and poly (ADP-ribose) polymerase (PARP) inhibitors, which cause DNA damage or increased demand for double strand break repair[6]. The HRD phenotype is utilized in promising clinical studies of germline BRCA1/BRCA2-mutated breast or ovarian cancer[3,7,8]. In addition, BRCA1/BRCA2-mutated breast cancers have been suggested to be more immunogenic than non-HR defective tumors[9–11]. The increase in immunogenicity may be related to better responses to checkpoint blockade response, although this remains to be proven.

Between 40% and 70% of TNBC tumors are reported to have a presumed HRD phenotype, which exceeds the number of cases with germline/somatic BRCA1/BRCA2 inactivating variants[3,12,13]. This suggests that other mechanisms and/or genes may confer a similar phenotype. DNA promoter hypermethylation could be an alternative mechanism of inactivating BRCA1. BRCA1 promoter hypermethylation has been reported in 16–57% of TNBCs across studies[14–20], superseding the frequency of germline BRCA1 alterations, however, with conflicting reports about association with prognosis (e.g. refs. [14–16,18,21]). Currently, we lack a detailed multi-layer comparison of BRCA1 hypermethylated versus BRCA1-mutated early stage TNBCs using current state-of-the-art profiling techniques that thoroughly investigates similarities and differences between the two groups.

In the current study, we pursued the hypothesis that BRCA1 promoter hypermethylation confers an omics phenotype identical to that of BRCA1-mutated TNBCs, and that the two entities have equivalent patient outcomes in response to standard of care chemotherapy. To this end, we analyzed a recently reported unselected population-based cohort of 237 early stage TNBC tumors profiled by comprehensive whole-genome sequencing (WGS), RNA sequencing, global DNA methylation analysis[22], further complemented with 54 additional BRCA1-mutated tumors from previous studies[23,24]. Herein, we sought the frequency of BRCA1 promoter hypermethylation, its tumor phenotype compared to tumors with BRCA1 inactivating genetic variants (somatic or germline, BRCA1-null), and its association with clinicopathological variables, molecular subtypes, and patient outcomes in early-stage TNBC. We demonstrate that BRCA1 hypermethylation is twice as frequent as BRCA1-null tumors in early-stage unselected TNBC and that elevated BRCA1 promoter methylation is detectable in peripheral blood DNA of patients with hypermethylation in the tumor. Moreover, we show that in terms of mutational, epigenetic, transcriptional, and immune infiltration profiles, BRCA1 hypermethylation confers a tumor phenotype practically identical to that of BRCA1-null cases. BRCA1 hypermethylation and BRCA1 mutation are equally associated with better outcome after adjuvant chemotherapy when compared to TNBC patients without BRCA1 inactivation, thus BRCA1 hypermethylation represents a promising DNA-based prognostic marker.

## Results

### BRCA1 mutations and promoter hypermethylation in TNBC.
Table 1, Fig. 1, and Supplementary Fig. 1 outlines patient demographics, selection, and study layout. The original 237 TNBC patients (hereinafter referred to as the SCAN-B cohort) reported by Staaf et al.[22] represent 58% of the total number of diagnosed TNBC cases in the studied healthcare region during the inclusion period (September 1 2010 to March 31, 2015), and has been shown to be representative of the total regional population with respect to key clinicopathological variables. Of these patients, 24.1% (57/237) were classified as BRCA1 promoter hypermethylated based on pyrosequencing, while 25 were BRCA1-null cases of which 19 carried germline variants and six somatic variants (Supplementary Data 1). Similar hypermethylation rates were observed across different years of diagnosis in the SCAN-B cohort: 21.0% hypermethylated cases diagnosed 2011, 22.2% in 2012, 26.4% in 2013, and 21.7% in 2014. Pyrosequencing classifications were corroborated by Illumina DNA methylation profiling data for BRCA1 gene associated CpGs (Fig. 2a), and markedly reduced BRCA1 mRNA expression from RNA sequencing for hypermethylated cases (Fig. 2b), similar to previous reports[25], that were in line with expression levels for cases with BRCA1 frame shift, nonsense and indel variants (Supplementary Fig. 2). Of the 57 hypermethylated cases, 51 (89.5%) showed concurrent LOH of BRCA1 (tumor cell content by WGS range 23–82%) with the six remaining cases having low estimated tumor cell content (between 11% and 23% by WGS),

## Table 1 Clinicopathological characteristics of SCAN-B TNBC patients.

| | BRCA1 hypermethylated | BRCA1-null[a] | non-BRCA1 |
|---|---|---|---|
| N | 57 | 25 | 155 |
| Age | | | |
| <35 years | 14.0% | 28.0% | 0.6% |
| 35–50 years | 28.1% | 12.0% | 11.0% |
| 50–70 years | 47.4% | 52.0% | 43.9% |
| ≥70 years | 10.5% | 8.0% | 44.5% |
| Germline BRCA1 variant[b] | 0% | 76% | 0% |
| Tumor size | | | |
| ≤20 mm | 57.9% | 52.0% | 47.7% |
| >20 mm | 42.1% | 48.0% | 52.3% |
| Nodal status | | | |
| Node negative | 70.2% | 48.0% | 65.2% |
| Node positive | 28.1% | 48.0% | 34.2% |
| Missing data | 1.8% | 4.0% | 0.6% |
| Tumor grade | | | |
| Grade 2 | 0% | 0% | 18.1% |
| Grade 3 | 98.2% | 96.0% | 80.0% |
| Missing data | 1.8% | 4.0% | 1.9% |
| ER-staining positivity[c] | | | |
| <1% | 89.5% | 84.0% | 87.6% |
| 1–10% | 10.5% | 16.0% | 12.4% |
| Therapy[d] | | | |
| Chemotherapy | 87.5% | 91.7% | 66.9% |
| Untreated | 12.5% | 8.3% | 33.1% |
| IDFS event[e] | 17.5% | 28.0% | 38.7% |
| Median IDFS for patients (years)[f] | 5.0 (0.1–7.1) | 4.8 (0.2–6.7) | 4.6 (0.6–7.2) |
| DRFI event[e] | 10.5% | 20.0% | 24.5% |
| Median DRFI for patients (years)[f] | 4.6 (0.1–7) | 4.1 (0.05–6.6) | 4.3 (0.4–7.2) |
| Death event[E] | 14.0% | 24.0% | 31.6% |
| Median OS for patients (years)[f] | 4.7 (0.2–7.1) | 4.1 (2.9–6.8) | 4.6 (2.7–7.1) |

Data obtained from the Swedish national breast cancer quality registry. Cases with missing data omitted from calculations if not shown as separate variable.
[a]BRCA1-null includes cases with both germline and somatic BRCA1 inactivating genetic variants.
[b]Based on whole genome sequencing data.
[c]In Sweden, ER-negativity is defined as ≤10% of cells with IHC-staining for ER.
[d]Includes all cases irrespective if eligible for outcome analysis, but excluding cases with palliative treatment.
[e]Includes all events, irrespective of eligibility for outcome analysis.
[f]Time and range for patients without an event, irrespective of eligibility for outcome analysis.

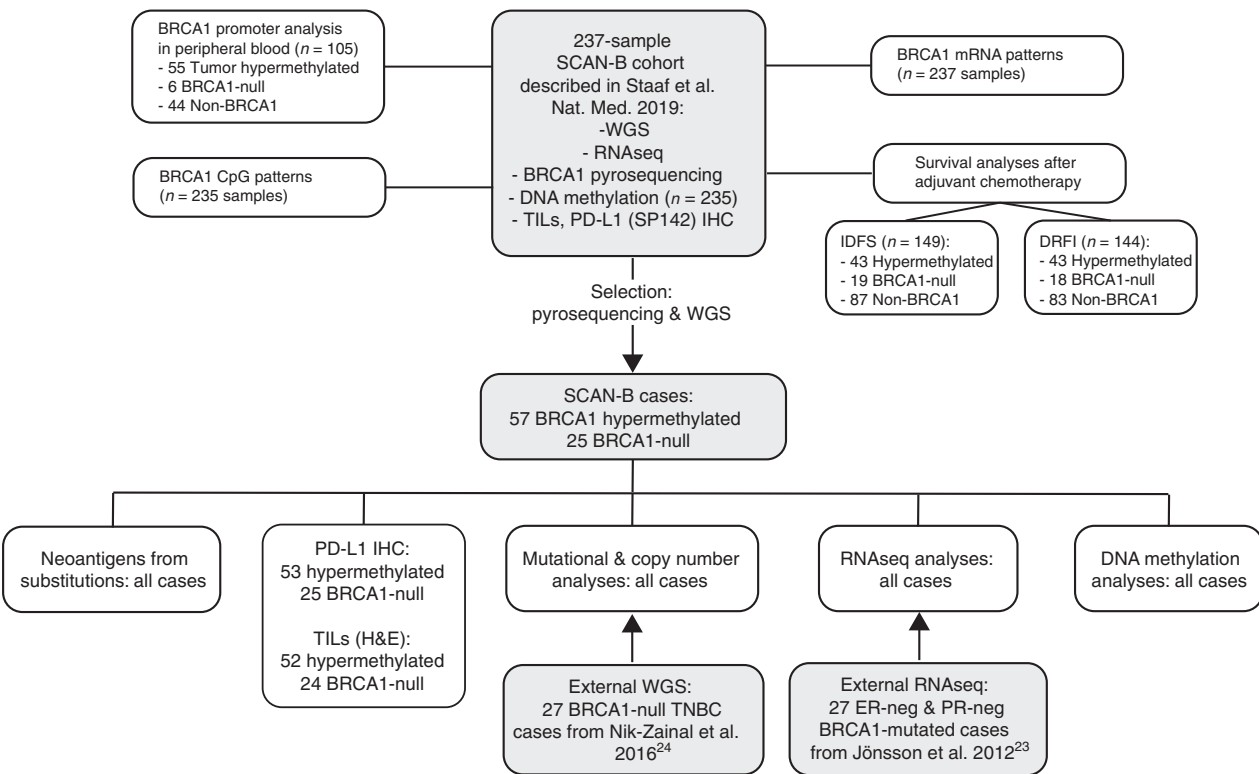

**Fig. 1 Study scheme, performed analyses, and cohorts used.** Gray boxes indicate a cohort of samples.

which possibly interfered with the copy number analysis. In comparison, 84% of *BRCA1* germline cases showed LOH of the wild-type allele, again with low associated tumor cell content for cases without LOH ($n = 3$, 11%, 18%, and 27% by WGS). Furthermore, *BRCA1* hypermethylated and *BRCA1* germline mutated cases without *BRCA1* LOH had similar proportions of rearrangement signature 3 and deletions with microhomology, representing prototypical signatures of *BRCA1*-deficient cancer[24] (Fig. 2c). This suggests that *BRCA1* hypermethylation or *BRCA1* germline alteration without concurrent LOH is rare in early-stage TNBC specimens and when observed may be due to tissue sampling limitations. Consistent with Fig. 2c, HRD classification from the previous report[22] revealed that 98.2% (56/57) of hypermethylated cases were called as HR deficient by two different HRD algorithms[13,26]. Presence of nonmalignant cells in tumors may skew analyses of genomic data obtained from bulk tumor analyses (such as RNA sequencing and DNA methylation). In the SCAN-B cohort, there was no statistical difference in tumor cell content estimated by WGS between hypermethylated and *BRCA1*-null cases (*t*-test, $p = 0.47$), suggesting that non-malignant infiltration should not affect group-level conclusions. Tumor cell content showed a strong linear correlation with *BRCA1* CpG allele methylation rate for hypermethylated cases when using WGS specific estimates also accounting for possible subclonality ($r^2 = 0.84$, slope $= 0.90$, Fig. 2d).

Overall, *BRCA1* hypermethylation was 2.3 times more frequent than *BRCA1*-null cases, and three times more frequent than *BRCA1* germline alterations. In the SCAN-B cohort, *BRCA1* hypermethylation was mutually exclusive with *BRCA1*-null cases and *BRCA2* tumors with only one exception. One of the ten *BRCA2*-null SCAN-B cases displayed *BRCA1* hypermethylation. This case, PD35990a, showed a pathogenic biallelic *BRCA2* variant (p.Pro3194Asnfs*2 germline mutation and *BRCA2* LOH) and distinct *BRCA1* hypermethylation (61% CpG allele methylation) together with *BRCA1* LOH. The tumor had genomic

patterns of Substitution Signature 3, Rearrangement Signatures 2 and 3 and loss of *BRCA1* gene expression, all characteristic of *BRCA1*- but not *BRCA2*-deficient tumors[24,27] (Fig. 2e).

**Elevated *BRCA1* promoter hypermethylation in peripheral blood.** Forty-six of the 237 SCAN-B patients underwent clinical *BRCA1/2* germline screening due to family history and/or young age at diagnosis according to Swedish guidelines. Nine cases showed germline inactivating *BRCA1* variants and three had germline inactivating *BRCA2* variants. Among the clinically screened patients without a pathogenic germline *BRCA1/BRCA2* variant, 16 out of 34 (47.1%) displayed tumor *BRCA1* hypermethylation while 18 did not. Patients with *BRCA1* hypermethylated tumors were significantly younger at diagnosis compared to the patients negative for both tumor methylation and *BRCA1/2* loss of function variants (median age 36 versus 50 years, *t*-test, $p = 8e-5$). In fact, age at diagnosis for hypermethylated patients was similar to patients with pathogenic germline *BRCA1* variants (median age 36 versus 32 years, *t*-test, $p = 0.82$) (Fig. 3a). Based on oncogenetic counseling data, only one of the 16 *BRCA1* hypermethylated patients had a first-degree relative with breast/ovarian cancer, indicating that these TNBC patients had likely been referred for screening based on young age rather than family history. SCAN-B patients with *BRCA1* hypermethylated tumors ($n = 16$) also showed higher levels of *BRCA1* promoter methylation in matched peripheral blood DNA than the 18 patients with unmethylated tumors and no *BRCA1/2* variants (hypermethylated: $n = 16$, mean CpG allele frequency $= 4.03$, median $= 3.5$, sd $= 1.53$; unmethylated: $n = 18$, mean $= 3.03$, median $= 3.0$, sd $= 0.61$; *t*-test $p = 0.024$, Wilcoxon's test, $p = 0.015$, Fig. 3b). Trends also remained significant after transformation of pyrosequencing methylation rates to *M*-values (*t*-test, $p = 0.02$).

To confirm these findings, we analyzed peripheral blood DNA from 71 additional SCAN-B cases from our cohort not subjected

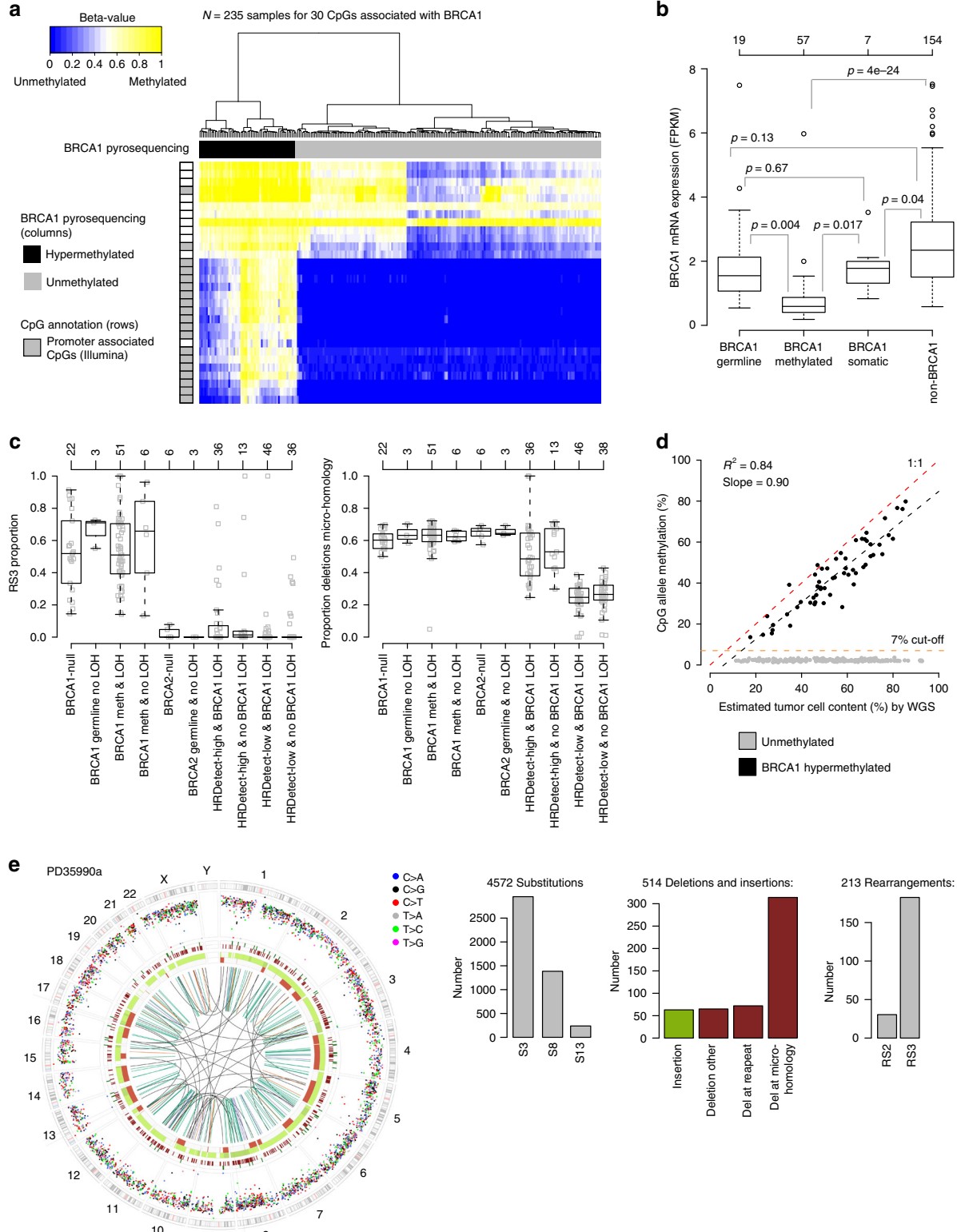

to prior clinical screening, including 39 cases with tumor *BRCA1* hypermethylation and 32 without. Again, higher *BRCA1* promoter methylation levels were found in blood DNA from patients with *BRCA1* tumor hypermethylation (*t*-test, *p* = 0.005, Wilcoxon's test, *p* = 0.01). The finding remained significant after transformation of methylation rates to *M*-values (*t*-test, *p* = 0.006, Wilcoxon's test, *p* = 0.01). When we combined both clinically screened and unscreened SCAN-B patients (*n* = 105 in

total, reanalyzed using the same instrument settings and reagent lots), we observed that the higher *BRCA1* blood DNA methylation levels appeared independent of age (two-way ANOVA interaction model with tumor hypermethylation status and age groups, *p* = 0.002, Fig. 3c). This finding remained significant after transformation of methylation rates to *M*-values (two-way ANOVA interaction model with tumor hypermethylation status and age groups, *p* = 0.002).

**Fig. 2 BRCA1 hypermethylation, gene expression, and HRD association. a** Hierarchical clustering (ward.D2 linkage, Euclidean distance) of DNA methylation data (beta-values shown as a heatmap) for 30 CpGs associated with the *BRCA1* gene (transcription start site (TSS): −1500b to +500 bp) in Illumina MethylationEPIC data for 235 SCAN-B TNBCs, including 57 tumors classified as hypermethylated by pyrosequencing (black column sample annotation bars). Gray CpG annotation bars (rows) indicate a promoter associated CpGs according to Illumina EPIC annotations. **b** *BRCA1* mRNA expression (FPKM) across the 237 SCAN-B cases stratified by gene abrogation status. *p* Values calculated using *t*-test. Top axis shows number of cases per group. **c** Left: proportions of rearrangement signature 3 (RS3)[24] versus patient stratifications based on *BRCA1/2* mutation status, *BRCA1* methylation status, and *BRCA1* LOH in the total SCAN-B cohort, excluding the small HRDetect-intermediate group. For non-BRCA1/2 cases, tumors are stratified by HRDetect-high or low classification[26]. Right: proportion of deletions with microhomology across the same patient subgroups. Top axes show number of cases per group. All cases do not have assigned RS3 rearrangements. **d** *BRCA1* CpG allele methylation versus estimated tumor % by the WGS specific Battenberg algorithm (https://github.com/cancerit/cgpBattenberg) for all 237 SCAN-B cases. Black dotted line corresponds to a linear regression fit for the 57 hypermethylated cases specifically. **e** Circos plot and depiction of mutational substitution (S3, S8, and S13) and rearrangement signatures (RS2 and RS3) as defined in ref. [24] of PD35990a. This case harbors both a *BRCA2* variant and *BRCA1* hypermethylation but has a genetic phenotype of *BRCA1*-deficient cancer. Circos plot depicting from outermost rings heading inwards: karyotypic ideogram outermost. Base substitutions next, plotted as rainfall plots (log10 intermutation distance on radial axis, Ring with short green lines, insertions; ring with short red lines, deletions. Major copy number allele ring (green, gain), minor copy number allele ring (red, loss), Central lines represent rearrangements (green, tandem duplications; red, deletions; blue, inversions; gray, interchromosomal events). FPKM fragments per kilobase of transcript per Million mapped reads. All *p* values reported from statistical tests are two-sided. Source data are provided as a Source Data file.

## Clinical and genomic features of BRCA1 methylated TNBC.

*BRCA1* promoter hypermethylation and *BRCA1*-null frequencies in subgroups of SCAN-B TNBC patients defined by clinicopathological variables and molecular subtypes are shown in Table 2. Hypermethylation frequency was especially high in patients under 50 years (46.2% frequency, Fisher's exact test, *p* = 8e−5). A trend of lower age at diagnosis for hereditary *BRCA1* cases, followed by *BRCA1* hypermethylated cases, *BRCA1/BRCA2* somatic cases, and non-altered cases was observed in the total SCAN-B cohort (Fig. 3d).

Several gene expression based subtyping schemes have been proposed in breast cancer (e.g., the general PAM50[28], CIT[29], and IntClust 10[30] as well as TNBC specific subtypes[31]). In all instances, *BRCA1* hypermethylation was strongly associated with the proposed basal-like phenotype (PAM50 basal-like *p* = 2e−5, CIT basal-like *p* = 1e−6, TNBCtype BL1 *p* = 0.02, and IntClust 10 cluster 10 *p* = 4e−6, Fisher's exact test performed in a 2 × 2 basal-like vs. non-basal context) (Table 2 and Fig. 3e). The subtype proportions for *BRCA1*-null and non-BRCA1 altered cases are shown in Supplementary Fig. 3, demonstrating similarity between hypermethylated and *BRCA1*-null cases.

*BRCA1* hypermethylation was also observed in clinicopathological and molecular subgroups not commonly associated with BRCA1-deficiency, including old patients, tumors with some ER staining (1–10% by immunohistochemistry), and in non-basal like gene expression subtypes, although at low relative frequencies similar to those observed in *BRCA1*-null cases (Table 2).

## BRCA1 hypermethylation and patient outcome in TNBC.

Association of *BRCA1* promoter hypermethylation status with outcome after adjuvant chemotherapy (mainly FEC ± taxane therapy) was investigated using invasive disease-free survival (IDFS) as the primary clinical endpoint in 149 eligible SCAN-B patients (Fig. 1). In both univariable Cox regression and Kaplan–Meier analyses, *BRCA1* hypermethylation alone or combined with *BRCA1*-null cases was associated with significantly longer IDFS than non-altered TNBC cases (Fig. 3f, which also shows univariate results for standard prognostic variables in breast cancer for reference and Fig. 3g). The 5-year IDFS was 88% for *BRCA1* hypermethylated patients versus 71% for non-methylated TNBC patients, while the 5-year distant relapse-free interval (DRFI) was 92% versus 80%, respectively. No difference in IDFS after adjuvant chemotherapy was observed between *BRCA1* hypermethylated and *BRCA1*-null cases (Fig. 3g, log-rank test, *p* = 0.29).

To assess the independent prognostic value after adjuvant chemotherapy of *BRCA1* hypermethylation alone or in combination with *BRCA1*-null status, we performed multivariable Cox regression analysis adjusting for tumor size (≤20 mm, >20 mm), patient age (<40, 40–59, and ≥60 years), tumor grade (2, 3), and lymph node status (N0, N+) using IDFS as the clinical endpoint. *BRCA1* hypermethylation was significantly associated with improved IDFS alone (HR = 0.33, 95% confidence interval (CI): 0.12–0.88) and when combined with *BRCA1*-null cases (HR = 0.35, 95% CI: 0.14–0.84), although the formal proportional hazard assumption was not fulfilled in these analyses (proportional hazard test, *p* < 0.05). When patient age was used as continuous variable in the IDFS analyses the results were borderline nonsignificant (Cox regression *p* = 0.07 for *BRCA1* hypermethylation). Borderline nonsignificant results for *BRCA1* hypermethylation versus non-methylated cases in multivariable Cox regression using the above models with stratified age bins (HR = 0.28, 95% CI: 0.08–1.02, *p* = 0.053) or continuous age (Cox regression *p* = 0.09) were also observed for DRFI. A significant association was seen for DRFI for the combined hypermethylation/*BRCA1*-null group, irrespective of whether binned age (HR = 0.23, 95% CI: 0.07–0.72) or continuous age (HR = 0.32, 95% CI: 0.11–0.94) was used. For untreated or neoadjuvantly treated SCAN-B patients the number of *BRCA1* hypermethylated cases was too low to allow for robust outcome analysis.

## Genetic phenotypes of BRCA1 methylated and BRCA1-null TNBC.

To test the hypothesis that *BRCA1* hypermethylation confers a similar genetic and genomic tumor phenotype as *BRCA1*-null tumors we compared an array of readouts from WGS between the two groups using the combined SCAN-B and Nik-Zainal et al.[24] cohorts (*n* = 57 hypermethylated and *n* = 52 *BRCA1*-null cases in total, Fig. 1).

The genome-wide landscape of copy number gain and loss appeared highly similar between the two groups (Fig. 4a). A full copy number analysis is available in Supplementary Fig. 4, demonstrating no statistical differences between the groups (FDR adjusted Fisher's exact test, *p* > 0.05). Concerning specific breast cancer driver genes/alterations reported by Nik-Zainal et al.[24] we observed some small frequency differences (typically < 5–10%, when excluding *MYC* amplifications and *TP53* mutations) between the two groups regarding copy number amplifications (Fig. 4b) and mutations (Fig. 4c). *RB1* mutation status appeared to show the largest difference between hypermethylated (*n* = 1/57, 1.8%) and *BRCA1*-null (*n* = 4/52, 7.8%) cases for the

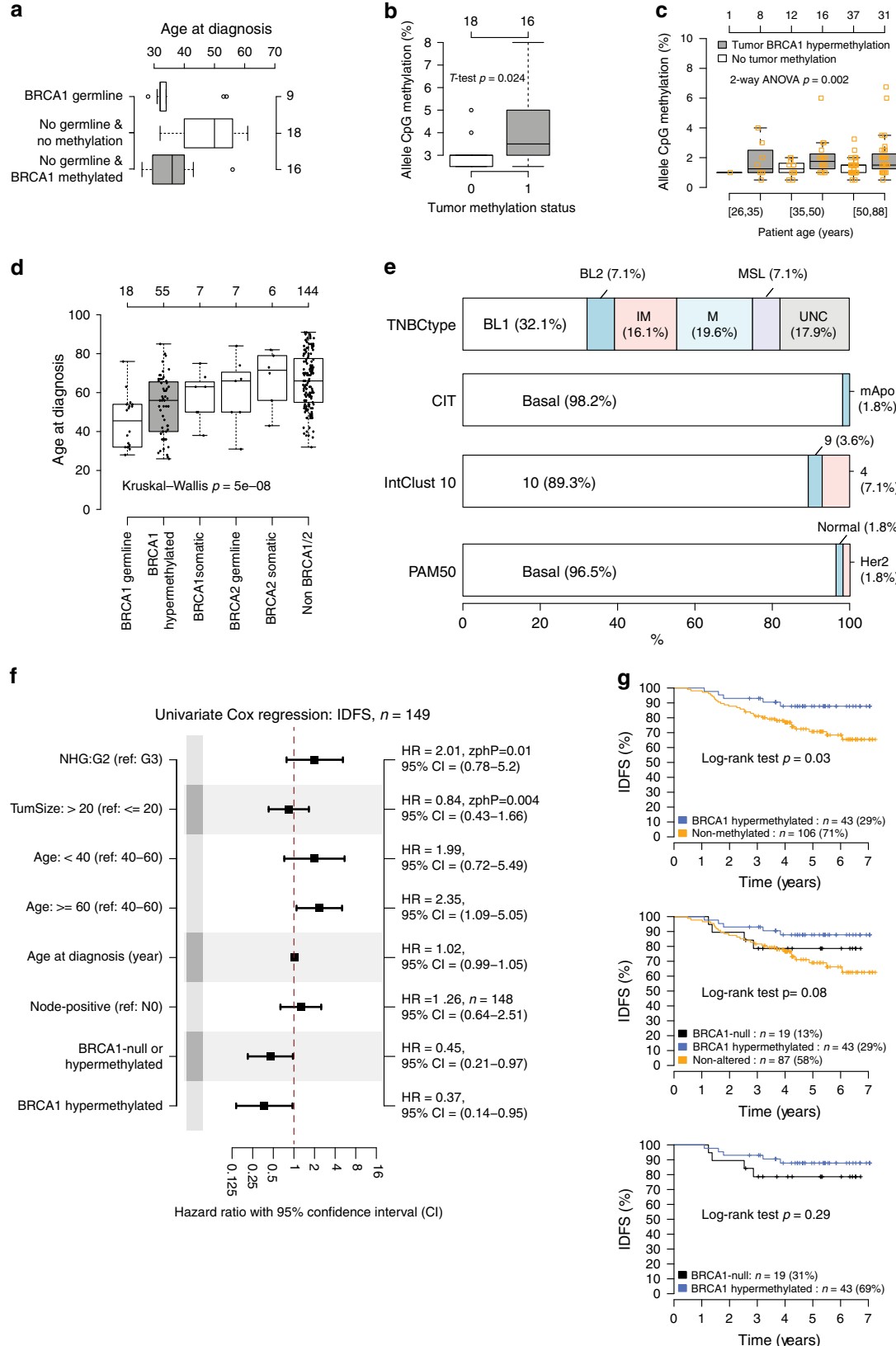

investigated specific driver alterations, albeit not significant possibly due to low numbers (Fisher's exact test, $p = 0.19$). A high frequency of *RB1* alterations, particularly intra-gene rearrangements, has previously been linked with *BRCA1*-null disease[23]. However, when considering all detected *RB1* mutations

(i.e., not filtered for specific drivers), proportions were similar between hypermethylated and *BRCA1*-null cases in the SCAN-B cohort specifically (35 and 32%, respectively). Only a small statistical difference in the number of indels (*BRCA1*-null: median = 410, hypermethylated median = 633) was observed

**Fig. 3 BRCA1 methylation in peripheral blood, gene expression subtypes, and prognosis after therapy. a** Distribution of age at diagnosis for 43 germline screened SCAN-B patients harboring germline *BRCA1* loss of function variants ($n = 9$) or no germline variants ($n = 34$) further stratified by tumor *BRCA1* hypermethylation status. **b** *BRCA1* CpG allele methylation frequency in matched peripheral blood DNA from non-germline SCAN-B patients stratified by their tumor methylation status (methylated = 1) from panel (**a**). **c** *BRCA1* CpG allele methylation frequency in peripheral blood DNA from a combined analysis of 105 SCAN-B cases analyzed using the same instrument settings and reagent lots, including 55 of 57 hypermethylated cases, and stratified by patient age and tumor methylation status. Hard brackets ([]) imply ≥ or ≤, respectively. **d** Age at diagnosis for 237 SCAN-B patients stratified by *BRCA1* and *BRCA2* status. The number of methylated cases is less than 57 as two cases have concurrent *BRCA2* mutations (one germline, one somatic nonpathogenic). **e** Molecular subtype proportions in *BRCA1* hypermethylated SCAN-B cases for PAM50, CIT, IntClust 10, and TNBCtype. CIT subtypes[29]; mApo, molecular apocrine. IntClust 10 subtypes[62]; cluster 10 corresponding to basal-like tumors by other subtyping schemes. TNBCtype subtypes[31]; BL1, basal-like 1: BL 2, basal-like 2: IM, immunomodulatory: M, mesenchymal: MSL, mesenchymal stem-like: LAR, luminal androgen receptor: UNC, uncertain. **f** Univariate Cox regression using invasive disease-free survival (IDFS) as clinical endpoint for different variables in 149 SCAN-B patients eligible for outcome analysis after standard of care adjuvant chemotherapy. HR: hazard ratio. NHG: grade, G2 equals grade 2, G3 equals grade 3. N0: node-negative. A Zph *p* value < 0.05 corresponds to that the proportional hazard assumption is not fulfilled. An ($n =$) indication in the right axis indicates that not all 149 cases were used due to missing values. **g** Kaplan–Meier analysis using IDFS as clinical endpoint for SCAN-B patients eligible for outcome analysis after adjuvant chemotherapy. Top panel shows the 149 patients stratified by *BRCA1* hypermethylation status alone, center panel shows stratification including also *BRCA1*-null cases, and bottom panel a comparison between only hypermethylated and *BRCA1*-null patients. All *p* values reported from statistical tests are two-sided. Source data are provided as a Source Data file.

between the two groups for cases sequenced with at least 30-fold sequence depth when comparing absolute numbers of substitutions, indels and rearrangements between the two groups (Fig. 4d). This absolute difference in numbers did, however, not correspond to a difference in proportions of indels at repeats or with microhomology, as previously shown (Fig. 2c). Moreover, these results imply a similar tumor mutational burden for hypermethylated and *BRCA1*-null cases.

For mutational signatures, *BRCA1*-null cases had higher proportions of substitution signatures suggested to be associated with age at diagnosis (Signatures 1 and 5), an APOBEC signature (Signature 2), and, to some extent, a substitution signature (Signature 8) reported to be elevated in cases with HRD[27] (Fig. 4e). When tested for differential proportions in the SCAN-B cases specifically (25 *BRCA1*-null versus 57 hypermethylated), only the specific APOBEC signature remained elevated in *BRCA1*-null cases (Wilcoxon's test, $p = 1e-05$), driven by a subset (8/25 cases) of *BRCA1*-null cases with elevated signature proportions. Strikingly, in hypermethylated cases no substitutions were assigned to the particular APOBEC signature (Signature 2), while many cases showed substitutions assigned to another reported APOBEC signature (Signature 13). For comparison, in non-basal tumors from Nik-Zainal et al.[24] the median APOBEC signature (Signature 2) proportions were 0.05–0.08 across PAM50 subtypes, with 7% of cases having no signature exposure (Fig. 4e).

No distinct differences were seen for the six rearrangement signatures reported by Nik-Zainal et al.[24], including the two BRCA1/BRCA2 associated signatures (RS3 and RS5) (Fig. 4f). Hierarchical clustering and principal component analyses of substitution and rearrangement signatures, or HRDetect[26] components, did not separate hypermethylated from *BRCA1*-null cases (Fig. 4g–i), supporting similar effects on DNA-repair.

**DNA methylation in BRCA1 methylated and BRCA1-null TNBC.** To investigate differences in global DNA methylation patterns between *BRCA1*-null and hypermethylated cases, Illumina MethylationEPIC profiles from 25 *BRCA1*-null and 57 hypermethylated SCAN-B cases were compared. Preprocessing and filtering left 614,977 informative CpGs sites. Supervised differential methylation analysis between the groups identified 32 significant CpGs (false-discovery rate (FDR) adjusted Wilcoxon's test, $p < 0.05$). Strikingly, all 32 CpGs were associated with *BRCA1*, with 28 CpGs within the canonical promoter region (+500 to −1500 base pairs, chr17:43124984-43126983) and four CpGs further upstream *BRCA1* (chr17:43169746–43171745).

Cluster analysis based on the 32 CpGs recreated, as expected, the division of *BRCA1* hypermethylated versus non-methylated tumors perfectly in the entire cohort of 235 DNA methylation profiled cases (Fig. 5a).

**Gene expression in BRCA1 methylated and BRCA1-null TNBC.** Transcriptomic differences between *BRCA1*-null and *BRCA1* hypermethylated cases were investigated in 52 *BRCA1*-null cases versus 57 hypermethylated cases by combining SCAN-B cases with 27 RNAseq analyzed *BRCA1*-null cases from Jönsson et al.[23] (Fig. 1). Unsupervised clustering using 7224 highly varying genes did not reveal any apparent subclusters specific for the two sample groups (Fig. 5b). This finding was further substantiated by both: (i) unsupervised consensus clustering using the same gene set that showed similar proportions of *BRCA1* hypermethylated and *BRCA1*-null cases across different cluster solutions (Fig. 5c), and (ii) principal component analysis showing that *BRCA1* hypermethylation/*BRCA1*-null status, cohort (SCAN-B or non-SCAN-B) or PAM50 subtypes did not contribute significantly to explaining the variation in gene expression among these tumors (Fig. 5d).

Differential gene expression analysis using significance analysis of microarrays (SAM)[32] between the groups was performed to identify differentially expressed genes using the same 7224 genes as for the cluster analyses. Merely eight genes, *BRCA1, UQCRHL, SOX6, HAPLN1, POLR2J3, H2AC20, MUCL1,* and *HYI*, were differentially expressed at a FDR of 1% (Supplementary Data 2). Of these, *BRCA1* (downregulated in hypermethylated cases, see Fig. 2b), *SOX6* (upregulated in hypermethylated cases), and *MUCL1* (downregulated in hypermethylated cases) showed differential expression also in SCAN-B cases (Wilcoxon's test, $p < 0.05$), while *HAPLN1* was borderline nonsignificant (Wilcoxon's test, $p = 0.051$) when analyzed separately (not accounting for multiple testing). Taken together, the low number of differentially expressed genes identified and the unsupervised analyses suggest that there is no strong transcriptional difference between the hypermethylated and *BRCA1*-null patient subgroups.

**Immune infiltration in BRCA1 methylated and BRCA1-null TNBC.** In silico estimated immune cell composition in *BRCA1* hypermethylated versus *BRCA1*-null cases was compared using three different bulk tissue de-convolution methods based on either gene expression (CIBERSORTx; 6 cell types[33], xCell; 64 cell types[34]) or DNA methylation (EpiDish; 9 cell types[35]) for the 82 hypermethylated or *BRCA1*-null SCAN-B cases. For neither of

**Table 2 BRCA1 promoter hypermethylation and BRCA1-null frequency in TNBC.**

| | BRCA1 hypermethylated (n = 57) | BRCA1-null (n = 25)[a] | BRCA1 non-altered (n = 155) |
|---|---|---|---|
| Total SCAN-B cohort (n = 237)[b] | 24.1% | 10.5% | 65.4% |
| Primary disease only (n = 231) | 24.2% | 10.4% | 65.4% |
| *Age* | | | |
| <50 years (n = 52) | 46.2% | 19.2% | 34.6% |
| 50–70 years (n = 108) | 25.0% | 12.0% | 63.0% |
| ≥70 years (n = 77) | 7.8% | 2.6% | 89.6% |
| *Tumor size* | | | |
| ≤20 mm (n = 120) | 27.5% | 10.8% | 61.7% |
| >20 mm (n = 117) | 20.5% | 10.3% | 69.2% |
| *Nodal status* | | | |
| Node negative (n = 153) | 26.1% | 7.8% | 66.0% |
| Node positive (n = 81) | 19.8% | 14.8% | 65.4% |
| *Tumor grade* | | | |
| Grade 2 (n = 28) | 0% | 0% | 100% |
| Grade 3 (n = 204) | 27.5% | 11.8% | 60.8% |
| *ER-staining positivity* | | | |
| <1% (n = 206) | 24.8% | 10.2% | 65.0% |
| 1–10% (n = 29) | 20.7% | 13.8% | 65.5% |
| *Adjuvant therapy[c]* | | | |
| Chemotherapy (n = 149, IDFS) | 28.9% | 12.8% | 58.4% |
| Untreated (n = 50) | 12.0% | 2.0% | 86.0% |
| *PAM50 subtypes[61]* | | | |
| Basal-like (n = 183) | 30.1% | 13.1% | 56.8% |
| HER2-enriched (n = 31) | 3.2% | 3.2% | 93.5% |
| Luminal A (n = 0) | 0% | 0% | 0% |
| Luminal B (n = 1) | 0% | 0% | 100% |
| Normal-like (n = 22) | 4.5% | 0% | 95.5% |
| *TNBC molecular subtypes[63]* | | | |
| Basal-like 1 (BL1, n = 46) | 39.1% | 13.0% | 47.8% |
| Basal-like 2 (BL2, n = 23) | 17.4% | 8.7% | 73.9% |
| Immunomodulatory (IM, n = 46) | 19.6% | 13.0% | 67.4% |
| Luminal androgen receptor (LAR, n = 30) | 0% | 0% | 100% |
| Mesenchymal (M, n = 41) | 26.8% | 12.2% | 61.0% |
| Mesenchymal stem-like (MSL, n = 14) | 28.6% | 7.1% | 64.3% |
| *IntClust 10 molecular subgroups[62]* | | | |
| 10 (n = 148) | 33.8% | 12.2% | 54.1% |
| 9 (n = 13) | 15.4% | 15.4% | 69.2% |
| 4 (n = 57) | 7.0% | 5.3% | 87.7% |
| 1 (n = 2) | 0% | 0% | 100% |
| 3 (n = 5) | 0% | 0% | 100% |
| 5 (n = 2) | 0% | 0% | 100% |
| 8 (n = 1) | 0% | 0% | 100% |
| *CIT molecular subtypes[29]* | | | |
| Basal-like (basL, n = 175) | 31.4% | 12.0% | 56.6% |
| Molecular apocrine (mApo, n = 46) | 2.2% | 4.3% | 93.5% |
| Luminal B (n = 1) | 0% | 0% | 100% |
| Luminal C (n = 2) | 0% | 0% | 100% |
| Normal-like (n = 4) | 0% | 0% | 100% |

Proportions calculated excluding missing data. Clinical data obtained from the Swedish national breast cancer quality registry.
[a]BRCA1-null includes both germline and somatic cases.
[b]Numbers for each reported group are provided for reference.
[c]Only includes cases eligible for outcome analysis by invasive disease-free survival (IDFS).

and 25 BRCA1-null SCAN-B patients, finding no significant difference in PD-L1 classification (≥1% staining in immune cells), but a borderline nonsignificant trend of higher PD-L1 scores in BRCA1-null cases (Wilcoxon's test, $p = 0.051$, Fig. 6a). Presence of TILs were evaluated on available whole slide H&E sections for 52 hypermethylated and 24 BRCA1-null SCAN-B cases, revealing no significant difference between the groups (Wilcoxon's test, $p = 0.70$) (Fig. 6b).

To investigate whether PD-L1 differences between BRCA1-null and hypermethylated cases were related to neoantigen burden, we calculated number of expressed neoantigens from somatic substitutions by integrating WGS-based HLA-typing, neoantigen prediction, and mRNA expression for 232 of 237 SCAN-B cases[36,37]. The Pearson correlation between the total substitution burden and predicted number of expressed neoantigens from substitutions was 0.91. PD-L1 positive cases (n = 120) had a higher number of expressed neoantigens than PD-L1 negative cases (n = 93) (Wilcoxon's test, $p = 0.02$). In BRCA1-null and hypermethylated cases there was however no statistical difference (Wilcoxon's test, $p > 0.05$) between the groups in total, or when stratified by PD-L1 status, irrespective of whether all predicted neoantigens were assessed (Fig. 6c, d) or weak or strong binders were assessed separately[36].

## Discussion

DNA hypermethylation of promoter CpG islands is associated with loss of gene expression and constitutes a long-known mechanism of functional inactivation of tumor suppressor genes in cancer cells. In the current study, we have comprehensively analyzed the occurrence of BRCA1 promoter hypermethylation in a population-based early-stage TNBC cohort, its readout on the tumor genome and immune microenvironment, and its implications on patient outcomes after adjuvant standard chemotherapy in the context of BRCA1-null tumors.

An important feature of our study is its population-based nature with integrated tissue sampling in conjunction with routine diagnostics[38], exemplified by the similar methylation rates observed across individual years of patient enrollment. This lends support to the reproducibility and generalizability of our results in the context of the studied patient demographics for both molecular and patient outcome findings, despite limited sample numbers for some comparisons. In the total SCAN-B cohort, 24.1% of TNBC patients were BRCA1 hypermethylated, a proportion in the lower to mid-range of previous reports[14–20]. However, in comparison to the observed BRCA1-null rate of 10.5% among all SCAN-B patients, BRCA1 hypermethylation is more than twice as frequent. Our analyses demonstrate a high frequency of BRCA1 hypermethylation in young patients (46.2% of TNBC patients less than 50 years). These women are often referred to clinical germline BRCA-screening, which is increasingly being considered for guiding the surgical procedure and post-operative treatment[39]. Indeed, almost half of the women who had undergone clinical BRCA1/2 sequencing without findings of pathogenic germline variants had a tumor with BRCA1 hypermethylation. These women had no clear family history of breast/ovarian cancer but had an age of diagnosis similar to patients with germline BRCA1 alterations. Moreover, they had a low but elevated level of BRCA1 promoter methylation also in peripheral blood DNA, an observation confirmed in additional clinically unscreened SCAN-B patients of varying age. Although these results must be interpreted with caution as the hypermethylation levels are at the limit of detection by pyrosequencing (and thus not suitable for prediction of somatic hypermethylation), they are intriguing in the context of both possible circulating tumor DNA and findings of mosaic constitutional BRCA1

the methods was a statistical difference in cell type proportions observed between hypermethylated and BRCA1-null tumors for any cell population (t-test, $p > 0.05$, data provided in Supplementary Data 1). Supporting these observations, the expression of 102 immune cell type associated genes was analyzed by unsupervised clustering and supervised differential gene expression in the 82 SCAN-B cases. While groups of tumors displayed distinct immune infiltration (e.g., consistent with the immune modulatory TNBCtype subtype) these were not defined by BRCA1-status or PD-L1 immunohistochemistry status, nor was any gene statistically different in expression between sample groups (FDR adjusted Wilcoxon's test $p > 0.05$) (Supplementary Fig. 5).

Next, PD-L1 immunohistochemistry scores (using the Roche SP-142 antibody) were examined in 53 BRCA1 hypermethylated

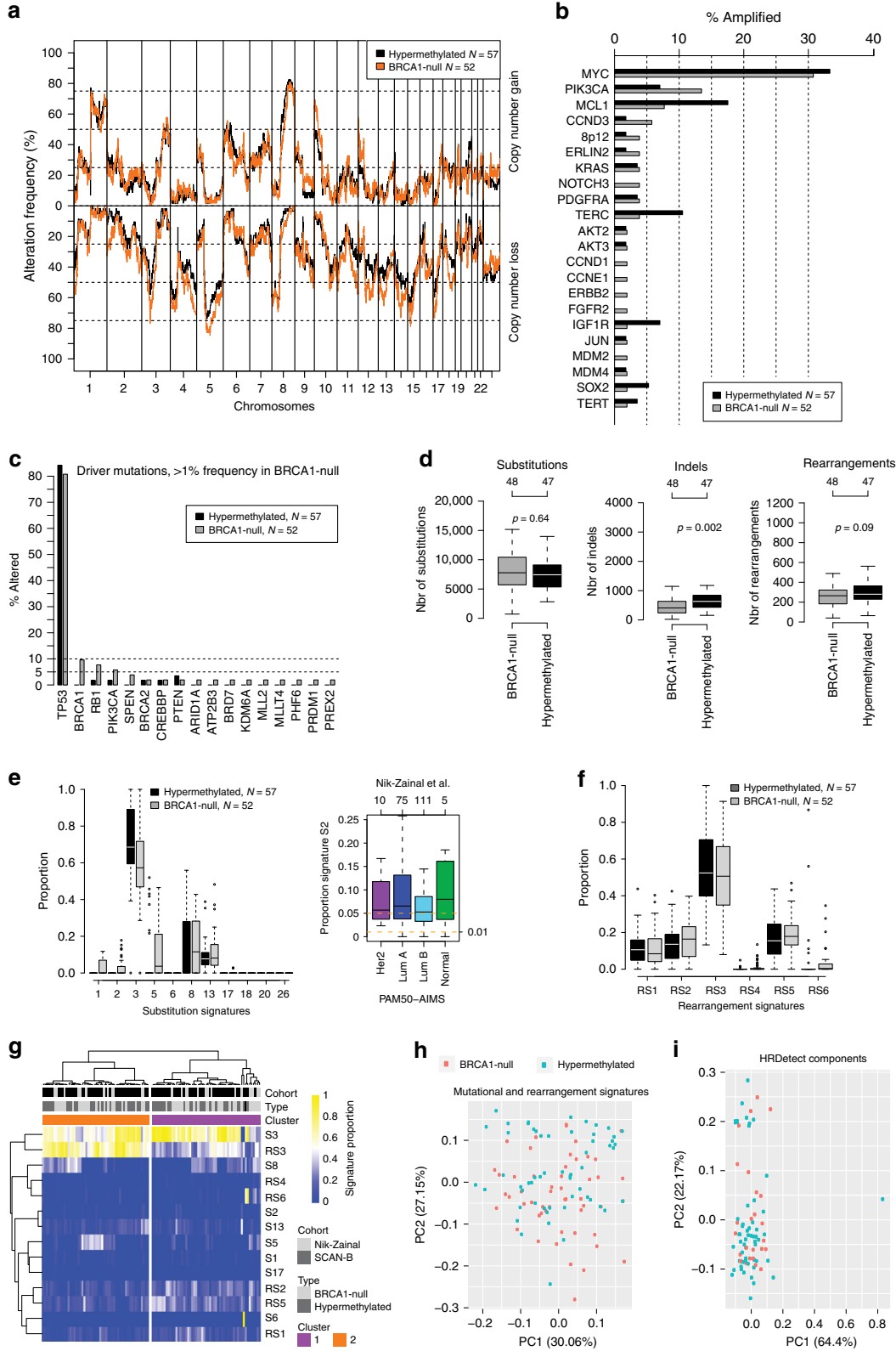

hypermethylation. Regarding the latter, mosaic constitutional *BRCA1* hypermethylation has been reported in 4–7% of newborn females[40,41] as well as promoter methylation of *BRCA1* or other cancer-related genes in peripheral blood in women who developed TNBC or high grade serous ovarian cancer[42–45] (see also Tang et al.[46] for review). Unfortunately, a lack of corresponding fresh normal tissue hindered us from analyzing whether a mosaic

methylation pattern was present also in non-malignant breast tissue. If elevated *BRCA1* hypermethylation levels in blood cells constitute a risk factor for TNBC (and ovarian cancer) development and are present prior to tumorigenesis, these observations raise important questions regarding potential screening, prevention and early detection, requiring development of more sensitive assays for patient classification. Irrespectively, our results

**Fig. 4 Genetic phenotypes of *BRCA1* hypermethylated and *BRCA1*-null TNBC.** The 57 hypermethylated and 25 *BRCA1*-null SCAN-B cases were combined with 27 *BRCA1*-null cases from Nik-Zainal et al.[24]. **a** Frequency of copy number alterations across the genome for *BRCA1* hypermethylated and *BRCA1*-null cases. **b** Frequency of copy number amplification for driver genes defined in Nik-Zainal et al.[24]. **c** Frequency of mutations (insertions, deletions, substitutions) for driver genes defined in Nik-Zainal et al.[24]. Only genes with >1% alteration in the *BRCA1*-null cohort are shown. Displayed mutations in *BRCA1* are somatic. **d** Total number of substitutions, indels, and rearrangements per sample for *BRCA1* hypermethylated versus *BRCA1*-null groups. Only cases sequenced to at least 30-fold depth included. *p* Values calculated using Wilcoxon's test. Top axes show number of cases per group. **e** Left panel shows distribution of mutational signature (defined in Nik-Zainal et al.[24]) proportions per sample between hypermethylated versus *BRCA1*-null cases. Proportions are calculated as the number of substitutions for a signature divided by the total number of substitutions from all signatures. Right panel shows proportion of the APOBEC Substitution Signature 2 in non-basal-like tumors from Nik-Zainal et al.[24]. Top axis indicates number of samples per group. All outliers are not shown due to *y*-axis scale. **f** Distribution of rearrangement signature[24] proportions per sample between hypermethylated and *BRCA1*-null cases. **g** Hierarchical clustering of combined substitution and rearrangement signature proportions using Pearson correlation and Ward.D linkage in the 109 combined cases. **h** Principal component analysis of proportions of substitution and rearrangement signatures in the 109 combined cases, illustrated by the first two principal components representing most variation. **i** Principal component analysis of the proportions of the contributions of HRDetect components (as defined in ref. [26]) per sample (obtained from[22]), illustrated by the first two principal components representing most variation. The analysis only included the 25 *BRCA1*-null and 57 hypermethylated SCAN-B cases. All *p* values reported from statistical tests are two-sided. Source data are provided as a Source Data file.

implicate *BRCA1* hypermethylation as the likely causative factor underlying a significant proportion of women with TNBC, particularly in those without a family history of disease.

In agreement with previous studies[19,23,24,47], *BRCA1* hypermethylation was strongly associated with classical features of *BRCA1*-null and basal-like disease. Despite this, *BRCA1* promoter hypermethylation was also present in subgroups such as old patients and in molecular subtypes not commonly associated with *BRCA1* deficiency in TNBC, albeit at lower frequency. The latter indicates that current gene expression phenotypes are not a perfect surrogate for identification of *BRCA1* deficiency. No case of combined *BRCA1* germline/somatic mutation and hypermethylation was observed, suggesting that these are separate routes for gene inactivation and that loss of heterozygosity or deletion of the wild type *BRCA1* copy is the dominant second hit in both hypermethylated and mutated cases. Moreover, the close relationship between *BRCA1* promoter allele methylation levels and tumor cell content support a view that the hypermethylation is already present in the main clone, as opposed to subclonal, and therefore early in evolution of the tumor (Fig. 2d). While *BRCA1* hypermethylation and germline/somatic mutations represent separate mechanisms for gene inactivation, an objective of this study was to explicitly and in detail test the hypothesis that these alterations result in similar genomic phenotypes. Based on comprehensive global genetic, gene expression, and DNA methylation analyses we found support for this hypothesis. *BRCA1* hypermethylated cases share, with few exceptions, similar frequencies of copy number alterations, proportions of reported driver genes in breast cancer, tumor mutational burden, mutational signatures, rearrangement signatures, gene expression patterns, immune cell infiltration, and DNA methylation patterns with *BRCA1*-null cases.

There are conflicting reports in the literature about whether *BRCA1* hypermethylation is associated with a better or worse prognosis in early-stage disease (e.g., refs. [14–16,18,19,21]). In metastatic TNBC, *BRCA1* hypermethylation has been reported not to be associated with a better treatment response to carboplatin compared with docetaxel[48] (although hypermethylation in this study was measured in the primary tumor tissue, not the actual metastatic tissue) or to single-agent carboplatin effect[49]. This has led to a hypothesis of a soft/plastic BRCAness phenotype for hypermethylated patients compared with germline *BRCA1*-altered cases[48], which remains to be proven by analysis of matched tissue from primary and relapsed tumors. Our survival analyses show that *BRCA1* promoter methylation appears as a marker for favorable response to current conventional adjuvant chemotherapy in early-stage TNBC, with a prognosis similar to

that of *BRCA1*-null cases. When considering the observed survival curves for hypermethylated cases and the typical early relapse pattern of TNBC, the borderline nonsignificant statistical findings for *BRCA1* hypermethylation are likely due to a relatively short follow-up time in the SCAN-B cohort (59% of cases have ≤5 years follow-up) and the limited sample size of the study. Similarly, we saw significant or borderline nonsignificant patterns also for other clinical endpoints like overall survival (OS) and DRFI. Moreover, the observation of *BRCA1* hypermethylation also in old TNBC patients, often excluded from adjuvant therapy and clinical trials, may warrant reconsideration in the clinical management of such patients. The explanation for the improved prognosis of *BRCA1* hypermethylated versus non-*BRCA1* altered cases may lie in the exclusive association with an HRD phenotype as demonstrated in our prior study[22]. A growing body of evidence suggests that TNBCs with an HRD phenotype (including non-*BRCA1/2* mutated cases with HRD inferred by other alterations) respond better to DNA damaging agents[3,13,22,50,51]. In this context, our observed high incidence of *BRCA1* hypermethylation implies that *BRCA1* hypermethylation likely constitutes the most common underlying cause of HRD in unselected early-stage TNBC. Recent randomized clinical trial data in ovarian cancer has showed that HRD-positive tumors without *BRCA1/2* alterations respond favorably to PARP-inhibitors as first-line maintenance therapy[8]. Similarly, advanced breast cancer patients with HRDetect-high tumors have been shown to be associated with clinical improvement on platinum-based chemotherapy[50], suggesting that also early stage patients with HRD-positive hypermethylated tumors may benefit from tailored therapies. If so, the early stage patient cohort that could be considered for tailored treatment would increase considerably, although the full treatment effect remains to be determined in clinical trials.

TNBC tumors have been associated with high TIL-levels (signaling immunogenicity) which are also prognostically favorable[52,53]. A recent study has suggested a higher frequency of TIL-positive tumors in *BRCA1/2*-mutated patients than in wild-type cases[54]. In the SCAN-B cohort, TIL-levels did not differ between hypermethylated and *BRCA1*-null cases, which is consistent with the similar prognosis in these groups after adjuvant chemotherapy. PD-L1 expression is also frequent in TNBC, including *BRCA1* hypermethylated tumors[19], but studies have not found that levels of PD-L1-positive TILs in TNBC cancers are driven by a high mutation rate or by *BRCA1* mutation status[52,54]. This lack of association may now be appreciated through our delineation of the *BRCA1/2*-wt subgroup by *BRCA1* hypermethylation, forming a major subgroup of TNBC with similar genetic and immune cell phenotypes as *BRCA1*-null cases. Taken

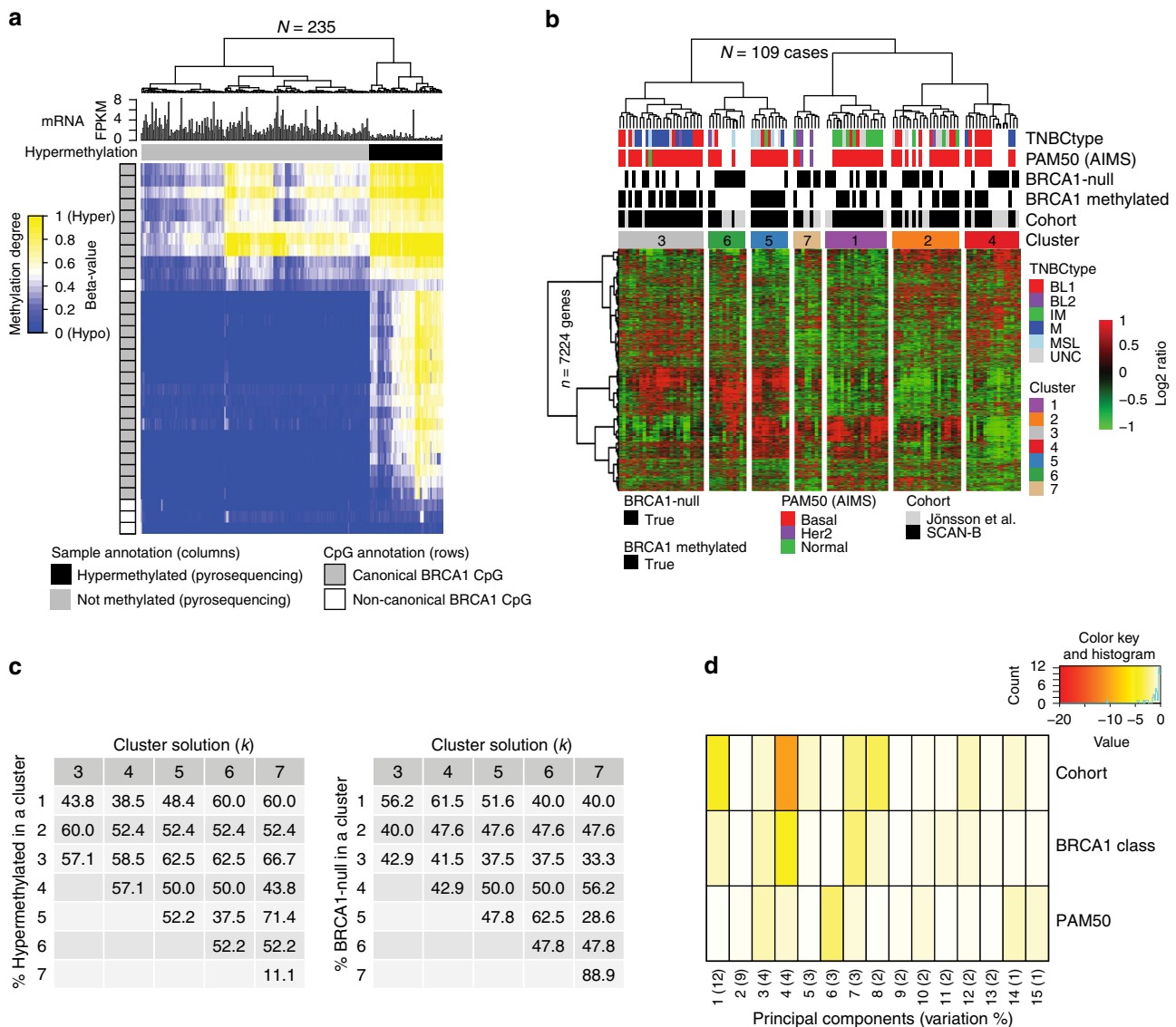

**Fig. 5 DNA methylation and transcriptional characteristics in *BRCA1* hypermethylated and *BRCA1*-null TNBC. a** Hierarchical clustering (ward.D2 linkage, Euclidean distance) of DNA methylation data (beta-values, heatmap) from 235 SCAN-B cases for 32 CpGs associated with the *BRCA1* gene. The 32 CpGs were differentially methylated between 25 *BRCA1*-null and 57 *BRCA1* hypermethylated SCAN-B cases through supervised analysis of Illumina EPIC data. Black sample annotations correspond to hypermethylated cases by pyrosequencing, gray to unmethylated. Gray CpG annotation bars (rows) indicate belonging to the canonical *BRCA1* promoter (+500 to −1500 base pairs, chr17:43124984–43126983). *BRCA1* mRNA expression (FPKM) is shown for each case as bar plot expression above the CpG heatmap. **b** Unsupervised hierarchical gene expression-based clustering of 52 *BRCA1*-null and 57 *BRCA1* hypermethylated cases combined from the SCAN-B and Jönsson et al.[23] cohorts using Pearson correlation as distance and Ward.D linkage based on 7224 genes with standard deviation >0.6 in mean-centered expression across all samples. Colored boxes indicate the seven top subclusters defined from the hierarchical tree. TNBCtype subtypes[31]; BL1, basal-like 1: BL 2, basal-like 2: IM, immunomodulatory: M, mesenchymal: MSL, mesenchymal stem-like: UNC, uncertain. PAM50 and TNBCtype subtypes were not available for the Jönsson et al. cases. **c** Summarized results from consensus clustering, using a range between 3 and 7 cluster solutions (k, columns) and the same gene set and samples as in (**b**). For each cluster solution, the percentage of *BRCA1* hypermethylated (left matrix) and *BRCA1*-null (right matrix) cases in the different defined clusters are shown (rows). **d** Principal component analysis based on the expression of the 7224 genes to relate variance in gene expression with *BRCA1* status. A more intense red color for a principal component (columns) with a variable (rows) implies a stronger association with variance in expression for that component. Components represent variation in decreasing strength (first component largest variation with contributed percentage variation listed). A variable strongly associated with variation in expression is thus represented by an intense red principal component capturing a high percentage of the variation. All *p* values reported from statistical tests are two-sided. Source data are provided as a Source Data file.

together, on a group level these results suggest that early-stage hypermethylated cases may have similar response to immune checkpoint inhibitors as the *BRCA1*-null group. Both groups are, however, clearly heterogeneous and include both inflamed and cold tumors with and without PD-L1 expression. Consequently, the trend of higher PD-L1 levels in *BRCA1*-null tumors compared

with hypermethylated cases warrants further investigation, as does the intersection between HRD-positivity and a variable immunogenicity.

In summary, our analyses show that the genomic characteristics of *BRCA1* hypermethylated and *BRCA1*-null TNBCs are highly similar, thus representing genomic phenocopies. This

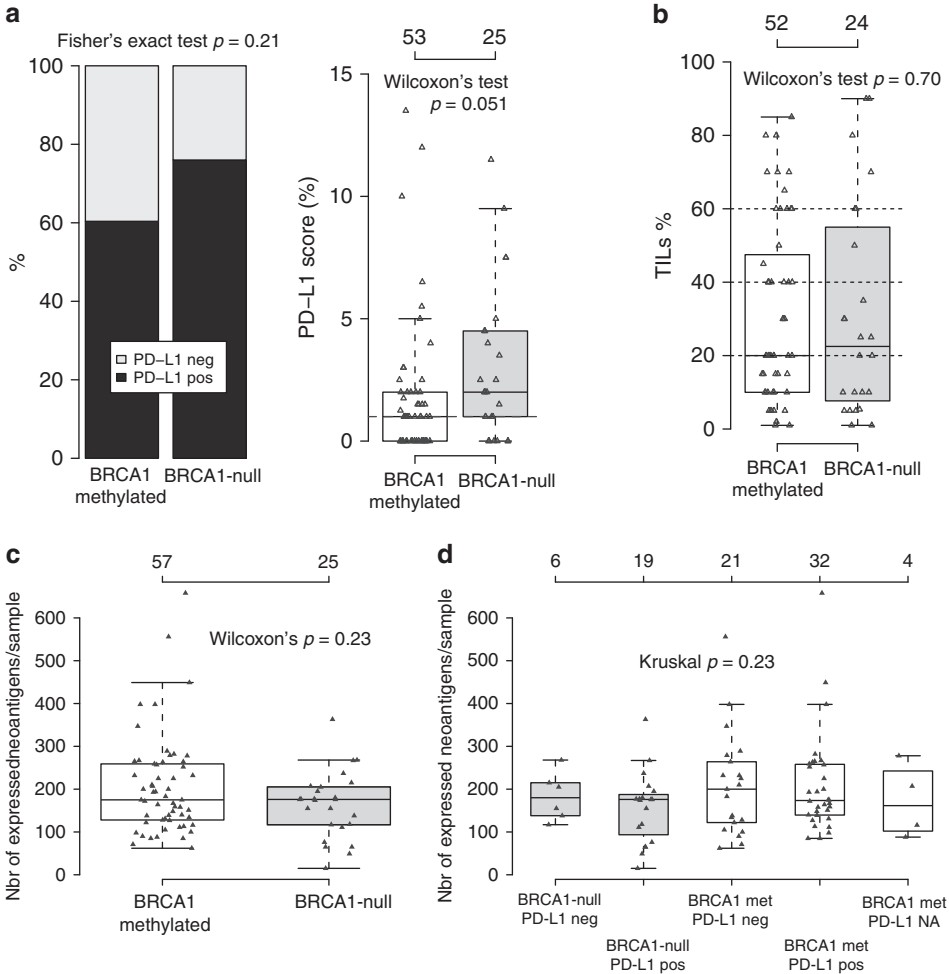

**Fig. 6 Immune cell infiltration phenotypes in *BRCA1*-null and hypermethylated cases. a** PD-L1 scoring of 53 *BRCA1* hypermethylated and 25 *BRCA1*-null SCAN-B cases using the Roche SP-142 antibody that is evaluated in immune cells. To the left, proportion of positive cases (≥1%), to the right distribution of actual PD-L1 scores for the two groups. **b** Scoring of tumor infiltrating lymphocytes (TILs) in 52 *BRCA1* hypermethylated and 24 *BRCA1*-null SCAN-B cases based on available whole section H&E slides. **c** Total number of expressed neoantigens per sample as calculated from substitutions by NeoPredPipe[36] for *BRCA1*-null and *BRCA1* hypermethylated SCAN-B cases. **d** Neoantigens as shown in (**c**), but stratified for sample groups also by PD-L1 IHC status. Four hypermethylated cases did not have available PD-L1 immunohistochemistry data. All *p* values reported from statistical tests are two-sided. Source data are provided as a Source Data file.

suggests that *BRCA1* hypermethylation is an early event in tumor development and that tumor progression proceeds along pathways common with *BRCA1* germline mutated cases. TNBC patients with *BRCA1* hypermethylated tumors share a similar beneficial outcome after standard of care adjuvant chemotherapy as *BRCA1*-null patients, suggesting that *BRCA1* hypermethylation may represent a DNA based prognostic biomarker that is detectable also in low-cellularity tumor tissue specimens. Whether *BRCA1* hypermethylated early-stage TNBC patients respond similar as *BRCA1*-null patients to platinum-based chemotherapy regimens, PARP inhibitors, or immune checkpoint inhibitors merits investigation. A key component in utilizing findings from this study is to include methylation testing in routine clinical diagnostics, requiring robust assays applicable to archival tissue to be developed, and to develop sensitive assays for further investigation of hypermethylation patterns in peripheral blood and their potential association with disease risk.

## Methods

**Ethics approval and consent to participate**. Patients were enrolled in the Sweden Cancerome Analysis Network—Breast (SCAN-B) study (ClinicalTrials.gov ID NCT02306096)[38,55,56], approved by the Regional Ethical Review Board in Lund,

Sweden (Registration numbers 2009/658, 2010/383, 2012/58, 2016/742, 2018/267, and 2019/01252). All patients provided written informed consent prior to enrollment. All analyses were performed in accordance with patient consent and ethical regulations and decisions.

**SCAN-B patient cohort**. From our[22] recently reported Swedish population representative TNBC cohort (SCAN-B) of 237 patients we identified 57 *BRCA1* hypermethylated cases based on pyrosequencing and 25 *BRCA1*-null cases by WGS. All identified cases had complete WGS, RNA sequencing, global DNA methylation profiles (performed for this study), and extensive clinical follow-up data. A CONSORT diagram is provided in Supplementary Fig. 1. Clinicopathological characteristics for the 82 SCAN-B patients with genetic or epigenetic *BRCA1* alterations are summarized in Table 1 and detailed in Supplementary Data 1. Of the 237 SCAN-B patients, 46 patients underwent clinical screening for germline *BRCA1* or *BRCA2* alterations based on national guidelines at the time.

**BRCA1 promoter hypermethylation analysis**. *BRCA1* pyrosequencing promoter hypermethylation status was available from a prior study[22] based on a protocol reported by Saal et al.[57]. Briefly, the pyrosequencing assay involved analysis of two CpG-dense amplicons containing five and four CpGs, respectively, with internal controls for bisulfite treatment (nucleotide positions are relative to the +1 adenine of the ATG translational start codon on the annotated Ensembl v36 *BRCA1* genomic sequence): *BRCA1* region 1, −1291 to −1267 (containing 5 CpGs); and region 2, −1332 to −1310 (4 CpGs). The following primers were used: BRCA1_Region1/2_PCR_F, 5′-NNTATTTTGAGAGGTTGTTGTT TAG-3′; BRCA1_Region1/2_PCR_R, 5′-biotin-TAA AAAACCCCACAACCTATCC-3′;

BRCA1_Region1 _Seq, 5′-GAATTATAGATAAATTAAAA-3′; and BRCA1_R-egion2_Seq, 5′-GGTAGTTTTTTGGTTT T-3′. CpGs covered by the assay include cg16630982, cg15419295, cg16963062, cg20187250, cg04658354, cg04110421, and cg21253966. Methylated and unmethylated controls were included in each bisulfite conversion and pyrosequencing run. For the methylated control samples, across ten pyrosequencing tumor runs, the mean and standard deviation for the percentage of BRCA1 allele methylation was 95.48 ± 1.22 for primer set 1 and 96.16 ± 1.32 for primer set 2. For the unmethylated control, the corresponding values were 2.68 ± 0.75 for primer set 1 and 1.55 ± 0.90 for primer set 2. The combined methylation estimates for unmethylated samples (2.26 ± SD 0.45; max 3.59%) are therefore in line with the estimates for the negative conversion controls. Taken together, this demonstrates that the bisulfite conversion efficiency is sufficiently high to ensure to avoid false-positive methylation results. A cut-off of 7% in allele methylation was used to call samples as hypermethylated or not. The chosen cut-off is between the highest methylation level of 3.6% for patients classified as unmethylated (mean 2.3% and standard deviation 0.48%), and the lowest observed methylation level of 13.6% for hypermethylated patients. Importantly, the 7% cut-off is relevant for pyrosequencing data performed according to the specified protocol. It may therefore not be suitable for calling BRCA1 promoter hypermethylation in Illumina Infinium Methylation beadchips.

In addition, pyrosequencing based BRCA1 hypermethylation analysis of blood DNA for constitutional methylation was first performed for 34 of the 46 patients that underwent clinical screening without germline BRCA1/2 findings. DNA from matched blood samples were extracted using the Qiagen Midiprep kit according to manufacturer's instructions at the Labmedicin Skåne Biobank, Lund, Sweden. Experiments were performed as a single batch experiment including bisulfite conversion and pyrosequencing (Supplementary Methods). In a second analysis, the original 34 patients were rerun together with 71 additional patients from the SCAN-B cohort ($n = 105$ in total) covering in total 55 of the 57 included cases with BRCA1 tumor hypermethylation. This second run was performed using the exact same instrument settings, reagent lots, and PCR product amounts for all cases, but using a separate bisulfite conversion for the 71 new cases.

**Global epigenetic profiling.** Global epigenetic profiling of BRCA1 hypermethylated and BRCA1-null cases were performed using Illumina Infinium MethylationEPIC beadchips (interrogating ~800,000 CpGs) according to manufacturer's instructions at the Center for Translational Genomics, Lund University and Clinical Genomics Lund, SciLifeLab. Input DNA was the same as for WGS, extracted from tumor tissue preserved in RNAlater (Qiagen, Hilden, Germany)[38,55]. Beadchip data analysis and immune cell deconvolution based on DNA methylation data using EpiDISH with Robust Partial Correlations (RPC)[35] were performed as detailed I Supplementary Methods. M-values were defined as log2(Beta/(1-Beta)) in line with[58].

**Gene-expression analyses.** Gene-expression profiling was performed using RNA sequencing (RNAseq) as outlined[55], and data has been reported elsewhere as Fragments Per Kilobase per Million reads (FPKM)[59,60]. Supervised and unsupervised analyses were performed as outlined in Supplementary Methods. Molecular subtype classification according to PAM50[61], IntClust 10[62], CIT[29], and reported TNBC subtypes (TNBCtype)[31,63] were obtained from[22]. To increase the number of BRCA1-null tumors in gene expression analyses for greater statistical power we generated new RNA sequencing data (according to the same experimental and data analysis protocol as for SCAN-B cases) from 27 ER- and PR-negative BRCA1 germline cases reported originally by Jönsson et al.[23]. These were added to the SCAN-B cases ($n = 52$ BRCA1-null in total). Immune cell deconvolution was performed for RNAseq data using xCell[34] and CIBERSORTx[33] as outlined in Supplementary Methods.

**Whole-genome sequencing analyses.** Whole-genome sequencing data for BRCA1 hypermethylated and BRCA1-null SCAN-B cases, including mutational calls, mutational and rearrangement signatures, copy number profiles, and HRD classifications (HRDetect[26] and genomic scars, HRD-score[13]) were obtained from ref.[22] (Supplementary Methods). Eighty-four percent of BRCA1-null and 82.5% of BRCA1 hypermethylated cases were sequenced with a 30X target depth, while the remaining cases were sequenced with a 15X target depth.

**Expanded BRCA1-null whole genome sequencing cohort.** To expand the BRCA1-null cohort for analyses based on WGS data we used the reported cohort of 560 WGS analyzed breast cancers by Nik-Zainal et al.[24]. In this cohort, 27 TNBC tumors were identified as germline BRCA1-null and added to the SCAN-B cases ($n = 52$ BRCA1-null cases in total).

**PD-L1 immunohistochemistry and TILs.** PD-L1 immunohistochemistry using the SP-142 antibody (Roche) was performed on a tissue microarray including 53 BRCA1 hypermethylated and 25 BRCA1-null SCAN-B tumors (two 1 mm cores/tumor) on a Ventana instrument (Roche) according to manufacturer's recommendations. The SP-142 antibody is the intended PD-L1 test for atezolizumab checkpoint inhibitor used in the Impassion 130 trial[64]. PD-L1 assessment was performed according to antibody instructions on immune cells by a board certified

breast cancer pathologist, using a ≥1% cut-off for positivity. Negative cases in both TMA cores were set to score 0.

TILs were scored on available whole section formalin-fixed paraffin embedded haematoxylin and eosin (H&E) slides by a board certified breast cancer pathologist and summarized as a percentage per sample. Scoring was performed according to the international consensus scoring recommendations of the International Immuno-Oncology Biomarker Working Group on Breast Cancer (www.tilsinbreastcancer.org). Scores were averaged when multiple slides were available per patient. All pathology scorings were performed blinded to clinicopathological and BRCA1 classifications.

**Neoantigen prediction.** NeoPredPipe[36] was used to predict putative neoantigens with substitution mutation calls provided by CaVEMan (https://cancerit.github.io/CaVEMan/) and HLA typing done with Polysolver[37] as input. hg19 was used as the human reference genome throughout all analysis for the neoantigen predictions. NeoPredPipe was run with default parameters except that options -c 1 2 -m where set. Polysolver was run on WGS data from blood DNA with options: (i) unknown ethnicity, (ii) use population-level allele frequencies as priors, and (iii) do not use empirical insert size distribution. Only variants with a PASS flag in the variant call file from CaVEMan was used as input to NeoPredPipe. Integration with RNAseq expression was done as outlined for NeoPredPipe, and only neoantigens with an expression >0.1 was kept for final analysis.

**Statistical analyses.** Survival analyses were performed in R (ver. 3.6.1) using the survival package with OS, IDFS, or DRFI as endpoints defined according to the STEEP guidelines[65] (see Supplementary Methods for endpoint definitions and analysis exclusion criteria). Survival curves were compared using Kaplan–Meier estimates and the log-rank test. Hazard ratios were calculated through univariable or multivariable Cox regression. Proportional hazard assumptions were tested using the cox.zph function in the R survival package (ver. 3.1-12). Survival analyses were performed using the 149 eligible cases (62.8%) from the total SCAN-B cohort treated with standard-of-care adjuvant chemotherapy according to national guidelines (in 96% of cases a FEC-based [combination of five fluorouracil, epirubicin, and cyclophosphamide] treatment ± a taxane) to contrast BRCA1-deficient groups versus non-altered patients. Full details on the exclusion criteria for outcome analysis and individual patient treatment are available in Staaf et al.[22]. All p values reported from statistical tests are two-sided if not otherwise specified. Box-plot elements corresponds to: (i) center line = median, (ii) box limits = upper and lower quartiles, (iii) whiskers = 1.5× interquartile range.

**Reporting summary.** Further information on research design is available in the Nature Research Reporting Summary linked to this article.

## Data availability
For BRCA1 hypermethylated and BRCA1-null SCAN-B cases mapping of clinical and molecular classifications to gene expression data GSE96058 [https://www.ncbi.nlm.nih.gov/geo/query/acc.cgi?acc=GSE96058] is available in Supplementary Data 1. Illumina DNA methylation data for BRCA1 hypermethylated and BRCA1-null SCAN-B cases is available as GSE148748 [https://www.ncbi.nlm.nih.gov/geo/query/acc.cgi?acc=GSE148748] at Gene Expression Omnibus. FPKM RNAseq data for the additional 27 BRCA1-mutated cases from Jönsson et al.[23] is available in an online repository associated with this study [https://doi.org/10.17632/2dbh285999.1]. Source data are provided with this paper. The source data underlying Figs. 2a–d, 3a–d, 4a, 4d–g, 5a–d, and 6a, b are provided as a Source Data file. Source data are provided with this paper.

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

## Acknowledgements

The authors would like to acknowledge the Center for Translational Genomics, Lund University and Clinical Genomics Lund, SciLifeLab for support with DNA methylation analyses, patients and clinicians participating in the SCAN-B study, personnel at the central SCAN-B laboratory at the Division of Oncology and Pathology, Lund University, the Swedish National Breast Cancer Quality Registry (NKBC), Regional Cancer Center South, RBC Syd, and the South Sweden Breast Cancer Group (SSBCG). *Financial support*: The funders had no role in study design, data collection and analysis, decision to publish, or preparation of the paper. *Ana Bosch*: The Governmental Funding for Young Clinical Researchers within the National Health Service (ALF) 2017–2019. Mrs. Berta Kamprad Foundation FBKS 2017-34-199. *Shamik Mitra*: Shamik Mitra is financially supported by the funding received from the European Community's Horizon 2020 Framework Program for Research and Innovation (H2020-MSCA-ITN-2014) under Grant Agreement no. 247634. *Mattias Aine*: Gunnar Nilsson Foundation grant (GN-2018-5). *Niklas Loman*. The Swedish research Council 2019–2021. The Governmental Funding within the National Health Service (ALF). Mrs. Berta Kamprad Foundation. *Åke Borg*: The Swedish Cancer Society CAN 2016/659. Mrs. Berta Kamprad Foundation FBKS 2018-3-166. Mats Paulsson Foundation IACD 2017. Lund-Lausanne L2-Bridge/Biltema Foundation (F 2016/1330). *Johan Staaf*: The Governmental Funding of Clinical Research within the National Health Service (ALF) 2018/40612. The Swedish Cancer Society (CAN 2018/685) and a 2018 Senior Investigator Award (SIA190013). Mrs. Berta Kamprad Foundation FBKS-2018-4-146. The Craafoord Foundation 20180543. The Gustav V:s Jubilee Foundation (174271 and 187041). The research foundation at Department of Oncology in Lund. *Serena Nik-Zainal*: Wellcome Trust Intermediate Clinical Fellowship WT100183MA. CRUK Advanced Clinician Scientist Award (C60100/A23916). CRUK Grand Challenge Award (C60100/A25274). Open access funding provided by Lund University. *Funding*: Financial support for this study was provided by the Swedish Cancer Society (CAN 2016/659 and CAN 2018/685 and a 2018 Senior Investigator Award [JS: SIA190013]), the Mrs Berta Kamprad Foundation (FBKS 2018-3-166 and FBKS-2018-4-146), the Lund-Lausanne L2-Bridge/Biltema Foundation (F 2016/1330), the Mats Paulsson Foundation (IACD 2017), the Craafoord Foundation (grant 20180543), the National Society of Breast Cancer Associations in Sweden, the Swedish Breast Cancer Group (SweBCG), the Swedish Research Council, BioCARE a Strategic Research Program at Lund University, the Gustav V:s Jubilee Foundation (174271), the research foundation at Department of Oncology in Lund, Region Skåne Regional funds, and Swedish governmental funding (ALF, grant 2018/40612). Whole genome sequencing and analysis was funded by a Wellcome Trust Intermediate Clinical Fellowship WT100183MA and a CRUK Advanced Clinician Scientist Award (C60100/A23916) and a CRUK Grand Challenge Award (C60100/A25274).

## Author contributions

Conception and design: J.S. Collection and assembly of data: J.S., S.N.-Z., D.G., A.B., A.K., C.R., K.H., J.V.-C., J.H., C.H. Provision of study material or patients: C.L., N.L., L.R., Å.B., M.M., A.E., L.H.S., C.H., H.E., A.K., E.N. Data analysis and interpretation: J.S., D.G., M.A., S.M., J.H. (IHC evaluations), S.N.-Z. Financial support: S.N.-Z., Å.B., J.S. Administrative support: J.S., S.N.-Z., D.G., J.V.-C., C.H., J.H. Paper writing: All authors. Final approval of the paper: All authors. Agree to be accountable for all aspects of the work: All authors.

## Competing interests

J.H. has received speakers honoraria and travel support from Roche, advisory board fees from MSD, Novartis and Roche, and institutional research grants from Cepheid and Novartis. Anna Ehinger has received speakers honoraria from Novartis, Amgen, Roche, and advisory board fees from Roche. Ana Bosch has participated in advisory boards for Novartis and Pfizer, and has received travel support from Roche. The remaining authors declare no competing interests.
