## [Peer Review File · Nature Communications]

Reviewers' comments:

Reviewer #1 (Remarks to the Author): Expert in TNBC

Glodzik et al., provides a comprehensive comparison between BRCA1 mutated versus BRCA1 methylated TNBC. Given the known heterogeneity of this disease, it is not surprising that no broad characteristics between these groups can be identified. The research is sound and provides valuable information for the community, however, the manuscript would improve if the following points would be addressed:

The cohort composition is rather complex, and the consort diagram needs to be improved to better illustrate the different comparisons. The 237 TNBC consist of 57 BRCA methylated and 25 BRCA null cases of which 19 were germline BRCA ... and the reader needs to assume 6 cases with somatic mutations.

It is not clear to the reviewer what is shown in Figure 1A? Please provide better labels and describe in more detail the message this figure should deliver?

A list of genes for which promoters are hyper or hypomethylated in BRCA1 methylated cases should be provided as a sup table.

Figure 1 B needs to include the number of cases and t-test btw each comparison.

Some germline BRCA1 cases and some methylated cases have high BRCA1-expression. The authors need to discuss how this could be a confounding factor throughout the whole manuscript. Can these tumours be really classified as BRCA1 deficient? For all further analyses, these cases have to be discussed separately.

Please provide a table for each patient, provided for each CpG island its methylation status, the level of BRCA expression, the somatic BRCA mutation, and the germline BRCA mutation status.

Supporting Figure 1C does not show HRD classification?

Which 2 HRD classification algorithms were used – please clarify in the manuscript.

Figure 1D: Since there is such a strong correlation btw tumour content and methylation, the authors need to first correct for this confounding factor or at least include this for any further analyses. Otherwise hypermethylated cases may be underestimated across this cohort. The points for hypermethylated cases in the graph should be in a colour pallet to display the level of BRCA1 expression – maybe this also explains why some cases show expression while methylated.

Are some CpG islands more affected by tumour content than others – could this be informative for testing BRCA methylation in the blood?

On line 303 – first time mentioned BRCA2 cases? Why is this case shown in a paper discussing BRCA1 methylation versus BRCA mutation, this needs further explanation or remove it?

On line 312: 46/237 underwent BRCA-screening – 34 were wt BRCA cases (8 were germline BRCA cases), correct? Please report explicitly.

– 16/34 cases were methylated in the blood – again please state explicitly why not all 46 screened patients were also part of blood methylation test.

For the reader it would be easier, to always clearly mention, if the author means blood or tumour-methylations.

Do all 16 blood-methylated patients have tumours with BRCA1-methylation?

Do any of the 18 hypomethylated cases have tumour BRCA1-methylation?

- AI line plot of beta-values should be provided, illustrating patient-matched methylation status

- Figure 2B Wilcoxon test is not informative here

-

Then additional 39 /70 cases showed BRCA blood-methylation – here the authors should again show the relationship btw blood and tumours BRCA methylation.

Are there cases in which BRCA is not methylated in tumours but it is in the blood – what is the BRCA expression levels in the tumour?

Given that the very complex relationship btw methylation in tumour and blood, mutation germline and somatic, and expression levels, subanalyses of cases with unambiguous expression – methylation, expression-germline status should be performed, see whether this further supports similarities or identifies differences between mutated and methylated BRCA cases.

A clear cohort definition needs to be made at the end of this section so that the reader gets a clear picture of what will be compared from now onwards.

Figure 2E needs to include barplots for BRCA null cases and no-BRCA cases, to reflect Table 2.

Age seems to be an interesting factor with regards to BRCA1 methylation and comes up in several parts of this manuscript. It would be good to combine it and discuss it once. Also BRCA2 needs to be specially mentioned if it is shown such as in Figure 2D or taken out.

More detailed information to Figure 2E & F needs to be provided, as there is lots of data shown and not discussed. How many BRCA-tumour methylated and BRCA carriers are included in 149 survival study?

Looking at Sup Fig2 – it is not clear what is BRCA altered – is this BRCA null e.g germline and somatic? BRCA2 should be excluded in this analyses – numbers of cases should be provided in Figure 2F forest plot

Figure 2 G Kaplan Meier curves cross and violate KM analyses, thus need to apply different survival analyses e.g restricted mean.

Copy number analyses – can the authors be sure that none of 52 BRCA null cases shows BRCA-methylation? Fisher tests need to be applied to see if there are any differences in losses or gains, and a sup table should be provided.

The authors should revise the TNT trial analyses, as BRCA methylated cases showed a better response in docetaxel (Tutt et al., Nat Med Figure 2D), and need to revise their statement. Careful assessment of clinical trial data should be performed and discussion should be extended.

Figure 4A is a redundant display to the supervised analyses. The individual 32 CpGs should be shown for each case, so the reader can see whether there are changes across them.

As mentioned above, the expression of BRCA1 needs to be shown to see if it confounds the analyses or not.

Can the authors add a table to illustrate correlation btw gene expression and methylation?

Figure 4B – can the authors provide a heatmap and gene lists what is different btw these 7 groups?

The gene list of differential gene expression should be provided as a sup table

What are the 8 differentially expressed genes, and what is their methylation or copy number

status?

It is not clear in the CONSORT diagram where one finds the 82 SCAN-B cases for immune feature gene expression analyses?

Figure 4E – again does the trend shift to significance if cases with BRCA hypermethylation but BRCA expression are removed? Please show in the graph and discuss in the manuscript.

TILs, immune gene expression and PDL1 expression could be used in more complex integrative analyses. Which gene expression goes with PDL1 – does this help to decipher small differences which are missed when these rather global analyses measures are applied?

A table for each case which TILs score, PDL1 score, and CIBERSORT population proportion should be provided as sup table.

References are in a different font
Reference 23 is not complete

Table 2 E Subtypes without any methylated cases are not listed. Not clear what this should mean?

Reviewer #2 (Remarks to the Author): Expert in methylation

In their manuscript, Dominique Glodzik and co-workers explored and compared the clinical, genetic, transcriptomic and epigenomic profiles of BRCA1-null and BRCA1 promoter hypermethylated triple negative breast cancer samples, leading to the basic conclusion that both subgroups essentially present the same phenotype. This has important clinical implications, particularly regarding treatment management. Overall, the paper is clear and well-written, and I have no major remarks that are likely to affect the general conclusion. Nevertheless, I have several comments that should be addressed and/or require additional clarification.

The exact definition of BRCA1 promoter hypermethylation is crucial regarding interpretation of the results and future practical use as biomarker, see e.g. Koch et al. Nature Rev Clin Oncol 2018. Therefore, it is essential to put this information in the main text rather than supplementary information and to mention the primer sets used as well as the exact regions targeted (and the CpGs included in the analyses). Moreover, the authors used an arbitrary 7% methylation cut-off rather than 10% "due to cleanness of the data", but this should be further substantiated. I fully agree that any cut-off used will be arbitrary, yet fig 1A suggests that the 10% cut-off would be more suitable (trough is between 0.1 and 0.2), raising the question why the 7% cut-off was selected. Moreover, cf. cleanness of the data & fact that two different bisulfite conversion kits were used, please provide the obtained bisulfite conversion efficiencies for the controls included (mean & sd). Also, to allow for comparison with and practical application in other studies: are these cut-offs also applicable for the Infinium Epic microarrays, for which (combination of) probes? (Latter is perhaps more suited for supplementary).

Other comments that should be addressed:

- Elaborate on the performed "variance filtering for outlier CpGs" (cf. supplementary, global DNA methylation analysis)
- Gene expression analyses (supplementary): why did the authors apply two different offsets, i.e. +1 and +0.1?
- New RNA-seq data was added for 27 BRCA1-null tumors (gene expression analyses), did the authors adjust for the batch effect? Or can the authors demonstrate that the impact is non-substantial? Batch effects may easily obscure other significant differences.

- Line 302 – “BRCA1 hypermethylation was mutually exclusive with BRCA1-null cases”: although I agree that they did not co-occur, calling them “mutually exclusive” requires more evidence (particularly samples), as there’s no reason to assume that they cannot co-occur at lower rates.
- Statistical testing of methylation results (lines 325 – 334): methylation data (beta-values) are heteroskedastic, was this taken into account for when using t-tests and ANOVA? Consider transforming to M-values, as also non-parametric tests suffer from heteroscedasticity. Moreover, the observation that higher BRCA1 methylation in blood occurs independent of age cannot be tested by one-way ANOVA, only by a 2-way ANOVA (preferably on M-values) that also considers the age/status interaction.
- Figure 3D shows a significant difference in number of indels between BRCA1-null & hypermethylated samples, contrasting the statement on lines 401 – 402
- The paper includes many analyses leading to the same conclusion, consider summarizing in one sentence & refer to the supplementary methods (e.g. lines 343 – 349; lines 453 – 462; and associated figures) to improve overall readability.
- Why the >50 years old cut-off, rather than considering age as a continuous variable (or using median as far less “selected” cut-off)? Please provide rationale or modify.

Minor remarks: ⇐⇐

- Abstract, first sentence: “caused by genetic or epigenetic alteration in key pathway genes.” This is obvious, can be removed from abstract.
- Introduction, line 155: study goes further than basic “genomic” phenotype, “omics” phenotype?
- Lines 167-168: hyphens are not required.
- Line 188: “as well as and” (remove “as well as” or “and”)
- Throughput paper: clearly mention whether pathologist(s) were blinded to sample status or not
- Throughout paper: mutational/substitution signatures are referred to by numbers, which are not really informative for the reader, is it possible to refer to those signatures by name/characteristic (and add the number between brackets for completeness)? See e.g. lines 403 to 412, biological/clinical relevance is unclear without additional information.
- Although I agree that low tumor purity will have an impact on e.g. LOH detection, please also mention the range of the tumor purity estimates for those samples where e.g. LOH was successfully detected.
- Supplementary information: as most readers won’t be familiar with the “INCA” technical platform” and the parameters (Swedish-only), please make sure to mention the different options in English as well. At least verify that all selected options are also mentioned in English, leaving the Swedish parameter settings in there for full reproducibility is ok with me.
- Supplementary information: part on DNA promoter methylation analysis requires additional proofreading, e.g. bisulfite vs. bisulphite, “were included at every individual each bisulfite conversion run”, “commercially available controls included in the bisulfite conversion were included in each pyrosequencing run to as quality controls and assessment of methylation thresholds” etc.
- Line 250: “in in” => “in the”
- Line 293: “is rare” => is rare in cancer samples (maybe rather common, but only leads to breast cancer in case of associated LOH or other event)
- Line 314: 16/34 = 47.1% (rather than 47.0%, after rounding)
- Line 318: were => was
- Lines 443 – 444: non-informative, please remove
- Lines 465 – 468: unclear sentence
- Line 520: is already present in the main clone
- How does the machine learning approach (cf. supplementary, expression analyses) fit in the paper?
- Table 2: line 847: “Hard brackets imply <.”? Line 856: UNS: uncertain => “UNC”

Reviewer #3 (Remarks to the Author): Expert in breast cancer genomics

The authors do a comprehensive analysis of methylation status and phenotypes of BRCA mutated breast cancers. This is proposed to be a population based study from within a clinical trial as a unique characteristic. However, prevalence of mutation and methylation have been previously determined in breast cancer populations not otherwise selected. Overall, the concepts are not new as the frequency of mutation and methylation are in accord with previous studies. Further, the correlations with outcomes are with FEC plus minus taxane chemotherapy as compared to more conventional chemotherapy approaches or PARP inhibitors limits the utility of the information to current management. The limited sample size and followup further limits the utility of the data. The observation that methylation can be detected in blood hematopoietic cells is of potential interest and significance. However, while the correlation is statistically significant it does not have sufficient power to preclude the need for tumor testing. Further this has been observed in previous studies. Similarly the PDL1 difference may be of interest but is not significant due to the small sample size. Otherwise the observation of similar characteristics in the methylation and mutation cases again is not surprising. The addition of this sample set and the detailed analysis is of interest. However, it is important to note that over 2500 cases of BRCA1 methylation and associated outcomes and characteristics have already been presented in the literature.

There are no specific concerns with the data or data presentation. The discussion does note the controversies in the field and the contribution of this study appropriately.

Detailed point-by-point response to the reviewers' comments on manuscript NCOMMS-20-07196-T

First, we would like to thank all reviewers for their constructive criticism and suggestions on how to improve and further clarify this manuscript, and the editor for the opportunity to revise our work. We are grateful for the feedback and comments, which have helped us to further improve and strengthen our manuscript.

Based on the reviewers' comments, we have revised the manuscript to address all their concerns, added more detail to the relevant sections of the manuscript, further clarifications, added associated citations to reference literature, and included several new analyses.

Please find below, the detailed point-by-point responses to the reviewers' comments. Reviewers' comments are presented in **bold font** and our responses are shown in regular text. Textual alterations made to the revised version of the manuscript are shown in *italics*, with underlined text showing modifications in existing text when deemed appropriate. All textual alterations are present in a marked-up version of the manuscript. Please note that the actual reference number for a specific literature reference may differ between the reference list in this document and the revised manuscript version that has been resubmitted. Page numbers refer to the revised manuscript (clean file).

We use the original figure/table numbers in this rebuttal, to be consistent with the figure/table numbers referred to by the reviewers. Figure mapping:

1. New figure 1
2. New figure 2: original figure 1
3. New figure 3: original figure 2
4. New figure 4: original figure 3
5. New figure 5: original figure 4 – gene expression
6. New figure 6: original figure 4 – TILs/PD-L1
7. New Table 1: original Table 1
8. New Table 2: original Table 2

SI figure/table mapping:

1. New supplementary Figure S1: original figure S1 - CONSORT diagram
2. New supplementary Figure S2: replaces original figure S2
3. New supplementary Figure S3
4. New supplementary Figure S4
5. New supplementary Figure S5
6. New supplementary Table S1
7. New supplementary Table S2

Finally, we have revised the title of the manuscript to be more in line with the broad comprehensive analysis performed to:

” Comprehensive molecular comparison of BRCA1 hypermethylated and BRCA1 mutated triple negative breast cancers ”

Reviewer #1 (Remarks to the Author): Expert in TNBC

Glodzik et al., provides a comprehensive comparison between BRCA1 mutated versus BRCA1 methylated TNBC. Given the known heterogeneity of this disease, it is not surprising that no broad characteristics between these groups can be identified. The research is sound and provides valuable information for the community, however, the manuscript would improve if the following points would be addressed:

Response: We thank the reviewer for the very detailed and thoughtful review which has greatly improved the quality of the manuscript.

→The cohort composition is rather complex, and the consort diagram needs to be improved to better illustrate the different comparisons. The 237 TNBC consist of 57 BRCA methylated and 25 BRCA null cases of which 19 were germline BRCA ... and the reader needs to assume 6 cases with somatic mutations.

Response: In the revised manuscript we have made Supplementary Figure S1C into a new Figure 1, and revised it to include details about *BRCA1* hypermethylated, *BRCA1*-null, and non-*BRCA1* cases included in the different analyses. Furthermore, we have revised the CONSORT diagram (now Supplementary Figure S1B), adding the details requested, and made a clearer reference to the original study in which explicit details about inclusion and exclusion criteria can be found. Finally, we have revisited the text to make clarifications throughout the manuscript concerning cohort naming, and comparisons made.

→It is not clear to the reviewer what is shown in Figure 1A? Please provide better labels and describe in more detail the message this figure should deliver? A list of genes for which promoters are hyper or hypomethylated in BRCA1 methylated cases should be provided as a sup table.

Response: The original Figure 1A shows the unsupervised clustering (with a heatmap) of beta values from 30 CpGs associated with the *BRCA1* gene (Promoter CpGs: -1500 to +500 bp relative to the TSS) in the Illumina EPIC DNA methylation array. Overlaid is the pyrosequencing classification of *BRCA1* status (black/gray column annotation bars in revised version). Furthermore, gray row labels indicate promoter associated CpGs as annotated by Illumina. As seen in the figure, one cluster is formed that perfectly mimics the pyrosequencing classification supporting the latter by an alternative method (array-based DNA methylation). In the revised manuscript we have updated the figure, updated the figure legend, and rephrased the main text (page 8 in revised manuscript) to improve clarity:

"Pyrosequencing classifications were corroborated by Illumina DNA methylation profiling data for BRCA1 gene associated CpGs (Figure 2A),"

Furthermore, the Source Data file required by the journal now includes all beta-values and CpG identifiers included in the figure.

The second request by the reviewer appears more related to the individual characteristics of *BRCA1* hypermethylated cases rather than to the difference between *BRCA1* hypermethylated and null samples. In this study, we do provide open access to the global DNA methylation data for *BRCA1*-null and hypermethylated cases, making it possible for anyone to perform this analysis (see Data Availability Statement in the revised manuscript).

→Figure 1 B needs to include the number of cases and t-test btw each comparison. Some germline *BRCA1* cases and some methylated cases have high *BRCA1*-expression. The authors need to discuss how this could be a confounding factor throughout the whole manuscript.

Can these tumours be really classified as *BRCA1* deficient? For all further analyses, these cases have to be discussed separately.

Response: Original Figure 1B includes the number of cases for each group (top axis). In the revised manuscript we have added the requested testing to the figure panel. Notably, the pattern observed in original Figure 1B is in agreement with previous studies, e.g. Figure 5 in Yang et al. ¹ that compared mRNA levels in samples with methylated, somatic mutated, germline mutated and wildtype *BRCA1* in ovarian cancer. Promoter methylated cases had the lowest mRNA expression.

As a further response to the first remark, we plotted the *BRCA1* tumor methylation percentage versus the mRNA expression (Rebuttal figure 1 below). As seen, the number of potential *BRCA1* hypermethylation outliers would be one (n=1, please see the WGS profile for the hypermethylated case with highest *BRCA1* FPKM in Rebuttal figure 7 below in response to another remark), obviously precluding any further robust sub-analyses. We have added this rebuttal figure as Supplementary Figure S2A in the revised manuscript.

Rebuttal figure 1. *BRCA1* mRNA expression (FPKM) versus *BRCA1* pyrosequencing methylation levels for 237 cases. FPKM: Fragments Per Kilobase of transcript per Million mapped reads

Next, regarding mRNA expression of *BRCA1* in tumors with inactivating *BRCA1* variants, there are studies in the literature that investigated this question, e.g. Findlay et al. ² (which focused on substitution variants), demonstrating that many missense mutations are likely to be expressed at some level, with some variant types having lower expression. While splice mutations change the expressed isoforms, the total level, considering that our expression data is summarized per gene not per isoform, will likely be less affected. For indels causing frame shifts it may be expected that the majority would result in lower mRNA expression due to nonsense-mediated decay (NMD). We analyzed the *BRCA1* mRNA expression pattern versus mutation type in our cohort and found that hypermethylated cases express as low amounts of *BRCA1* as tumors with *BRCA1* frame-shift, indels, and nonsense variants (Rebuttal figure 2, below). In summary, in response to the first remark we have added these new results to the new Supplementary Figures S2A and S2B, and updated the text in the first result section (including a new reference) (page 8):

“Pyrosequencing classifications were corroborated by Illumina DNA methylation profiling data for *BRCA1* gene associated CpGs (Figure 2A), and markedly reduced *BRCA1* mRNA expression from RNA sequencing for hypermethylated cases (Figure 2B), similar to previous reports ¹, that were in line with expression levels for cases with *BRCA1* frame shift, nonsense and indel variants (Supplementary Figure S2).”

Rebuttal figure 2. *BRCA1* mRNA expression (FPKM) versus hypermethylated cases, cases with *BRCA1* variants grouped according to type, and cases without *BRCA1* alterations (BRCA1-wt) for the merged set of SCAN-B cases (n=237) and the additional *BRCA1*-mutant cases from Jönsson et al. ³ (n=27). Top axis indicates number of cases per group.

Regarding the second remark of whether specific cases can be considered *BRCA1*-deficient or not, the original figure 1D shows the pyrosequencing data for all 237 SCAN-B cases in the cohort. The figure demonstrates the clear correlation between *BRCA1* promoter methylation level and the purity of tumor samples. In the revised manuscript we have updated

original figure 1D using tumor cell content estimates from the Battenberg algorithm which is WGS-based and can account for tumor subclones when estimating tumor copy number (in contrast to the originally used ASCAT method). Here, the linear relationship is even more striking for the hypermethylated cases with $r^2 = 0.84$ (meaning 84% explanation) and slope=0.9 (close to 1) based on simple linear regression. Considering all data presented (e.g., WGS, DNA methylation, and pyrosequencing) it appears unlikely that other SCAN-B samples are hypermethylated. For instance, original figure 1A demonstrates perfect overlap with pyrosequencing classifications when clustering beta-values for CpGs associated with *BRCA1* (Promoter CpGs: -1500 to +500 bp relative transcription start site). It should be stressed that all *BRCA1*-null and hypermethylated cases in this study have been analyzed by WGS and classified according to state-of-the art HRD algorithms (HRDetect). All cases were classified as HRD (acknowledging one case with concurrent MMRd, see ⁴), analyzed for germline alterations in >160 breast cancer and HR related genes, and display highly rearranged genomes and mutational and rearrangement signatures prototypical of *BRCA1*-deficient cancers ^{5, 6} (exemplified by original figure 1E, and Rebuttal figure 7 below). For details about samples and genetic classifications, etc., we refer to our first study of the SCAN-B cohort, Staaf *et al. Nat Med* 2019 ⁴ and ^{5, 6}, as also cited in several places in the submitted manuscript and supplementary information. WGS data from *BRCA1*-null tumors were inspected case by case, and cases carry either inactivating germline variants or somatic variants, with LOH of corresponding allele. Moreover, there was perfect agreement between WGS results and germline clinical screening results for a subset of patients analyzed as outlined in Staaf *et al. Nat Med* 2019 ⁴. Thus, we are confident that these cases are *BRCA1*-deficient.

In summary, it is important to point out that our response to this comment also addresses several other comments from this reviewer, in which specific sub-analyses are requested for samples with e.g. high mRNA expression inactivating variants versus remaining samples. Based on our data, we see that such analyses would be speculative in nature due to small sample sizes (massive sample sizes would be required to have sufficient number of “outliers”) and of questionable value, given available literature and present data. This is an important consideration to later requests raised by the reviewer.

→Please provide a table for each patient, provided for each CpG island its methylation status, the level of BRCA expression, the somatic BRCA mutation, and the germline BRCA mutation status.

Response: Firstly, we would like to point to that all DNA methylation data from the EPIC DNA methylation arrays for *BRCA1*-null and hypermethylated cases in this study, if accepted for publication, will be open access as outlined in the Data Availability Statement. Secondly, in the Source Data file required by the journal we have deposited *BRCA1* EPIC DNA methylation data (beta-values) for all 235 of the 237 analyzed SCAN-B cases shown in original figure 1A. Thirdly, all DNA mutation data and germline status of included samples are publicly available from our first study of the SCAN-B cohort, Staaf *et al. Nat Med* 2019 ⁴.

However, we acknowledge the reviewer’s point and have therefore added a new Supplementary Table S1, summarizing in detail the clinicopathological and molecular characteristics for the 25 *BRCA1*-null and 57 hypermethylated SCAN-B cases focused on in this study. This table includes requests also made by other reviewers and contains a wealth of information about the samples.

→Supporting Figure 1C does not show HRD classification? Which 2 HRD classification algorithms were used – please clarify in the manuscript.

Response: All 237 SCAN-B cases were HRD classified as they were analyzed by WGS in our first study of the SCAN-B cohort⁴. HRD classification information (references) is available in the original Methods section, “Whole genome sequencing analysis” in the text: ‘HRD classifications (HRDetect⁷ and “genomic scars”, HRD-score⁸) were obtained from⁴

→Figure 1D: Since there is such a strong correlation btw tumour content and methylation, the authors need to first correct for this confounding factor or at least include this for any further analyses. Otherwise hypermethylated cases may be underestimated across this cohort. The points for hypermethylated cases in the graph should be in a colour pallet to display the level of BRCA1 expression – maybe this also explains why some cases show expression while methylated.

Response: The reviewer is correct that tumor content and DNA methylation readout is connected when assayed in bulk tumor tissue. In fact, tumor cell content affects all bulk omic analyses, not just DNA methylation. In response to this comment, we tested whether a difference in estimated tumor cell content existed between hypermethylated and BRCA1-null cases and found that this was not the case (Rebuttal figure 3 below). This analysis has been added to the first Results section in the manuscript to acknowledge the important concern by the reviewer (first paragraph, Page 9):

“Presence of non-malignant cells in tumors may skew analyses of genomic data obtained from bulk tumor analyses (such as RNA sequencing and DNA methylation). In the SCAN-B cohort, there was no statistical difference in tumor cell content estimated by WGS between hypermethylated and BRCA1-null cases (T-test $p=0.47$), suggesting that non-malignant infiltration should not affect group-level conclusions.”

Rebuttal figure 3. Box plot of WGS estimated (by the ASCAT algorithm) tumor cell content for BRCA1-null and BRCA1 hypermethylated cases.

Consequently, *BRCA1*-null and *BRCA1* hypermethylated cases on a group level should be equally affected by non-malignant infiltration, with respect to bulk gene expression and DNA methylation. Additionally, analyses detailed in the manuscript that were performed with three different state-of-the-art sample deconvolution algorithms, using RNA-sequencing or DNA methylation data, also failed to detect any difference in cell type composition between the *BRCA1*-null and hypermethylated samples. As there is no detectable statistical difference in tumor cell content or the composition of the non-tumor compartment between the compared groups, nothing appears to be gained from adjusting the methylation or gene expression data with respect to these variables. In essence, for the *BRCA1*-promoter (and all other loci following the same dynamics) the methylation estimates for the unmethylated and hypermethylated tumors are affected by infiltration to the same extent. The effect of non-tumor cells on aggregate DNA methylation is, however, only detectable when the methylation state of the tumor and non-tumor compartment are diametrically opposed, whereupon the observed linear relationship emerges. New *BRCA1*-methylated cases could not appear as a result of adjusting for tumor cell fraction, because all conceivable correction algorithms can only boost a preexisting difference in DNA methylation. Our analyses of copy number alterations (CNAs) are not affected by different tumor cell content as the preprocessing by ASCAT⁹ corrects for it in the creation of allele specific copy numbers. Similarly, mutation calling has been optimized at the Sanger institute where the WGS was performed in this respect as well, and given that mutations are detected, mutational analyses (e.g. driver analyses, and signature analyses) should generally also not be affected. In addition, prior WGS quality filtering in our first study of the SCAN-B cohort removed cases deemed to have insufficient tumor cell content⁴.

The original figure 1D shows the clear separation of two distinct populations of samples with different percentages of *BRCA1* promoter hypermethylation. As mentioned above, it therefore appears unlikely that other samples in the cohort are hypermethylated given the excellent agreement between two independent analytical methods (pyrosequencing & Illumina Infinium DNA methylation, original figure 1A). Consequently, approaches for correcting DNA methylation data for tumor cell content would NOT change the pyrosequencing based group labels used in the actual downstream analyses of RNAseq, DNA methylation, WGS, and IHC data. This is a key point that we wish to make in response to the comment by Reviewer 1.

In response to the request to color the methylated samples on a graph of mRNA expression, we have analyzed and discussed the mRNA expression of *BRCA1* versus the pyrosequencing methylation % and mutation status (Rebuttal figure 1 above) and added it as a supplementary figure to the revised manuscript (Supplementary Figure S2).

Regarding correcting for tumor cell content in gene expression group analyses, we do not know of a bioinformatical method commonly used by the scientific community to correct global bulk RNAseq data for tumor cell content determined from methods applied to WGS or DNA methylation. Such a correction would be very complex, given that tumor cell content estimations are not always exact (in silico or in situ) and that there is a complex mix of different cell populations and non-malignant cell types present in a bulk tumor specimen. Methods such as CIBERSORT can be used to approximately deconvolve aggregate (known) cell type distributions from bulk RNA-seq data. Deconvolution-based cell type estimates cannot simply be applied to individual genes as a blanket correction factor as the measured expression of a given gene represents a function of both the cell type composition of the aggregate sample as well as the expression levels within the individual cell types. The only feasible correction strategy for adjusting gene expression levels based on the tumor cell content variable would

require the assumption of a uniform expression state within all cells of the non-tumor compartment for a given gene as well as a linear relation between gene expression levels and nuclear genome content (from which the tumor fraction is estimated). These assumptions would almost certainly be false for most genes with cell-type specific function, as a small number of cells could contribute the majority of the observed signal at the RNA level. We therefore conclude that adjusting the gene expression data based on tumor cell content imposes a risk for the introduction of bias into downstream analyses that is essentially unquantifiable and probably at least as great as the source of potential bias that one would be attempting to correct for.

Regarding the comment about correcting for tumor cell content in DNA methylation analyses, we performed the global DNA methylation analysis (i.e. differences between *BRCA1*-null and hypermethylated) using a rank-based test, which is insensitive to the magnitude of difference between two groups, and thus in our case identifies markers that follow the same pattern as the pyrosequencing-based hypermethylation vs “null” label. Considering that infiltration does not differ on a group level between hypermethylated and *BRCA1*-null SCAN-B cases (tumor cell content, Rebuttal figure 3 above), then correcting for tumor content would not change the end result substantially. Moreover, as we only test hypermethylated vs *BRCA1*-null SCAN-B samples, even if a few *BRCA1*-null samples could possibly be hypermethylated (which we believe is now clearly demonstrated not to be the case) this is likely to have limited influence in a group analysis. This applies particularly in the current setting in which there is not even an indication of a global difference in methylation between the respective groups. Simply put, if a substantial number of CpGs existed in the genome that follow the methylation status of the *BRCA1*-promoter and differ in *BRCA1*-null samples, these would be subject to the same amount of non-tumor signal as the hypermethylated samples and would therefore produce a statistical signal of the same magnitude as that observed in the *BRCA1*-locus. We see no evidence of this. In response to this reviewer’s comment, we have performed the same statistical test on M-transformed EPIC beta values to account for the heteroscedasticity in beta-values and observed no substantial difference in results (31/32 probes remain significant).

From a purely statistical standpoint, if there is an infiltration process that affects *BRCA1*-related global methylation in proportion to tumor cell content it should not be completely immune to a rank-based test. Indeed, this is supported by the fact that the only significant CpGs after correction for multiple testing are situated in the *BRCA1*-locus.

Regarding the methods used for correcting DNA methylation for tumor cell content, even though it is well known that the non-tumor compartment influences DNA methylation measurements, most developed tools have been aimed at quantifying tumor purity, not correcting for it. Among the published methods for deriving purity estimates from DNA methylation data are, e.g., Zheng *et al.*¹⁰, Aran *et al.*¹¹, Zheng *et al.*¹², and Benelli *et al.*¹³. However, none of these have reached widespread use within the broader research community, and to our knowledge those that have been developed into actual tools are only implemented on the older Infinium 450K array platform and not the EPIC platform used in the current study. Frequently, these methods also make use of normal reference samples when estimating tumor purity. With respect to the performance of these methods for estimating sample purity, sequencing-based estimates have generally constituted the gold standard measure of comparison. Several methods for deriving differentially methylated positions (DMPs) have been developed and incorporated in R-packages such as “minfi”¹⁴ and InfiniumPurify¹⁵. However, most of these methods do not account for tumor purity and those that do, such as InfiniumPurify, typically base correction on the assumption of a single homogenous methylation state within the tumor compartment. This assumption is violated by the *BRCA1*-

locus itself. An additional approach is to incorporate tumor purity as a covariate in a statistical model but this approach does not yield substantially different results, as the two compared populations (*BRCA1* null vs hypermethylated) do not differ with respect to purity (Rebuttal figure 3 above).

In summary, we believe our detailed response has demonstrated that our conclusions from the analyses performed on DNA, RNA, and DNA methylation in the original manuscript are valid regardless of whether an adjustment for tumor cell content could, hypothetically, have been performed.

→Are some CpG islands more affected by tumour content than others – could this be informative for testing BRCA methylation in the blood?

Response: For the pyrosequencing data, the Pearson correlation in methylation % between the two primer sets (summarized as methylation level per primer set) was 0.997 (n=229 measurements) and 0.988 for hypermethylated cases specifically, when analyzed in raw data. The Pearson correlation between the primer sets and WGS estimated tumor cell content (%) was 0.831 and 0.833, respectively. The conclusion is therefore that pyrosequencing primer sets do not seem to be differentially affected by tumor cell content. The pyrosequencing based *BRCA1* hypermethylation analysis of blood DNA was performed using the same primer sets as for the tumor DNA analyses. We have expanded the Methods section on the pyrosequencing analysis (under the subheading *BRCA1 promoter hypermethylation analysis*) in response to a comment from reviewer 2 and to provide more information on this approach. It is also worth noting that the WGS based estimation of tumor cell content is an in silico generated estimate based on computed allele specific copy number estimates, and so may not always represent the cellular truth.

As evident from original figure 1A, the *BRCA1* promoter associated CpGs (by Illumina annotation) measured using the EPIC platform vary in their concordance with pyrosequencing data and between themselves on a probe level. This is as expected, because the location of CpGs is important. We must also emphasize that while the pyrosequencing assay has been specifically designed to accurately quantify *BRCA1*-promoter methylation, the Illumina EPIC array is considered to be a general-purpose screening tool and, as such, more weight should be given to aggregate patterns seen across a larger number of samples and CpGs than to the observed measurement value for any individual probe (CpG) in a sample. With regards to the EPIC array, the probe design and positioning has been performed algorithmically and needs to balance the trade-off between being able to include a large number of probes compatible with a uniform hybridization protocol and maintaining an acceptable performance at the level of individual CpGs. As the EPIC platform is hybridization based, one can also expect a higher degree of variability in measurement accuracy compared with methods such as pyrosequencing. As such, it is to be expected that while individual CpG probes may exhibit slightly different dynamics with respect to baseline methylation level and measurement accuracy, local and global trends can still effectively be quantified using the platform. Considering the reviewer's question in the context of the EPIC DNA methylation data, we note that *BRCA1* promoter associated CpGs show higher intervariability than WGS estimated tumor cell content on the level of individual CpGs, but that aggregation of these measurements effectively converges on the same sample labels as those obtained by pyrosequencing (original figure 1A). This is consistent with the statistical expectation that a larger number of CpGs that

essentially measure the same process, i.e., *BRCA1*-inactivation, should better approximate reality (Rebuttal figure 4, below).

Rebuttal figure 4. Scatterplots of 9 Illumina EPIC CpGs associated with *BRCA1* based on Illumina annotations versus WGS estimated tumor cell content (%) by ASCAT algorithm. Red line corresponds to a 1:1 relationship.

With respect to the question regarding whether other CpG islands are more affected by tumor content, the simple answer is no. The number of methylated or non-methylated alleles contributed by a non-tumor cell to the aggregate methylation measurement will in essence be the same for each and every CpG position (and island) across the entire genome. The only thing that varies is the ability to detect the difference between an allele contributed by the tumor and an allele contributed by a non-tumor cell. The alleles contributed by normal cells can, in principle, only reliably be detected at CpG positions at which the tumor and normal cells have both uniform AND diametrically opposed methylation states. As such, the pyrosequencing results for the *BRCA1*-promoter more or less constitute the best achievable real-life performance for a group of CpGs that are completely and specifically methylated in the tumor compartment and for which the methylation state in the non-tumor compartment is largely invariable and diametrically opposed. With respect to additional CpGs irrespective of localization in islands, we find no evidence of additional positions in proximity of *BRCA1*, the inclusion of which could expand the number of hypermethylated cases in a meaningful way. As stated above, it is possible to find sporadic Illumina probes (CpGs) with seemingly elevated methylation levels in a limited number of “unmethylated” cases. However, unlike the cases that receive “hypermethylated” calls on both the pyrosequencing and EPIC arrays, this signal is not amplified by the inclusion of methylation estimates from additional nearby probes, arguing more in favor of experimental noise than biological signal. Therefore, while it is conceivable that this type of variation in the methylation of, e.g., a single CpG, could represent actual biological signal, there is no way of distinguishing this from technical noise. This means that the detected number of hypermethylated cases should be seen as a robust and orthogonally

verified estimate of the proportion of tumors in which *BRCA1*-inactivation occurs through promoter DNA hypermethylation.

→On line 303 – first time mentioned BRCA2 cases? Why is this case shown in a paper discussing BRCA1 methylation versus BRCA mutation, this needs further explanation or remove it?

Response: BRCA2 is mentioned in the first Results section because of the PD35990a case (original Figure 1E). This case carries a pathogenic germline *BRCA2* variant + LOH, which would intuitively make one think that it has the genetic characteristics of *BRCA2*-deficient disease. But this is not the case. Instead, the tumor carries the full features of *BRCA1*-deficient disease, (presumably) due to the concurrent existence of *BRCA1* hypermethylation. Although this is just one case that is likely of limited clinical value, we believe it would be dishonest not to highlight the existence of it in our series – which we believe most readers would not have expected.

→On line 312: 46/237 underwent BRCA-screening – 34 were wt BRCA cases (8 were germline BRCA cases), correct? Please report explicitly.

Response: 46 SCAN-B cases underwent clinical screening, with nine germline *BRCA1* findings and three germline *BRCA2* findings. In the revised manuscript we have clarified by including a new sentence in the beginning of the second Results section (page 10):

“Nine cases showed germline inactivating BRCA1 variants and three had germline inactivating BRCA2 variants.”

→16/34 cases were methylated in the blood – again please state explicitly why not all 46 screened patients were also part of blood methylation test. For the reader it would be easier, to always clearly mention, if the author means blood or tumour-methylations.

Response: Blood methylation analyses were performed in two rounds. First, the 34 SCAN-B cases without clinical germline *BRCA1/2* alterations were tested as stated in the text based on the finding that there was a large age discrepancy between cases with somatic tumor methylation and those without. This is described by (page 10):

“Among the clinically screened patients without a pathogenic germline BRCA1/BRCA2 variant, 16 out of 34 (47.1%) displayed tumor BRCA1 hypermethylation while 18 did not.”

And (page 10):

“SCAN-B patients with BRCA1 hypermethylated tumors (n=16) also showed higher levels of BRCA1 promoter methylation in matched peripheral blood DNA than the 18 patients with unmethylated tumors and no BRCA1/2 variants (hypermethylated: n=16, mean CpG allele frequency=4.03, median=3.5, sd=1.53; unmethylated: n=18, mean=3.03, median=3.0, sd=0.61; T-test p=0.024, Wilcoxon’s test p=0.015, Figure 3B).”

In a second run, we confirmed findings in additional cases from the SCAN-B cohort.

Please note that we do not call cases with increased levels of *BRCA1* hypermethylation in peripheral blood “methylated”, as we believe that we do not have the absolute analytical sensitivity to do so at the moment (a shortcoming related to many current analytical methods). In the revised manuscript we have made this important limitation clearer Discussion (Page 18):

“Although these results must be interpreted with caution as the hypermethylation levels are at the limit of detection by pyrosequencing (and thus not suitable for prediction of somatic hypermethylation), they are intriguing in the context of both possible circulating tumor DNA and findings of mosaic constitutional BRCA1 hypermethylation.”

→Do all 16 blood-methylated patients have tumours with BRCA1-methylation?

Response: Yes, please see response above.

→Do any of the 18 hypomethylated cases have tumour BRCA1-methylation?

Response: No, please see above.

→All line plot of beta-values should be provided, illustrating patient-matched methylation status

Response: Please see response to remark below for a figure (Rebuttal Figure 6) that we believe is asked for.

→Figure 2B Wilcoxon test is not informative here

Response: In the original figure 2B we only display the T-test result. In the text, however, we present both the T-test (parametric) and Wilcoxon’s test (non-parametric) results. We present both due to the challenge of the data that was acquired close to the limit of detection of the pyrosequencing system. We would have been concerned if we had reached significance in only one of the two tests and believe it is thus appropriate to supply both results.

→Then additional 39 /70 cases showed BRCA blood-methylation – here the authors should again show the relationship btw blood and tumours BRCA methylation. Are there cases in which BRCA is not methylated in tumours but it is in the blood – what is the BRCA expression levels in the tumour?

Response: Firstly, we were able to include one more sample in the analysis that had previously been left out as a control, raising the second patient cohort to 71 SCAN-B patients and 105 in total. We report the observation of increased *BRCA1* promoter methylation levels in blood DNA extracted through standard protocols at a regional biobank facility. The original figures 2B and 2C clearly demonstrate that the assay at hand cannot be used for classification as the absolute analytical sensitivity is lacking (as we discussed previously and as pointed out by other reviewers). This sensitivity issue may hopefully be resolved in the near future by sensitive sequencing methods (which we discuss at the end of the manuscript). To note, the same blood

DNA was also used for the matched tumor-normal WGS analyses. To ascertain that the observed pattern of increased *BRCA1* promoter methylation levels in blood DNA from patients with somatic *BRCA1* hypermethylation in tumors can be reproduced by an orthogonal method, we tested 20 of our cases (10 patients with somatic tumor hypermethylation and 10 without) using an NGS-based assay under development by a Norwegian research group (Nikolaienko O, Lønning PE and Knappskog S. Unpublished method) (Rebuttal figure 5, below).

Rebuttal figure 5. (A) *BRCA1* promoter methylation levels in 10 patients with tumor hypermethylation versus 10 without, using a developmental NGS-based assay. Blood DNA was from the same aliquot as the pyrosequencing-based analysis. The NGS-assay targets 17 positions in the promoter region of *BRCA1*, and the average of these are displayed in the boxplots for each sample. (B) Corresponding pyrosequencing estimates for the same samples.

Finally, in response to the request by the reviewer in this and a previous remark we have generated the plots shown in Rebuttal figure 6, below. However, we are not confident what these plots would add to the revised manuscript and have thus not included them but are open to revision at the discretion of the editor.

Rebuttal figure 6. (A) Pyrosequencing tumor *BRCA1* methylation levels vs blood methylation levels in 105 dual analyzed cases. (B) Tumor mRNA *BRCA1* levels (FPKM) vs blood pyrosequencing methylation levels in 105 dual analyzed cases.

→Given that the very complex relationship btw methylation in tumour and blood, mutation germline and somatic, and expression levels, subanalyses of cases with unambiguous expression – methylation, expression-germline status should be performed, see whether this further supports similarities or identifies differences between mutated and methylated BRCA cases.

Response: As discussed in answer to a previous comment, correcting the tumor DNA methylation levels obtained by pyrosequencing would not change sample group labels in downstream analyses. We do note that in the literature and in our own data that mRNA expression of *BRCA1* is possible in cases with *BRCA1* inactivating variants (Rebuttal figure 2, above). However, the number of outliers that would fall in the categories suggested by the reviewer appears very small, precluding meaningful subanalyses. Furthermore, as the number of samples subject to any individual type of “anomaly” noted by the reviewer is very small, conclusions on a group level would be expected to hold true. We would also like to emphasize that the kind of stratified analyses requested by the reviewer would mostly produce “N of one” or “N of few” subgroups, which would neither be robust nor effectively evaluable from a biological and clinical perspective. Rebuttal figure 2 illustrates that the level of substratification for any given category of adjustment requested by the reviewer would result in very small subgroups that take, e.g., mutation type AND expression status, into account. Controlling for possible sources of heterogeneity or confounding is of course important, but one must also take into account that substantial bias can be introduced into analyses by overstratification as well.

We would also like to point out that with respect to a small number of seemingly “outlier” samples with respect to *BRCA1* gene expression, it is not possible to unambiguously assign the observed FPKM estimates to the tumor compartment. The observed signal could also be derived from the non-tumor compartment, as the number of mRNA copies in a cell can theoretically vary by orders of magnitude between different cells and cell types. We question the recurring emphasis of the reviewer on a “complex relationship between methylation in the tumor and blood”. The relationship in the extracted tumor sample can be near-perfectly described by a linear model using the sequencing derived tumor fraction as the independent variable. A linear relationship, such as the one shown in original figure 1D, is fully explained by all tumor cells having 100% methylation of the *BRCA1* promoter while all non-tumor cells have 0% methylation of the same CpGs. Given this relationship, there will be a direct linear dependence between the measured DNA methylation level and the fraction of DNA molecules that are contributed by tumor cells to the total sample. It is important to note that this differs from the case for mRNA estimates. For the promising observations in patients’ blood, the increased signal in samples showing *BRCA1* methylation could be the result of a small amount of circulating tumor DNA (which we know is hypermethylated) or a readout of rare somatic cells with *BRCA1* hypermethylation. Either way, the observed methylation levels in blood are consistent with the presence of a small number of hypermethylated alleles being present in the blood together with a large abundance of unmethylated alleles, the overwhelming majority of which are attributable to normal nucleated cells of the blood.

→A clear cohort definition needs to be made at the end of this section so that the reader gets a clear picture of what will be compared from now onwards.

Response: We believe the new Figure 1 addresses this remark. It clearly shows the different performed analyses, the cohort sizes used, and the composition of hypermethylated, germline, and somatic cases in respective analysis. We thank the reviewer for pointing out the need to clarify this.

→Figure 2E needs to include barplots for BRCA null cases and no-BRCA cases, to reflect Table 2.

Response: We thank the reviewer for pointing out the need to add these data. In the revised manuscript we have added a new Supplementary Figure S3 showing these proportions as pie-charts. We cannot unfortunately fit all display items into the original Figure 2. The following text have been added to the third subsection of the Results (page 11):

“The subtype proportions for BRCA1-null and non-BRCA1 altered cases are shown in Supplementary Figure S3, demonstrating similarity between hypermethylated and BRCA1-null cases.”

→Age seems to be an interesting factor with regards to BRCA1 methylation and comes up in several parts of this manuscript. It would be good to combine it and discuss it once. Also BRCA2 needs to be specially mentioned if it is shown such as in Figure 2D or taken out.

Response: As mentioned in response to a previous comment, we described BRCA2 very briefly in the Results section due to the existence of one case with both *BRCA1* hypermethylation and a *BRCA2* inactivating variant + *BRCA2* LOH. We therefore do not consider it necessary to mention this single case in a discussion about age at diagnosis and hypermethylation.

We agree with the reviewer that age at diagnosis is an interesting factor for the *BRCA1* hypermethylated cases. Patient age is intentionally discussed in different parts of the Discussion. E.g., patient age is used as the introduction to the discussion about blood methylation levels, as those analyses originated from the observation of different age spectra in SCAN-B patients with or without somatic hypermethylation who underwent clinical germline screening. The second occurrence of age in the Discussion is in the context of hypermethylation in subtypes, following the germline screening / blood methylation section. Here, only a short note is made on that we observe hypermethylation in non-traditional BRCA groups (including older patients). Finally, in the Discussion section related to patient outcome we make a short speculative note on age and treatment decisions. Here, based on the observation of hypermethylation also in older patients, coupled to the HRD exclusiveness and better prognosis of hypermethylated cases in general, we argue that older TNBC patients should not be default be excluded from adjuvant therapy given appropriate genomic profiling. Given the current structure of the Discussion, it would require an extensive rewrite to accommodate this specific request and, as no other reviewer has requested it, we have opted not to do so.

→More detailed information to Figure 2E & F needs to be provided, as there is lots of data shown and not discussed. How many BRCA-tumour methylated and BRCA carriers are included in 149 survival study?

Response: We believe the reviewer is referring to the original figures 2F and 2G. The univariate Cox regression analyses in the original figure 2F show hazard ratios for patients treated with adjuvant chemotherapy stratified by *BRCA1* hypermethylation or standard prognostic clinicopathological variables in breast cancer (and molecular subtypes). We provide the latter as reference for the readership to be able to compare hazard ratios for hypermethylation to standard clinical variables. Figure 2G shows the number of hypermethylated and *BRCA1*-null cases included for IDFS analysis. Acknowledging the reviewer remark, the new figure 1 and revised CONSORT diagram (Supplementary Figure S1B) now also includes details of the sample groups used for the IDFS and DRFI endpoints. We have also updated the original Figure 2F with group size data when not all 149 cases were included due to missing data, removed the molecular subtypes (as they were not generated by a clinical assay like ProSigna), and updated the Results text with (Page 12):

“In both univariable Cox regression and Kaplan-Meier analyses, BRCA1 hypermethylation alone or combined with BRCA1-null cases was associated with significantly longer IDFS than non-altered TNBC cases (Figure 3F, which also shows univariate results for standard prognostic variables in breast cancer for reference and Figure 3G). The five-year IDFS was 88% for BRCA1 hypermethylated patients versus 71% for non-methylated TNBC patients, while the five-year DRFI was 92% versus 80%, respectively.”

→Looking at Sup Fig2 – it is not clear what is BRCA altered – is this BRCA null e.g germline and somatic? BRCA2 should be excluded in this analyses – numbers of cases should be provided in Figure 2F forest plot

Response: In the revised manuscript, we have removed the original Supplementary Figure S2 in response to a remark by reviewer 2, as the main text now includes the important DRFI results. Detailed sample data of the 149 cases included in the univariate IDFS Cox regression is available in Staaf et al. 2019⁴, as described in the Methods section and also for the hypermethylated and *BRCA1*-null cases in the new Supplementary Table S1.

Moreover, the original Figure 2F and corresponding legend have been updated with an indication of whether the proportional hazard assumption is not fulfilled, and we have added that for nodal status there was one missing value (148 instead of 149 cases analyzed).

Figure legend:

“Univariate Cox regression using invasive disease-free survival (IDFS) as clinical endpoint for different variables in 149 SCAN-B patients eligible for outcome analysis after standard of care adjuvant chemotherapy. HR: hazard ratio. CI: confidence interval. A Zph p-value <0.05 corresponds to that the proportional assumption is not fulfilled. An (n=) indication in the right axis indicates that not all 149 cases were used due to missing values.”

→Figure 2 G Kaplan Meier curves cross and violate KM analyses, thus need to apply different survival analyses e.g restricted mean.

Response: This remark concerns the Kaplan-Meier center panel in original figure 2G that shows the 3-group comparison of hypermethylated, *BRCA1*-null, and non-*BRCA1* altered cases. In this panel, two survival curves cross (the *BRCA1*-null and non-altered groups). In the

panel we report the log-rank p-value. We do not report a hazard ratio by Cox regression, for which the prerequisite is a fulfilled proportional hazard assumption. Crossing survival curves in a Kaplan-Meier plot is a typical indication that the proportional hazard assumption may not be fulfilled. However, for the actual comparison, a univariate Cox regression model with a variable containing the 3 groups, a proportional hazard test is actually fulfilled (R `cox.zph()` function test, $p=0.47$).

The log-rank test does not have a specific counter hypothesis and can be used in the situation shown in the panel, i.e., the proportional hazard assumption does not need to be fulfilled. For Cox regression analyses presented in the study we have checked the proportional assumption through the R `cox.zph` test. The reviewer is correct that a restricted mean analysis can be performed in the aforementioned situation. However, as the main take-home message in the main text is related to the top and bottom panel in which there is no crossover and proportional hazards are also fulfilled (`cox.zph` $p>0.05$), we have opted not to add the suggested analysis.

For transparency, we have calculated the restricted mean survival time (RMST) for hypermethylated vs non-methylated (similar to the top panel in the original figure 2G), resulting in a p-value of 0.015 (using the R package `survRM2`). In addition, for the three-group analysis (center panel in the original figure 2G), using *BRCA1*-null status as a covariate, we obtained a p-value of 0.07. Both RMST p-values are very similar to the original log-rank p-values and so the conclusions we draw are still valid.

→Copy number analyses – can the authors be sure that none of 52 BRCA1-null cases shows BRCA-methylation? Fisher tests need to be applied to see if there are any differences in losses or gains, and a sup table should be provided.

Response: The 52 *BRCA1*-null cases included in the analysis presented in the original figure 3A includes *BRCA1*-null cases from the SCAN-B cohort, and *BRCA1*-null cases from Nik-Zainal *et al.* ⁵. All have germline/somatic *BRCA1* inactivating variants. Based on both the pyrosequencing data and the *BRCA1* DNA methylation data presented in the original figure 1A we could not detect a single case with, e.g., biallelic alteration and hypermethylation of *BRCA1* in the SCAN-B cohort. While extensively analyzed, not all cases in the Nik-Zainal *et al.* cohort were profiled by Illumina 450K DNA methylation arrays. As such, although it cannot be ruled out that cases in this cohort were hypermethylated, it seems highly unlikely to be the case and it would not affect group level conclusions.

It is unclear from the remark what type of supplementary table the reviewer is requesting. In the revised manuscript we now provide the summarized frequency estimates for the original figure 3A in an online repository; as the corresponding text data size is close to 42Mb (data from >900 000 markers) online availability is the only option. Moreover, all source copy number data used to generate calls of gain and loss is publicly available from our first study of the SCAN-B cohort in Staaf *et al. Nat Med* 2019 ⁴. Furthermore, we created a new Supplementary Figure S4 that details the difference in gain and loss between hypermethylated and *BRCA1*-null cases on a genome wide level, showing that across the genome only a few stretches appear to differ by >10% in frequency. Moreover, we performed Fisher tests for each marker (>900 000 markers) for gain and loss separately, as suggested by the reviewer. After multiple testing adjustment by FDR, zero ($n=0$) markers remained significant ($p<0.05$). This non-significance is a combination of sample sizes, small proportional differences between

groups (i.e. biological signal), and massive multiple testing that we need to correct for. Based on these results, it seems clear that there are no characteristic group defining regions of copy number gain or loss. In addition, the histograms of the calculated Fisher p-values for gain and loss demonstrate that there is no apparent signal (i.e., a large proportion of markers with small p-values) between the groups. These analyses are added to the new Supplementary Figure S4 and cited in the text in the fifth subsection of the Results in response to the reviewers' comment (page 13).

“A full copy number analysis is available in Supplementary Figure S4, demonstrating no statistical differences between the groups (FDR adjusted Fisher’s exact test $p > 0.05$).”

→The authors should revise the TNT trial analyses, as BRCA methylated cases showed a better response in docetaxel (Tutt et al., Nat Med Figure 2D), and need to revise their statement. Careful assessment of clinical trial data should be performed and discussion should be extended.

Response: The TNT trial is a study performed in metastatic disease. Acknowledging the remark by the reviewer we have rephrased and expanded the discussion in the revised manuscript (page 20):

“In advanced disease, BRCA1 hypermethylation has been reported not to be associated with a better treatment response to carboplatin compared with docetaxel ¹⁶ (when hypermethylation was assessed in primary tumor tissue) or to single-agent carboplatin effect ¹⁷. This has led to a hypothesis of a “soft/plastic” BRCAness phenotype for hypermethylated patients compared with germline BRCA1-altered cases ¹⁶, which remains to be proven by analysis of matched tissue from primary and relapsed tumors.”

In our cohort, the 5-year DRFI for hypermethylated patients given standard of care chemotherapy was 92%. This implies that there were only three distant relapses meaning that robust analyses of the differential characteristics of relapse versus non-relapse cases is not possible.

→Figure 4A is a redundant display to the supervised analyses. The individual 32 CpGs should be shown for each case, so the reader can see whether there are changes across them.

As mentioned above, the expression of BRCA1 needs to be shown to see if it confounds the analyses or not.

Response: In the revised manuscript we have now created a heatmap for all 32 CpGs, marked them as to whether they are in the canonical promoter region or not, and also added a bar for *BRCA1* mRNA FPKM expression levels, as requested. The latter demonstrates that *BRCA1* mRNA expression does not confound the analyses. Clustering was performed as in the original figure 1A for consistency. The methylation data relating to this figure (for all 235 cases) is now available in the Source Data file requested by the journal.

→Can the authors add a table to illustrate correlation btw gene expression and methylation?

We are not sure what the reviewer is suggesting in this request. In the new Supplementary Table S1 we have included tumor BRCA1 tumor DNA methylation level by pyrosequencing and BRCA1 tumor mRNA expression in FPKM for each sample. Individual CpG methylation data for *BRCA1* CpGs are available in the Source Data file for corresponding figures. We hope that this answer addresses the request.

→Figure 4B – can the authors provide a heatmap and gene lists what is different btw these 7 groups?

Response: The aim of the original figure 4B was to illustrate the fact that hypermethylated and *BRCA1*-null cases do not form distinct transcriptional clusters based on unsupervised clustering of >7000 genes. The figure serves as a snapshot illustration of the aggregated consensus clustering and principal component analyses presented in original figures 4C-D. In the revised figure we have redone the clustering using ward.D linkage and Pearson correlation for coherency and added a corresponding heatmap as requested. Please note that due to the number of clustered genes we cannot provide gene level or gene cluster detail in the heatmap. In the Source Data file, we provide the gene identifiers used in the hierarchical clustering, consensus clustering, and PCA analysis. In the new Supplementary Table S1 we provide sample mapping identifiers to the deposited gene expression data for the samples.

We understand the reviewer's second request as asking for gene lists that are characteristic for the different clusters in the original figure 4B. These have to be generated through supervised differential gene expression analysis. As the reviewer knows, hierarchical clustering, as portrayed in the original figure 4B, can be performed in a variety of different ways and using different gene sets. Each perturbation can yield slightly different hierarchical trees or clusters. The aim of the clustering was not to "claim" or illustrate the existence of specific transcriptional subgroups in the analyzed patient subset (*BRCA1*-null and hypermethylated patients with TNBC). Such an analysis requires validation cohorts, more extensive bootstrap/consensus clustering, or even integrative clustering of different omics layers. Moreover, such analyses should also be performed in the context of TNBC as a whole entity. We therefore believe that the requested gene lists are not relevant, as they would only represent a snapshot view with questionable robustness in independent cohorts. Based on the data made publicly available from this study, anyone should be able to do the specific analysis.

→The gene list of differential gene expression should be provided as a sup table What are the 8 differentially expressed genes, and what is their methylation or copy number status?

Response: In the revised manuscript, subsection 7 of the Results, we have named the eight differentially expressed genes (page 16):

"Merely eight genes, BRCA1, UQCRHL, SOX6, HAPLN1, POLR2J3, H2AC20, MUCL1 and HYI, were differentially expressed at a false discovery rate of 1% (Supplementary Table S2)."

An important conclusion from the supervised analysis is that there does not seem to exist strong transcriptional differences between *BRCA1*-null and hypermethylated cases, in line with the unsupervised analyses presented in the original figure 4B-D. In the revised manuscript, we have reinforced this conclusion at the end of subsection 7 of the Results (page 16):

“Taken together, the low number of differentially expressed genes identified and the unsupervised analyses suggest that there is no strong transcriptional difference between the hypermethylated and BRCA1-null patient subgroups.”

In response to the reviewer’s request we have also created a new Supplementary Table S2 (excel file) containing the following information for the eight differentially expressed genes:

- Gene expression measurements for all analyzed cases + SAM analysis characteristics. For the SAM analysis, note that the precision in the delta calculation in the R siggenes package means that a q-value rounded to 0.01 is counted as significant.
- Copy number status for SCAN-B cases. No gene reached statistical significance when testing for differences in observed copy number status (LOH, amplification, normal, duplication, other) between sample classes (Chi-square test $p > 0.05$).
- DNA methylation data. Note, for two genes there were no CpG associated data according to the original Illumina annotations (UCSC_RefGene_Name). In the table we provide all CpGs matching to the genes.

→It is not clear in the CONSORT diagram where one finds the 82 SCAN-B cases for immune feature gene expression analyses?

Response: As stated in the Methods section and in the original manuscript, the analyses performed on the 82 SCAN-B cases relates to the 57 *BRCA1* hypermethylated and 25 *BRCA1* null cases in the SCAN-B data set reported in Staaf et al. *Nat Med* 2019⁴. This is also made clear in the CONSORT diagram and the new Figure 1.

→Figure 4E – again does the trend shift to significance if cases with BRCA hypermethylation but BRCA expression are removed? Please show in the graph and discuss in the manuscript.

Response: Firstly, based on extended IHC analyses in the 237-sample SCAN-B cohort one additional hypermethylated tumor (n=1) could be evaluated for PD-L1 expression, raising the total of analyzed cases to 53 of the 57 hypermethylated cases. Case numbers have been updated in the revised manuscript and figures (new Figure 6A).

Secondly, please see the discussion in relation to a previous comment by the reviewer about the important fact that group labels (hypermethylation status) would not change based on, e.g., tumor cell content compensation, and illustrations of *BRCA1* mRNA expression showing that the number of potential outliers appears to be very low. For instance, when the hypermethylated case with the highest *BRCA1* mRNA expression (PD31106a, see Rebuttal figure 7 below for WGS characteristics, clearly identifying this case as having a BRCA1-like appearance^{5, 6, 7}) is removed, the Fisher exact p-value changes from $p=0.21$ for all cases to $p=0.3038$. If we continue by removing the top two *BRCA1* expressing hypermethylated cases

($p=0.305$), then the top three ($p= 0.4314$), top four ($p= 0.3057$), and top five *BRCA1* expressing cases ($p= 0.428$) results remain non-significant. Removal of the top 5 expressing samples in the hypermethylated group corresponds to approximately 10% of the “outliers” with the highest *BRCA1* expression. Based on these data, we conclude that the data presented in the original figure 4E is not affected by hypermethylated cases with “outlier” mRNA expression of *BRCA1*. Consequently, we have not added the proposed analysis to the revised manuscript. Again, all data will be available in Supplementary Table S1 for reproducibility.

Rebuttal figure 7. WGS characteristics of sample PD31106a which has the highest *BRCA1* mRNA expression (FPKM=5.973104) of all hypermethylated cases.

→TILs, immune gene expression and PDL1 expression could be used in more complex integrative analyses. Which gene expression goes with PDL1 – does this help to decipher small differences which are missed when these rather global analyses measures are applied?

Response: We fully agree with the reviewer that more complex integrative analyses are of interest in this area, particularly considering the emergence of immune checkpoint inhibitors as a realistic treatment option in TNBC. Integrative analyses using only gene expression would likely imply correlative analyses, as the bulk gene expression would be heavily affected by the unknown cellular composition of each tumor (as discussed previously).

In line with the presented IHC, TILs, and in silico immune cell deconvolution methods, the supervised gene expression analysis did not identify immune-related genes to be differentially expressed between the two sample groups. To substantiate this, and in response to the reviewer request, we analyzed 102 immune associated genes by supervised and

unsupervised analysis, adding the results as the new Supplementary Figure S5, and the following text to the eighth subsection of the Results (page 16):

“Supporting these observations, the expression of 102 immune cell type associated genes was analyzed by unsupervised clustering and supervised differential gene expression in the 82 SCAN-B cases. While groups of tumors displayed distinct immune infiltration (e.g. consistent with the immune modulatory TNBCtype subtype) these were not defined by BRCA1-status or PD-L1 IHC status, nor was any gene statistically different in expression between sample groups (FDR adjusted Wilcoxon’s test $p > 0.05$) (Supplementary Figure S5).”

We believe this snapshot analysis combined with the cell type deconvolution analyses highlight that additional data layers beyond gene expression are needed to figure out, e.g., the differences in PD-L1 SP142 scores, while at the same time being supportive of the main observations regarding the lack of clear immune landscape differences between *BRCA1*-null and hypermethylated cases.

Consequently, complex integrated analyses should include, e.g., HLA-corrected neoantigen predictions, extended IHC analyses of immune cell markers, and proteogenomic data, which are beyond what is currently available for the total cohort. However, in response to the reviewer’s request, we have added neoantigen analysis based on somatic substitutions from our WGS data at the end of the PD-L1/TILs result section, including them also in the Source Data file (page 17):

“To investigate whether PD-L1 differences between BRCA1-null and hypermethylated cases were related to neoantigen burden, we calculated number of expressed neoantigens from somatic substitutions by integrating WGS-based HLA-typing, neoantigen prediction, and mRNA expression for 232 of 237 SCAN-B cases^{18, 19}. The Pearson correlation between the total substitution burden and predicted number of expressed neoantigens from substitutions was 0.91. PD-L1 positive cases ($n=120$) had a higher number of expressed neoantigens than PD-L1 negative cases ($n=93$) (Wilcoxon’s test $p=0.02$). In BRCA1-null and hypermethylated cases there was however no statistical difference ($p > 0.05$) between the groups in total, or when stratified by PD-L1 status, irrespective of whether all predicted neoantigens were assessed (Figures 6C-D) or weak or strong binders were assessed separately¹⁸.”

While we are generating additional, more in depth neoantigen predictions and extended immune cell marker IHC data on all SCAN-B cases in ongoing projects, the intention of the analyses reported in the current study is to highlight the patterns of what we believe are two important clinical markers in TNBC relating to prognosis (TILs) and treatment prediction (PD-L1, and here, specifically, the SP142 antibody marker, which is the companion diagnostic marker to determine the use of atezolizumab in metastatic TNBC). Our results show that *BRCA1*-null and hypermethylated cases appear to have similar general characteristics concerning these important markers. One can speculate that this could be an argument for considering them as a single entity with respect to these variables. In response to this remark, and a remark by reviewer 3 we have revised the discussion around TILs and PD-L1 in the Discussion section (page 21):

“TNBC tumors have been associated with high TIL-levels (signaling immunogenicity) which are also prognostically favorable^{20, 21}. A recent study has suggested a higher frequency of TIL-positive tumors in BRCA1/2-mutated patients than in wild type cases²². In the SCAN-B cohort, TIL-levels did not differ between hypermethylated and BRCA1-null cases, which is consistent with the similar prognosis in these groups after adjuvant chemotherapy. PD-L1 expression is also frequent in TNBC, including BRCA1 hypermethylated tumors²³, but studies

have not found that levels of PD-L1 + TILs in TNBC cancers are driven by a high mutation rate or by BRCA1 mutation status^{20, 22}. This lack of association may now be appreciated through our delineation of the BRCA1/2-wt subgroup by BRCA1 hypermethylation, forming a major subgroup of TNBC with similar genetic and immune cell phenotypes as BRCA1-null cases. Taken together, on a group level these results suggest that early-stage hypermethylated cases may have similar response to immune checkpoint inhibitors as the BRCA1-null group.”

→A table for each case which TILs score, PDL1 score, and CIBERSORT population proportion should be provided as sup table.

Response: This has been included in the new Supplementary Table S1 for BRCA1-null and hypermethylated cases. Thank you for pointing this out.

→References are in a different font

Response: Addressed, thank you.

→Reference 23 is not complete

Response: Addressed, thank you.

→Table 2 E Subtypes without any methylated cases are not listed. Not clear what this should mean?

Response: It meant that subtypes without any hypermethylated cases included were not listed in the table. In the revised manuscript we have added all subtypes with at least one classified case in the 237 cohort for completeness. We thank the reviewer for pointing this out.

Reviewer #2 (Remarks to the Author): Expert in methylation

In their manuscript, Dominique Glodzik and co-workers explored and compared the clinical, genetic, transcriptomic and epigenomic profiles of BRCA1-null and BRCA1 promoter hypermethylated triple negative breast cancer samples, leading to the basic conclusion that both subgroups essentially present the same phenotype. This has important clinical implications, particularly regarding treatment management. Overall, the paper is clear and well-written, and I have no major remarks that are likely to affect the general conclusion. Nevertheless, I have several comments that should be addressed and/or require additional clarification.

Response: We thank the reviewer for the very detailed and thoughtful review which has greatly improved the quality of the manuscript.

→The exact definition of BRCA1 promoter hypermethylation is crucial regarding interpretation of the results and future practical use as biomarker, see e.g. Koch et al. Nature Rev Clin Oncol 2018. Therefore, it is essential to put this information in the main text rather than supplementary information and to mention the primer sets used as well as the exact regions targeted (and the CpGs included in the analyses).

Response: We absolutely agree with the reviewer about the exact definition of analyzed CpGs. We do provide the reference to a study that in turn referenced an original study from our department that defines the assay²⁴, but we agree that this referencing was suboptimal. In the *Methods* section “*BRCA1 promoter hypermethylation analysis*” we now detail the pyrosequencing analysis with the expanded text below and citations of relevant papers:

“BRCA1 promoter hypermethylation status for tumors based on pyrosequencing was available from a prior study⁴ based on a protocol described by Saal et al.²⁴. Briefly, the pyrosequencing assay involved analysis of two CpG-dense amplicons containing five and four CpGs, respectively, with internal controls for bisulfite treatment (nucleotide positions are relative to the +1 adenine of the ATG translational start codon on the annotated Ensembl v36 BRCA1 genomic sequence): BRCA1 region 1, -1291 to -1267 (containing five CpGs); and region 2, -1332 to -1310 (four CpGs). The following primers were used: BRCA1_Region1/2_PCR_F, 5'-NNTATTTTGAGAGGTTGTTGTT TAG-3'; BRCA1_Region1/2_PCR_R, 5'-biotin-TAA AAAACCCCAACCTATCC-3'; BRCA1_Region1_Seq, 5'-GAATTATAGATAAATTAATA-3'; and BRCA1_Region2_Seq, 5'-GGTAGTTTTTTGGTTT T-3'. CpGs covered by the assay include cg16630982, cg15419295, cg16963062, cg20187250, cg04658354, cg04110421, and cg21253966.”

→Moreover, the authors used an arbitrary 7% methylation cut-off rather than 10% “due to cleanness of the data”, but this should be further substantiated. I fully agree that any cut-off used will be arbitrary, yet fig 1A suggests that the 10% cut-off would be more suitable (trough is between 0.1 and 0.2), raising the question why the 7% cut-off was selected.

Response: We believe this remark is a misunderstanding of what our original figure 1A is showing. Original figure 1A shows beta-values (heatmap) for 30 CpGs present on the Illumina EPIC DNA methylation array associated with *BRCA1* (transcription start site (TSS): -1500b to +500bp). This is stated in text and figure legends. It does not show pyrosequencing methylation rates. Reviewer 1 also commented on figure 1A, and we acknowledge the need to further clarify data presented in this panel. Consequently, we have updated the figure (e.g. adding a reference line for the 7% methylation cut-off), the figure legend, and text in the first Result section referencing the panel:

Figure legend:

(A) “Hierarchical clustering (ward.D2 linkage, Euclidean distance) of DNA methylation data (beta-values shown as a heatmap) for 30 CpGs associated with the BRCA1 gene (transcription start site (TSS): -1500b to +500bp) in Illumina MethylationEPIC data for 235 SCAN-B TNBCs, including 57 tumors classified as hypermethylated by pyrosequencing (black column sample annotation bars). Grey CpG annotation bars (rows) indicate a promoter associated CpGs according to Illumina EPIC annotations.”

Result text (page 8):

“Pyrosequencing classifications were corroborated by Illumina DNA methylation profiling data for BRCA1 gene associated CpGs (Figure 2A), and markedly reduced BRCA1 mRNA expression from RNA sequencing for hypermethylated cases (Figure 2B), similar to previous reports¹, that were in line with expression levels for cases with BRCA1 frame shift, nonsense and indel variants (Supplementary Figure S2B).”

We believe the panel that is most relevant to address the reviewer’s remark about the cut-off point is the original figure 1D. This figure shows the allele methylation rates identified by pyrosequencing versus the estimated tumor cell content by WGS for the actual cases included in this study. We believe that this illustrates the robustness of the analytical method, by showing the very low background values of samples apparently not methylated in tumor tissue. In the revised manuscript we have updated the original figure 1D with tumor cell content estimates from the Battenberg algorithm, which is WGS-based and can account for tumor subclones when estimating tumor copy number (which ASCAT that was used originally cannot account for). Here, the linear relationship is even more striking for the hypermethylated cases with $r^2=0.84$ (meaning 84% explanation) and slope=0.9 (close to 1) based on linear regression. Briefly, for the pyrosequencing results on which methylation state labels are based, the highest observed methylation in “unmethylated” samples was 3.595% and the lowest among the “hypermethylated” was 13.57%. Thus, any cut-off point between these two values produces the same end classification result. For the unmethylated samples, the mean was 2.26% and the SD 0.48%. The chosen cut-off point is arbitrary, as this reviewer mentions, but it is between the two extreme values of the data, so it is reasonable. In the revised manuscript we have added this detailed information to the Material and Methods section describing the pyrosequencing analysis:

“A cut-off of 7% in allele methylation was used to call samples as hypermethylated or not. The chosen cut-off is between the highest methylation level of 3.6% for patients classified as unmethylated (mean 2.3% and standard deviation 0.48%), and the lowest observed methylation level of 13.6% for hypermethylated patients.”

We believe that the high quality of the pyrosequencing data is partly due to the excellent quality of the DNA from the SCAN-B cases, which was used also for WGS. To note, although we also used a 7% methylation cut-off in our original study⁴, in which 254 SCAN-B cases were analyzed by WGS, only 237 cases passed the WGS quality filtering and were included in analyses in both our first study of the SCAN-B cohort⁴, and the current study. The main reason for cases failing WGS quality was low tumor cell content. Please see our article on the first study of the SCAN-B cohort⁴, for further information about WGS quality filtering and sample exclusions.

→Moreover, cf. cleanness of the data & fact that two different bisulfite conversion kits were used, please provide the obtained bisulfite conversion efficiencies for the controls included (mean & sd).

Response: In the bisulfite conversion we include cell line controls for methylation (full methylation and unmethylated). After conversion, we run a methylation specific PCR with

primers versus DPAK1, which are included in the kits. PCR products are next run on agarose gels to verify that bands are observed for the positive (methylated) control but not for the negative (unmethylated) control. The converted control samples are included as controls in the corresponding pyrosequencing run.

In the revised manuscript, we have updated the Supplementary Information (subsection headed: ‘*DNA promoter methylation analysis by pyrosequencing - Tumor DNA*’) with more details of the protocol and referencing the original description of the analysis ²⁴. In further response to the reviewer’s remark, we have calculated the corresponding pyrosequencing allele methylation rates for the control samples summarized per primer set for the tumor analysis runs and included these estimates in the revised Methods section related to pyrosequencing:

“For the methylated control samples, across 10 pyrosequencing tumor runs, the mean and standard deviation for the percentage of BRCA1 allele methylation was 95.48 ± 1.22 for primer set 1 and 96.16 ± 1.32 for primer set 2. For the unmethylated control, the corresponding values were 2.68 ± 0.75 for primer set 1 and 1.55 ± 0.90 for primer set 2. The combined methylation estimates for “unmethylated” samples ($2.26 \pm SD 0.45$; max 3.59%) are therefore in line with the estimates for the negative conversion controls.”

For the blood DNA methylation analyses similar ranges were observed (primer 1 methylated: range 92-94%, primer 1 unmethylated: range 1-2%, primer 2 methylated: range 96-98%, primer 2 unmethylated: range 2-3%), however, the limited number of conversion runs precludes a meaningful mean and standard deviation to be computed.

→Also, to allow for comparison with and practical application in other studies: are these cut-offs also applicable for the Infinium Epic microarrays, for which (combination of) probes? (Latter is perhaps more suited for supplementary).

Response: The question (as we understand it) of whether a 7% cut-off is applicable also to Illumina EPIC DNA methylation data is interesting. EPIC DNA methylation data for 30 CpGs associated with *BRCA1* (transcription start site (TSS): -1500b to +500bp) included in the EPIC array is shown in original figure 1A. The actual data values are now included in the Source Data file requested by the journal. Thus, researchers are free to investigate the question by combining the source data from this study. It is evident from the original figure 1A that not all EPIC CpGs reflect pyrosequencing methylation status equally, as is also shown in detail in Rebuttal figure 8 below and as discussed extensively in response to remarks by Reviewer 1. With regards to the Illumina EPIC array, individual CpG probes show higher baseline variability. This is to be expected given that it is a general-purpose genome-wide screening platform and that the EPIC platform uses hybridization as the mode of detection. The effect of technical between-probe variability on the Illumina platform could, however, be expected to cancel out when considering multiple probes in aggregate and it is therefore reassuring that the between platform (Illumina and pyrosequencing) results converge in the full promoter analysis (clustering showed in original figure 1A).

Rebuttal figure 8. Scatterplots of 9 Illumina EPIC CpGs associated with *BRCA1* based on Illumina annotations versus pyrosequencing-based allele methylation %. Dotted red line corresponds to a 1:1 relationship.

Thus, a more careful analysis is needed to determine which EPIC CpGs are appropriate for analysis, and then an optimization analysis could be performed to determine an optimal beta-value cut-off. This type of analysis optimally requires a validation cohort separate from the training cohort. Although of interest, it is not within the primary scope of this study to derive an *BRCA1* hypermethylation cut-off optimized for EPIC DNA methylation arrays. Moreover, we believe that EPIC array data can be further improved by novel algorithms correcting for tumor cell content, and this is an avenue of research that we are currently pursuing.

Other comments that should be addressed:

→Elaborate on the performed “variance filtering for outlier CpGs” (cf. supplementary, global DNA methylation analysis)

In the revised Supplementary Methods (in the section headed ‘*Global DNA methylation analysis*’) we have expanded the description and purpose of this filtering:

Briefly, the variance filtering was set so that, per CpG, there had to be an absolute difference in beta-value of at least 0.1 between the sample with the 5th lowest beta and the sample with the 5th highest beta in the 235 cases with DNA methylation data. In practice, this filter removes CpGs with a close to zero standard deviation in beta-value, that are uninformative in downstream supervised/unsupervised analyses.

This approach produces very similar results (95.6% concordant) to a standard bottom 20% variation cut-off based on probe SD, but corresponds to a more directly interpretable minimum beta-value difference between the five samples with the lowest and highest beta values.

→Gene expression analyses (supplementary): why did the authors apply two different offsets, i.e. +1 and +0.1?

Response: The +1 offset has been used in previous studies for subtype classifications of RNAseq data from SCAN-B cases^{4, 25, 26}. For this study, we obtained existing classifications from the public data deposited as part of our first study of the SCAN-B cohort⁴; no subtype classifications were performed specifically for this study. For the unsupervised clustering and supervised SAM analysis performed specifically in this study, we used the lower offset to account for more variability in the data (the low range). For instance, with an offset of +1, 3329 genes remained at the same standard deviation threshold as >7000 in the original analysis. A control PCA analysis for the +1 offset (n=3329 genes) again demonstrated that BRCA1 class and cohort contributed little to explaining the overall variability of the data, in contrast to, e.g., the suggested molecular subtypes in TNBC (TNBCtype²⁷) (Rebuttal figure 9 below).

Rebuttal figure 9. PCA analysis using the swamp R-package for 109 *BRCA1*-null and hypermethylated cases using an FPKM offset of +1 prior to log₂ transformation. Analyzed cases were combined from the SCAN-B cohort and Jönsson et al.³ as outlined in the submitted manuscript. PCA was performed on 3329 genes that passed a filter step requiring a standard deviation >0.6 in mean-centered expression across all samples. BRCA1class corresponds to a sample being hypermethylated or *BRCA1*-null (TRUE) or not altered (FALSE).

→New RNA-seq data was added for 27 BRCA1-null tumors (gene expression analyses), did the authors adjust for the batch effect? Or can the authors demonstrate that the impact is non-substantial? Batch effects may easily obscure other significant differences.

Response: The RNAseq analysis for the 27 new cases from Jönsson *et al.*³ were performed on fresh frozen RNA, using the same protocol, instruments, lab facilities, and analysis pipelines as for the SCAN-B cases. The reason for including the 27 cases was to increase sample size to improve the statistical power. For the question at hand, this was important as we had seen in the SCAN-B cases alone that the two groups appeared to have very similar genomic patterns. The concern of the reviewer is, however, very important and relevant. Consequently, we performed a number of control analyses to verify that the cohort was not confounding the conclusions, including, separate analyses in SCAN-B cases only, as described below.

In response to this comment, the original figures 4B and 4D are important. In the clustering shown in original Figure 4B, the first patient annotation bar shows whether a patient belongs to the SCAN-B cohort or the additional *BRCA1*-mutant cohort from Jönsson *et al.*³. Importantly, the cohorts are intermingled. Figure 4D shows a principal component analysis demonstrating that the cohort-variable does not contribute significantly to explaining the variation in gene expression observed in the data. We do note that the cohort variable is stronger in component 4, which is estimated to account for 4% of the observed variation. The presence of a small cohort component is why we also compare results from the total cohort with SCAN-B cases only (e.g., for the supervised analysis). We reasoned in a similar way for the WGS analyses, as different preprocessing and mutation analyses/annotations may produce small differences in the final output data.

Re-performing the gene expression PCA analysis in SCAN-B cases only (n=57 hypermethylated and n=25 *BRCA1*-null cases) yielded the same result concerning *BRCA1*-status (Rebuttal figure 10 below).

Rebuttal figure 10. PCA analysis using the swamp R-package for 82 *BRCA1*-null and hypermethylated cases from the SCAN-B cohort alone, using an FPKM offset of +0.1 prior to log₂ transformation. PCA was performed on 9650 genes that passed a filter step requiring a standard deviation >0.6 in mean-centered expression across all samples. *BRCA1*class corresponds to a sample being hypermethylated or *BRCA1*-null (TRUE) or not altered (FALSE).

Re-performing the consensus clustering for SCAN-B cases alone using the same cut-off in standard deviation (n=6190 genes) showed similar results as shown in the original Figure 4C, i.e., that the consensus expression clusters formed harbored both *BRCA1* hypermethylated and *BRCA1*-null cases (Rebuttal figure 11, below).

		Cluster solution (k)				
		3	4	5	6	7
A) % methylated in a cluster	1	55.9	72.7	72.7	72.7	72.2
	2	76.5	75	78.3	84.6	83.3
	3	85.7	35.7	27.3	27.3	40
	4		85.7	69.2	70	70
	5			84.6	69.2	62.5
	6				84.6	80
	7					85.7

		Cluster solution (k)				
		3	4	5	6	7
B) % BRCA1-null in a cluster	1	44.1	27.3	27.3	27.3	27.8
	2	23.5	25	21.7	15.4	16.7
	3	14.3	64.3	72.7	72.7	60
	4		14.3	30.8	30	30
	5			15.4	30.8	37.5
	6				15.4	20
	7					14.3

Rebuttal figure 11. Consensus clustering of gene expression data for SCAN-B cases. The 57 hypermethylated and 25 *BRCA1*-null cases were submitted to consensus clustering, performed as in the submitted manuscript using 6190 genes passing a standard deviation filter of >0.6 in mean centered expression. Tables show summarized results from consensus clustering, using a range of 3 to 7 cluster solutions (k, columns). For each cluster solution, the percentage of *BRCA1* hypermethylated (left matrix) and *BRCA1*-null (right matrix) cases in the defined clusters are shown (rows).

When re-performing the supervised SAM analysis using the same standard deviation cut-off and offset in SCAN-B cases alone, one gene, *BRCA1*, was significantly differentially expressed between the sample groups ($q < 0.01$). Compared to our original supervised analysis, we believe that this result is likely due to the lower statistical power combined with the absence of a “true” larger biological signal between the groups.

Together, we believe these results are supportive of the conclusion that cohort bias is not influencing the group level conclusions presented in the manuscript.

→Line 302 – “BRCA1 hypermethylation was mutually exclusive with BRCA1-null cases”: although I agree that they did not co-occur, calling them “mutually exclusive” requires more evidence (particularly samples), as there’s no reason to assume that they cannot co-occur at lower rates.

Response: In the revised manuscript we have rephrased this sentence to allow for alternative interpretations given more data, as follows:

“In this cohort, BRCA1 hypermethylation was mutually exclusive with BRCA1-null cases.”

→Statistical testing of methylation results (lines 325 – 334): methylation data (beta-

values) are heteroskedastic, was this taken into account for when using t-tests and ANOVA? Consider transforming to M-values, as also non-parametric tests suffer from heteroscedasticity. Moreover, the observation that higher BRCA1 methylation in blood occurs independent of age cannot be tested by one-way ANOVA, only by a 2-way ANOVA (preferably on M-values) that also considers the age/status interaction.

Response: The aforementioned analysis is based on pyrosequencing estimates (not EPIC data) and the range of the methylation levels are highly limited (<10%) compared with the full range of 0-100% or 0-1 for the EPIC arrays. In the original analysis we did not transform data to M-values. In response to the reviewer's remark we transformed pyrosequencing estimates to M-values ($M = \log_2(\text{Beta}/(1-\text{Beta}))$) for the original figure 2B, finding that the statistical association remained (T-test $p=0.02$, Wilcoxon's test $p=0.02$). This piece of information has been added to the second subsection of the Results in the revised manuscript (page 10, below) and we defined M-values in the Material and Methods section:

"Trends also remained significant after transformation of methylation rates to M-values ($p=0.02$)."

We next transformed the pyrosequencing estimates in the original Figure 2C to M-values. We were able to include one additional SCAN-B sample, which was previously left out as a control in the re-run analysis. This increased the total number of analyzed SCAN-B cases to 105 (55 cases with somatic tumor hypermethylation and 50 without). All re-run data are available in the Source Data file for the figure. As the M-value is a transformation of beta, the pattern across age groups was retained.

The reviewer is correct that the statistical test reported in the original figure 2C is not appropriate for the text formulation, and we thank the reviewer for highlighting this flaw. In the revised manuscript, we have performed a 2-way ANOVA using an interaction model (methylation~ TumorStatus+Age groups + TumorStatus:Age groups), as suggested, on both the original pyrosequencing estimates and the transformed M-values for the age group analysis. P-values for the tumor status variable were significant in both instances. We also tested a different age stratification (<40, 40-59, >=60, as reported for the revised outcome analysis) and this also showed statistical significance for *BRCA1* tumor methylation status. The following text is now included in the second subsection of the Results (page 10):

"To confirm these findings, we analyzed peripheral blood DNA from 71 additional SCAN-B cases from our cohort not subjected to prior clinical screening, including 39 cases with tumor BRCA1 hypermethylation and 32 without. Again, higher BRCA1 promoter methylation levels were found in blood DNA from patients with BRCA1 tumor hypermethylation (T-test $p=0.005$, Wilcoxon's test $p=0.01$). The finding also remained significant after transformation of methylation rates to M-values (T-test $p=0.006$, Wilcoxon's test $p=0.01$). When we combined both clinically screened and unscreened SCAN-B patients ($n=105$ in total, reanalyzed using the same instrument settings and reagent lots), we observed that the higher BRCA1 blood DNA methylation levels appeared independent of age (two-way ANOVA, $p=0.002$, interaction model with tumor hypermethylation status and age groups, Figure 3C). This finding also remained significant after transformation of methylation rates to M-values ($p=0.002$)."

In summary, to address the comments of this reviewer, we have included M-value calculations, updated original figure 2C, and added the suggested statistical test. In terms of the global analysis comparing EPIC methylation levels between *BRCA1*-null and hypermethylated samples, running the R-function "wilcox.test" test on M-transformed data

instead of ordinary betas resulted in 31 of the 32 originally discovered probes being significant after correction for multiple testing (same conversion formula as above with beta range capped at 0.01 and 0.99 prior to conversion, Bonferroni corrected $p < 0.05$). Thus M-transformation does not alter the conclusion of an apparent lack of global differences between *BRCA1*-null and hypermethylated samples. For visualization and biological interpretation purposes, however, we prefer the more intuitive betas to the non-linearly transformed M-values.

→Figure 3D shows a significant difference in number of indels between *BRCA1*-null & hypermethylated samples, contrasting the statement on lines 401 – 402

Response: The reviewer is correct that the statistical difference is significant ($p=0.01$). This is likely due to the sample sizes, as the effect size does appear to be small. For instance, the median in the *BRCA1*-null group is 394 indels compared with 553 indels in the hypermethylated group, which we believe does not constitute a defining characteristic of hypermethylated versus *BRCA1*-null TNBC tumors. To make this analysis more stringent we re-performed the analysis using only cases with at least 30X sequence depth, as the ability to identify somatic mutations may be affected by both tumor cell content and sequence depth. The original figure 3D has been updated accordingly; the results remain the same. We have rephrased the sentence in the fifth subsection of the Results to (page 14):

*“Only a small statistical difference in the number of indels (*BRCA1*-null: median=410, hypermethylated median=633) was observed between the two groups for cases sequenced with at least 30-fold sequence depth. This absolute difference in numbers did, however, not correspond to a difference in proportions of indels at repeats or with microhomology, as previously shown (Figure 2C). Moreover, these results imply a similar tumor mutational burden for hypermethylated and *BRCA1*-null cases.”*

→The paper includes many analyses leading to the same conclusion, consider summarizing in one sentence & refer to the supplementary methods (e.g. lines 343 – 349; lines 453 – 462; and associated figures) to improve overall readability.

Response: Lines 343-349 in the original manuscript describe proportions of molecular subtype for hypermethylated cases. Lines 453-462 describe immune cell deconvolution by RNAseq, PDL1 IHC, and TILs in situ results. These are separate analyses, and it is therefore not clear how the conclusion could be summarized in one sentence. As a general response to the remark, we have changed the subheadings of the Results section in an attempt to differentiate the analyses.

- Why the >50 years old cut-off, rather than considering age as a continuous variable (or using median as far less “selected” cut-off)? Please provide rationale or modify.

Response: As recently highlighted, cut-offs in age differ between studies²⁸ making it hard to interpret results. Moreover, age is not linearly correlated with risk of dying for breast cancer and may vary with subtype / clinical grade²⁹. Age as a continuous variable was not associated with IDFS or DRFI in univariate Cox regression in the SCAN-B cohort ($p > 0.05$). We believe that using median age as a cut-off could be slightly problematic as it would make the cut-off strongly related to the cohort at hand, risking skewness, especially for smaller cohorts. In

addition, the IDFS clinical endpoint includes death of any cause as an event according to the STEEP guidelines³⁰, introducing the issue of competing risks.

In response to the reviewer's remark, we have therefore changed the age binning in the survival analysis to align it with a recent large-scale SEER analysis²⁸ - binning patients by young age (<40), middle age (40-59), and older (≥ 60 years). Replacing the original age binning in the multivariate model for *BRCA1* hypermethylated cases vs non-methylated maintained hypermethylation as significant (HR=0.3261, 95% CI=0.12-0.88) for IDFS while borderline non-significant for DRFI ($p=0.053$). Similarly, the combined group of *BRCA1*-null and hypermethylated patients was associated with a better IDFS (HR=0.35, 95% CI=0.14-0.84) and DRFI (HR=0.23, 95% CI=0.074-0.72) than non-altered patients. However, the proportional hazard assumption was not fulfilled in any of the IDFS analyses, but was fulfilled in the DRFI analyses, illustrating the issue with including death not otherwise specified as an event in survival analyses.

For transparency, we have now included age as a continuous variable in the multivariable Cox regression models in the revised manuscript. Inclusion of age as a continuous variable renders the IDFS results for hypermethylation alone and combined borderline non-significant ($p=0.07$), while the DRFI association remains significant. Cohort size is likely a limiting factor in these analyses, as is acknowledged in the original Discussion section.

In addition, we have removed the AIMS PAM50 single sample classification, as it is not a standard clinicopathological variable and not benchmarked against the ProSigna clinical assay.

In line with these changes, we have also updated the main text in the fourth subsection of the Results (page 12):

"To assess the independent prognostic value after adjuvant chemotherapy of BRCA1 hypermethylation alone or in combination with BRCA1-null status, we performed multivariable Cox regression analysis adjusting for tumor size (≤ 20 mm, > 20 mm), patient age (<40, 40-59, and ≥ 60 years), tumor grade (1, 2, 3), and lymph node status (N0, N+) using IDFS as the clinical endpoint. BRCA1 hypermethylation was significantly associated with improved IDFS alone (HR=0.33, 95% CI=0.12-0.88) and when combined with BRCA1-null cases (HR=0.35, 95% CI=0.14-0.84), although the formal proportional hazard assumption was not fulfilled in these analyses ($p < 0.05$). When patient age was used as continuous variable in the IDFS analyses the results were borderline non-significant ($p=0.07$). Borderline non-significant results for BRCA1 hypermethylation vs non-methylated cases in multivariable Cox regression using the above models with stratified age bins (HR=0.28, 95% CI= 0.08-1.02, $p=0.053$) or continuous age ($p=0.09$) were also observed for distant relapse-free interval (DRFI). A significant association was seen for DRFI for the combined hypermethylation/BRCA1-null group, irrespective of whether binned age (HR=0.23, 95% CI=0.07-0.72) or continuous age (HR=0.32, 95% CI=0.11-0.94) was used."

and to the Discussion (page 20):

"When considering the observed survival curves for hypermethylated cases and the typical early relapse pattern of TNBC, the borderline non-significant statistical findings for BRCA1 hypermethylation are likely due to a relatively short follow-up time in the SCAN-B cohort (59% of cases have ≤ 5 years follow-up) and the limited sample size of the study."

We have removed the original Supplementary Figure S2 as the relevant data on DRFI is now provided in the main text.

Minor remarks:

→**Abstract, first sentence: “caused by genetic or epigenetic alteration in key pathway genes.” This is obvious, can be removed from abstract.**

Response: Addressed. Abstract is now rewritten to fit with journal requirements.

→**Introduction, line 155: study goes further than basic “genomic” phenotype, “omics” phenotype?**

Response: Addressed, thank you

→**Lines 167-168: hyphens are not required.**

Response: Addressed, thank you

→**Line 188: “as well as and” (remove “as well as” or “and”)**

Response: Addressed, thank you

→**Throughout paper: clearly mention whether pathologist(s) were blinded to sample status or not**

Response: The board-certified breast cancer pathologist scoring samples for TILs and PD-L1 was indeed blinded to all data / classifications. This has been added to the ‘*PD-L1 immunohistochemistry and tumor infiltrating lymphocytes (TILs)*’ Methods section. We thank the reviewer for pointing out this need for clarification.

→**Throughout paper: mutational/substitution signatures are referred to by numbers, which are not really informative for the reader, is it possible to refer to those signatures by name/characteristic (and add the number between brackets for completeness)? See e.g. lines 403 to 412, biological/clinical relevance is unclear without additional information.**

Response: We acknowledge the remark by the reviewer. The signatures do unfortunately not have “official” easy to interpret names that makes it easy to refer to them. In the revised manuscript we have made an effort to change wordings whenever possible in line with the recommendations by the reviewer. Textual examples include (subsection 5 of the Results, page 14):

“For mutational signatures, BRCA1-null cases had higher proportions of substitution signatures suggested to be associated with age at diagnosis (Signatures 1 and 5), an APOBEC

signature (Signature 2), and, to some extent, a substitution signature (Signature 8) reported to be elevated in cases with HRD ⁶

Subsection 5 of the Results, Page 14:

“No distinct differences were seen for the six rearrangement signatures reported by Nik-Zainal et al. ⁵, including the two BRCA1/BRCA2 associated signatures (RS3 and RS5)”

→Although I agree that low tumor purity will have an impact on e.g. LOH detection, please also mention the range of the tumor purity estimates for those samples where e.g. LOH was successfully detected.

Response: Addressed. The following sentence is now included in the first subsection of the Results (page 8):

“Of the 57 hypermethylated cases, 51 (89.5%) showed concurrent LOH of BRCA1 (tumor cell content by WGS range 23-82%) with the six remaining cases having low estimated tumor cell content (between 11-23% by WGS), which possibly interfered with the copy number analysis.”

→Supplementary information: as most readers won't be familiar with the “INCA” technical platform” and the parameters (Swedish-only), please make sure to mention the different options in English as well. At least verify that all selected options are also mentioned in English, leaving the Swedish parameter settings in there for full reproducibility is ok with me.

Response: The Swedish terms were used to allow for exact reproducibility. In the revised manuscript we have simplified the descriptions and made them more readable, removing the exact Swedish terms. We thank the reviewer for the comment.

→Supplementary information: part on DNA promoter methylation analysis requires additional proofreading, e.g. bisulfite vs. bisulphite, “were included at every individual each bisulfite conversion run”, “commercially available controls included in the bisulfite conversion were included in each pyrosequencing run to as quality controls and assessment of methylation thresholds” etc.

Response: Addressed. Thank you.

→Line 250: “in in” => “in the”

Response: Addressed, thank you

→Line 293: “is rare” => is rare in cancer samples (maybe rather common, but only leads to breast cancer in case of associated LOH or other event)

Response: Addressed, thank you

→Line 314: 16/34 = 47.1% (rather than 47.0%, after rounding)

Response: Addressed, thank you

→Line 318: were => was

Response: Addressed, thank you

→Lines 443 – 444: non-informative, please remove

Response: Addressed, thank you

→Lines 465 – 468: unclear sentence

Response: In response to the remark we have removed the sentence and the corresponding figure panel in the revised manuscript.

→Line 520: is already present in the main clone

Response: Addressed, thank you

→How does the machine learning approach (cf. supplementary, expression analyses) fit in the paper?

Response: Addressed, by removing. No longer applicable.

→Table 2: line 847: “Hard brackets imply <.”? Line 856: UNS: uncertain => “UNC”

Response: Clarified in the revised figure legend. Thank you

Reviewer #3 (Remarks to the Author): Expert in breast cancer genomics

The authors do a comprehensive analysis of methylation status and phenotypes of BRCA mutated breast cancers. This is proposed to be a population based study from within a clinical trial as a unique characteristic. However, prevalence of mutation and methylation have been previously determined in breast cancer populations not otherwise selected. Overall, the concepts are not new as the frequency of mutation and methylation are in accord with previous studies. Further, the correlations with outcomes are with FEC plus minus taxane chemotherapy as compared to more conventional chemotherapy approaches or PARP inhibitors limits the utility of the information to current management. The limited sample size and followup further limits the utility of the data.

Response: We agree with the reviewer that the occurrence of *BRCA1* hypermethylation or pathogenic mutations in a TNBC population is not a new concept, nor do we claim that. In the original *Introduction* section we cite several studies reporting *BRCA1* hypermethylation frequencies in TNBC, showing the quite variable estimates reported (references have also been updated in response to another remark by this reviewer). This variability is likely reflective of cohort selection bias, an issue even for clinical trials that often involve strict patient exclusion criteria (e.g. concerning age). In contrast, we demonstrate that on a yearly basis similar proportions of hypermethylated cases are identified in the SCAN-B cohort (in the lower end of the scale compared to many previous reports), lending support to the generalizability of the results (even if one argues it is limited to a single healthcare region).

While the prevalence of hypermethylation and mutation is important (and has been reported before) it is not the key focus of the current study. Instead, the current study takes the comparison of these groups a considerable number of steps forward compared to the existing literature by comprehensively analyzing the genetic, epigenetic, gene expression, and immune cell landscapes in and between the groups. To the best of our knowledge, this has not been done before in a similar, comprehensive, way using current state-of-art genomic methods. An important aspect of the study, as the reviewer acknowledges later, is the value of all data made publicly available. We believe that the presented comprehensive analysis of early stage tumors demonstrates that if one would not know the mutation status of *BRCA1* it would be difficult to tell these tumors apart. In the revised Introduction we have rephrased text to further emphasize the knowledge gap that we believe the current study fills (page 6-7):

“Currently, we lack a detailed multi-layer comparison of BRCA1 hypermethylated versus BRCA1-mutated early-stage TNBCs using current state-of-the-art profiling techniques that thoroughly investigates similarities and differences between the two groups”

Concerning treatment, despite the success of PARP inhibitors they are still not in routine clinical adjuvant or neoadjuvant use for TNBC (Europe/US), meaning that routine (neo)adjuvant treatment is still primarily based on chemotherapy. The comprehensive genomic profiling of *BRCA1*-null versus hypermethylated early-stage TNBC performed in this study demonstrates that the genomic characteristics of the two groups appear highly similar with HRD-positivity as a common characteristic. Given the successful results of PARP-inhibitor treatment, but also e.g. platinum-based chemotherapy, in *BRCA1/2* mutants it may be speculated that this would be then also apply to hypermethylated cases in an early-stage setting. Importantly, as hypermethylation is twice as frequent as the *BRCA1*-null phenotype and perhaps three-times as frequent as germline *BRCA1* alterations alone in population-based disease cohorts, the number of patients that may be reconsidered for treatment increases substantially. Moreover, we clearly show that a hypermethylated phenotype is exclusively associated with a genetic HRD phenotype, representing thus the most likely cause of HRD in early-stage TNBC. In ovarian cancer, the PAOLA study³¹ demonstrated that HRD-positive but *BRCA1/2* mutant negative cases responded favorably to PARP-inhibitors, which is likely to be practice changing and highlights the need for clinical testing in that disease. In the revised manuscript we have expanded the Discussion to include the above aspects (Page 21):

“Recent randomized clinical trial data in ovarian cancer has showed that HRD-positive tumors without BRCA1/2 alterations respond favorably to PARP-inhibitors as first-line maintenance therapy³¹. Similarly, advanced breast cancer patients with HRDetect-high tumors have been shown to be associated with clinical improvement on platinum-based chemotherapy³², suggesting that also early stage patients with HRD-positive hypermethylated tumors may benefit from tailored therapies. If so, the early-stage patient cohort that could be

considered for tailored treatment would increase considerably, although the full treatment effect remains to be determined in clinical trials.”

To further address the remark of the reviewer we have moved and rephrased text in the Discussion section to acknowledge the limited sample size in the outcome analyses (page 20):

“When considering the observed survival curves for hypermethylated cases and the typical early relapse pattern of TNBC, the borderline non-significant statistical findings for BRCA1 hypermethylation are likely due to a relatively short follow-up time in the cohort (59% of cases have ≤ 5 years follow-up) and the limited sample size of the current study.”

→The observation that methylation can be detected in blood hematopoietic cells is of potential interest and significance. However, while the correlation is statistically significant it does not have sufficient power to preclude the need for tumor testing. Further this has been observed in previous studies.

Response: We agree with the reviewer about the current limitations with blood based hypermethylation results, as we also acknowledge in the original discussion (and further stressed in the revised manuscript, see below). This is a general shortcoming in the field and not specific to this study but can hopefully soon be addressed by sensitive and robust sequencing-based methods. We also cite previous literature in the original submitted manuscript in line with the reviewer’s comment.

Still, we believe that demonstrating existence of elevated hypermethylation levels in a population-based TNBC tumor cohort of this size profiled by different omics technologies merits being reported and will motivate further investigation and development of refined analysis techniques. For instance, while previous studies have been limited by the number of cases with tumor hypermethylation, we report a total of 55 profiled cases with somatic tumor hypermethylation and demonstrate that elevated levels are not due to, factors such as younger age. Moreover, in response to a remark by Reviewer 1, we have provided unpublished data of 20 cases demonstrating that the observed patterns can be reproduced by an orthogonal NGS-based assay using the same DNA as input (Rebuttal figure 5). While the exact origin/cause of the elevated methylation levels remains to be determined, the presence of them may have profound effects on diagnosis as discussed in the submitted manuscript. To acknowledge the remark by the reviewer we have revised the discussion about the elevated levels to emphasize that they at the moment (when analyzed by pyrosequencing) cannot be used for prediction of the tumor hypermethylation state (Discussion, Page 18):

“Although these results must be interpreted with some caution as the hypermethylation levels are at the limit of detection by pyrosequencing (and thus not suitable for prediction of somatic hypermethylation), they are intriguing in the context of both circulating tumor DNA and findings of mosaic constitutional BRCA1 hypermethylation.”

→Similarly the PDL1 difference may be of interest but is not significant due to the small sample size. Otherwise the observation of similar characteristics in the methylation and mutation cases again is not surprising.

Response: The PD-L1 difference between *BRCA1*-null and hypermethylated cases (original figure 4E) the reviewer refers to is not statistically significant when classified according to SP142 antibody classification guidelines (positive/negative). The elevated scores in *BRCA1*-null cases are interesting but needs to be further investigated in the context of, for example, neoantigen prediction and the tumor microenvironment. Our extended analyses of expressed neoantigens from substitutions did not disclose a cause for this difference (see response to one of the last remarks by reviewer 1). In the revised Discussion we have made a note of this (Page 22):

“Both groups are, however, clearly heterogeneous and include both “inflamed” and “cold” tumors with and without PD-L1 expression. Consequently, the trend of higher PD-L1 levels in BRCA1-null tumors compared with hypermethylated cases warrants further investigation, as does the intersection between HRD-positivity and a variable immunogenicity.”

While our study is limited (due the costs of e.g. WGS), hypermethylation analysis on, e.g., archival (FFPE) tissue remains challenging with a considerable proportion of inconclusive results (see e.g. the TNT trial and other studies^{16, 23}). Thus, good data on PD-L1, TILs etc. versus hypermethylation in TNBC is still scarce. Considering the reviewer’s remark, our data provide further delineation of the *BRCA*-wt group in TNBC with respect to TILs and PD-L1 compared to the literature by stratifying it further based on hypermethylation (which we show in the manuscript is associated with a *BRCA1*-null like phenotype). Taken together, we believe the current study reinforces the need for considering *BRCA1*-null and hypermethylated cases as similar on a group level with respect to immunotherapy treatment in the primary setting (as discussed in the original manuscript). Based on expanded literature searches to include more recent studies, we have expanded the discussion around TILs and PD-L1 (page 21):

“TNBC tumors have been associated with high TIL-levels (signaling immunogenicity) which are also prognostically favorable^{20, 21}. A recent study has suggested a higher frequency of TIL-positive tumors in BRCA1/2-mutated patients than in wild type cases²². In the SCAN-B cohort, TIL-levels did not differ between hypermethylated and BRCA1-null cases, which is consistent with the similar prognosis in these groups after adjuvant chemotherapy. PD-L1 expression is also frequent in TNBC, including BRCA1 hypermethylated tumors²³, but studies have not found that levels of PD-L1+ TILs in TNBC cancers are driven by a high mutation rate or by BRCA1 mutation status^{20, 22}. This lack of association may now be appreciated through our delineation of the BRCA1/2-wt subgroup by BRCA1 hypermethylation, forming a major subgroup of TNBC with similar genetic and immune cell phenotypes as BRCA1-null cases. Taken together, on a group level these results suggest that early-stage hypermethylated cases may have similar response to immune checkpoint inhibitors as the BRCA1-null group.”

→The addition of this sample set and the detailed analysis is of interest. However, it is important to note that over 2500 cases of BRCA1 methylation and associated outcomes and characteristics have already been presented in the literature.
Response: The reviewer is correct that several studies have reported *BRCA1* promoter methylation in breast cancer but also ovarian cancer previously. However, as we do believe the reviewer acknowledge in this remark, none have done it to the detailed and comprehensive level presented in the current manuscript, at the same time contrasting it to *BRCA1*-null cases.

In response to both this and the first remark raised by the reviewer we have searched the literature to include additional references to both the Introduction and Discussion reporting frequencies, outcome data, and blood-based findings (including ^{23, 33, 34, 35}) to acknowledge previous studies.

→There are no specific concerns with the data or data presentation. The discussion does note the controversies in the field and the contribution of this study appropriately.
Response: Thank you.

REFERENCES FOR REBUTTAL

1. Yang D, *et al.* Association of BRCA1 and BRCA2 mutations with survival, chemotherapy sensitivity, and gene mutator phenotype in patients with ovarian cancer. *JAMA* **306**, 1557-1565 (2011).
2. Findlay GM, *et al.* Accurate classification of BRCA1 variants with saturation genome editing. *Nature* **562**, 217-222 (2018).
3. Jonsson G, *et al.* The retinoblastoma gene undergoes rearrangements in BRCA1-deficient basal-like breast cancer. *Cancer research* **72**, 4028-4036 (2012).
4. Staaf J, *et al.* Whole-genome sequencing of triple-negative breast cancers in a population-based clinical study. *Nature medicine* **25**, 1526-1533 (2019).
5. Nik-Zainal S, *et al.* Landscape of somatic mutations in 560 breast cancer whole-genome sequences. *Nature* **534**, 47-54 (2016).
6. Nik-Zainal S, Morganella S. Mutational Signatures in Breast Cancer: The Problem at the DNA Level. *Clin Cancer Res* **23**, 2617-2629 (2017).
7. Davies H, *et al.* HRDetect is a predictor of BRCA1 and BRCA2 deficiency based on mutational signatures. *Nature medicine* **23**, 517-525 (2017).
8. Telli ML, *et al.* Homologous Recombination Deficiency (HRD) Score Predicts Response to Platinum-Containing Neoadjuvant Chemotherapy in Patients with Triple-Negative Breast Cancer. *Clin Cancer Res* **22**, 3764-3773 (2016).
9. Van Loo P, *et al.* Allele-specific copy number analysis of tumors. *Proceedings of the National Academy of Sciences of the United States of America* **107**, 16910-16915 (2010).
10. Zheng X, *et al.* MethylPurify: tumor purity deconvolution and differential methylation detection from single tumor DNA methylomes. *Genome biology* **15**, 419 (2014).
11. Aran D, Sirota M, Butte AJ. Systematic pan-cancer analysis of tumour purity. *Nat Commun* **6**, 8971 (2015).

12. Zheng X, Zhang N, Wu HJ, Wu H. Estimating and accounting for tumor purity in the analysis of DNA methylation data from cancer studies. *Genome biology* **18**, 17 (2017).
13. Benelli M, Romagnoli D, Demichelis F. Tumor purity quantification by clonal DNA methylation signatures. *Bioinformatics* **34**, 1642-1649 (2018).
14. Aryee MJ, *et al.* Minfi: a flexible and comprehensive Bioconductor package for the analysis of Infinium DNA methylation microarrays. *Bioinformatics*, (2014).
15. Qin Y, Feng H, Chen M, Wu H, Zheng X. InfiniumPurify: An R package for estimating and accounting for tumor purity in cancer methylation research. *Genes Dis* **5**, 43-45 (2018).
16. Tutt A, *et al.* Carboplatin in BRCA1/2-mutated and triple-negative breast cancer BRCAness subgroups: the TNT Trial. *Nature medicine* **24**, 628-637 (2018).
17. Isakoff SJ, *et al.* TBCRC009: A Multicenter Phase II Clinical Trial of Platinum Monotherapy With Biomarker Assessment in Metastatic Triple-Negative Breast Cancer. *J Clin Oncol* **33**, 1902-1909 (2015).
18. Schenck RO, Lakatos E, Gatenbee C, Graham TA, Anderson ARA. NeoPredPipe: high-throughput neoantigen prediction and recognition potential pipeline. *BMC Bioinformatics* **20**, 264 (2019).
19. Shukla SA, *et al.* Comprehensive analysis of cancer-associated somatic mutations in class I HLA genes. *Nature biotechnology* **33**, 1152-1158 (2015).
20. Sobral-Leite M, *et al.* Assessment of PD-L1 expression across breast cancer molecular subtypes, in relation to mutation rate, BRCA1-like status, tumor-infiltrating immune cells and survival. *Oncoimmunology* **7**, e1509820 (2018).
21. Loi S, *et al.* Tumor-Infiltrating Lymphocytes and Prognosis: A Pooled Individual Patient Analysis of Early-Stage Triple-Negative Breast Cancers. *J Clin Oncol* **37**, 559-569 (2019).
22. Solinas C, *et al.* BRCA gene mutations do not shape the extent and organization of tumor infiltrating lymphocytes in triple negative breast cancer. *Cancer Lett* **450**, 88-97 (2019).
23. Jacot W, *et al.* BRCA1 Promoter Hypermethylation is Associated with Good Prognosis and Chemosensitivity in Triple-Negative Breast Cancer. *Cancers (Basel)* **12**, (2020).
24. Saal LH, *et al.* Recurrent gross mutations of the PTEN tumor suppressor gene in breast cancers with deficient DSB repair. *Nature genetics* **40**, 102-107 (2008).
25. Dihge L, *et al.* Prediction of Lymph Node Metastasis in Breast Cancer by Gene Expression and Clinicopathological Models: Development and Validation within a Population-Based Cohort. *Clin Cancer Res* **25**, 6368-6381 (2019).

26. Vallon-Christersson J, *et al.* Cross comparison and prognostic assessment of breast cancer multigene signatures in a large population-based contemporary clinical series. *Sci Rep* **9**, 12184 (2019).
27. Lehmann BD, *et al.* Identification of human triple-negative breast cancer subtypes and preclinical models for selection of targeted therapies. *J Clin Invest* **121**, 2750-2767 (2011).
28. Wang C, *et al.* Validation of CTS5 model in large-scale breast cancer population and the impact of menopausal and HER2 status on its prognostic value. *Sci Rep* **10**, 4660 (2020).
29. Cluze C, *et al.* Analysis of the effect of age on the prognosis of breast cancer. *Breast cancer research and treatment* **117**, 121-129 (2009).
30. Hudis CA, *et al.* Proposal for standardized definitions for efficacy end points in adjuvant breast cancer trials: the STEEP system. *J Clin Oncol* **25**, 2127-2132 (2007).
31. Ray-Coquard I, *et al.* Olaparib plus Bevacizumab as First-Line Maintenance in Ovarian Cancer. *The New England journal of medicine* **381**, 2416-2428 (2019).
32. Zhao EY, *et al.* Homologous Recombination Deficiency and Platinum-Based Therapy Outcomes in Advanced Breast Cancer. *Clin Cancer Res* **23**, 7521-7530 (2017).
33. Azzollini J, *et al.* Constitutive BRCA1 Promoter Hypermethylation Can Be a Predisposing Event in Isolated Early-Onset Breast Cancer. *Cancers (Basel)* **11**, (2019).
34. Brianese RC, *et al.* BRCA1 deficiency is a recurrent event in early-onset triple-negative breast cancer: a comprehensive analysis of germline mutations and somatic promoter methylation. *Breast cancer research and treatment* **167**, 803-814 (2018).
35. Tang Q, Cheng J, Cao X, Surowy H, Burwinkel B. Blood-based DNA methylation as biomarker for breast cancer: a systematic review. *Clin Epigenetics* **8**, 115 (2016).

REVIEWERS' COMMENTS:

Reviewer #1 (Remarks to the Author):

Glodzik et al., has thoroughly revised the manuscript. I appreciate that the title has been updated, figure 1 displaying the new consort diagram gives a much better overview of which samples have been used in which analyses. Figure 2A gives a clear picture of BRCA1 CpG islands either hyper/unmethylated. The explanation of methylation and expression for BRCA1 has been extensively addressed. Thanks for adding Sup table 1- I am sure it will be used and thus cited in many other publications. Rebuttal figure 6 does not need to be included. Interestingly, increasing tumour size seems to be a good factor for IDFS. For clarification BRCA1 methylation was tested in primary tumours in TNT trial - not the metastatic disease -, the response to treatment was assessed in the metastatic disease, please change your sentence accordingly. Figure 4 B - spelling mistake - amplified. I appreciate the additional analyses of neoantigens and immune cell. Overall, I have no further comments and think this paper provides valuable information for the community.

Reviewer #2 (Remarks to the Author):

The authors did a major effort to take into account my comments in the revised version of the manuscript. The large majority of comments led to appropriate modifications of the manuscript, whereas some other ones were rightfully rebutted. Other remarks (cf. indel frequency difference; batch effects RNA-seq) could in my opinion not be fully resolved, yet were appropriately demonstrated to have no impact on the overall conclusion. I have therefore no major additional comments. Nevertheless, I have some remaining minor comments that could lead to further improvement of this study.

1. Explicitly indicate that the bisulfite conversion efficiency is sufficiently high to ensure to avoid false positive methylation results.
2. As appropriately demonstrated by the authors, methylation cut-offs used cannot be exactly ported to the Infinium EPIC platform, please mention this explicitly to avoid other researchers relying on this cutoff.
3. The authors retained the title "variance filtering for outlier CpGs". Though the implementation is ad hoc, their general strategy is commonly used to improve power (custom statistical testing) and stability (machine learning) of the results. However, please clarify (or modify accordingly) the title, as it erroneously suggests that you want to filter out "outlier CpGs". Unless I misunderstand the authors, their goal is to remove invariant CpGs, and the strategy was devised in such a way that outlier CpGs (i.e. highly variant, but only in very few samples and therefore practically useless) had minimal impact on filtering.

Reviewer #3 (Remarks to the Author):

This reviewer would like to thank the authors for the careful, thorough, and reasoned responses to the concerns of the three reviewers. The strength of this manuscript arises from the detailed analysis of multiple characteristics associated with BRCA1 methylation in early breast cancer patients. In one manner the case number is low compared to the overall prevalence of the disease and BRCA1 methylation. In another manner the depth of analysis of each case is a positive. Nevertheless, the new information and conclusions from this manuscript compared to the data in current literature remain limited.

Thus the only major limitation of the manuscript is whether there is sufficient new information or understanding arising from the analysis to warrant publication.

Detailed point-by-point response to the reviewers' comments on manuscript NCOMMS-20-07196-T

First, we would like to thank all reviewers for their time and effort and the editors for the opportunity to revise our work. Responses to the reviewers' comments are in plain text, with textual changes indicated by italics.

REVIEWERS' COMMENTS:

Reviewer #1 (Remarks to the Author):

Glodzik et al., has thoroughly revised the manuscript. I appreciate that the title has been updated, figure 1 displaying the new consort diagram gives a much better overview of which samples have been used in which analyses. Figure 2A gives a clear picture of BRCA1 CpG islands either hyper/unmethylated. The explanation of methylation and expression for BRCA1 has been extensively addressed. Thanks for adding Sup table 1- I am sure it will be used and thus cited in many other publications. Rebuttal figure 6 does not need to be included. Interestingly, increasing tumour size seems to be a good factor for IDFS. For clarification BRCA1 methylation was tested in primary tumours in TNT trial - not the metastatic disease -, the response to treatment was assessed in the metastatic disease, please change your sentence accordingly. Figure 4 B - spelling mistake - amplified. I appreciate the additional analyses of neoantigens and immune cell. Overall, I have no further comments and think this paper provides valuable information for the community.

Author Response: We thank the reviewer for these final suggestions. In the revised manuscript we have addressed the spelling mistake and rephrased the TNT sentence to:

"In metastatic TNBC, BRCA1 hypermethylation has been reported not to be associated with a better treatment response to carboplatin compared with docetaxel ⁴⁸ (although hypermethylation in this study was measured in the primary tumor tissue, not the actual metastatic tissue) or to single-agent carboplatin effect ⁴⁹."

Reviewer #2 (Remarks to the Author):

The authors did a major effort to take into account my comments in the revised version of the manuscript. The large majority of comments led to appropriate modifications of the manuscript, whereas some other ones were rightfully rebutted. Other remarks (cf. indel frequency difference; batch effects RNA-seq) could in my opinion not be fully resolved, yet were appropriately demonstrated to have no impact on the overall conclusion. I have therefore no major additional comments. Nevertheless, I have some remaining minor comments that could lead to further improvement of this study.

Author Response: We thank the reviewer for the additional suggestions addressed as outlined below.

1. Explicitly indicate that the bisulfite conversion efficiency is sufficiently high to ensure to avoid false positive methylation results.

Author Response: The following sentence has been added to the Material and Methods section concerning the BRCA1 hypermethylation analyses:

“Taken together, this demonstrates that the bisulfite conversion efficiency is sufficiently high to ensure to avoid false positive methylation results.”

2. As appropriately demonstrated by the authors, methylation cut-offs used cannot be exactly ported to the Infinium EPIC platform, please mention this explicitly to avoid other researchers relying on this cutoff.

Author Response: The following sentence has been added to the Material and Methods section concerning the BRCA1 hypermethylation analyses when the cut-off is discussed:

“Importantly, the 7% cut-off is relevant for pyrosequencing data performed according to the specified protocol. It may therefore not be suitable for calling BRCA1 promoter hypermethylation in Illumina Infinium Methylation beadchips.”

3. The authors retained the title “variance filtering for outlier CpGs”. Though the implementation is ad hoc, their general strategy is commonly used to improve power (custom statistical testing) and stability (machine learning) of the results. However, please clarify (or modify accordingly) the title, as it erroneously suggests that you want to filter out “outlier CpGs”. Unless I misunderstand the authors, their goal is to remove invariant CpGs, and the strategy was devised in such a way that outlier CpGs (i.e. highly variant, but only in very few samples and therefore practically useless) had minimal impact on filtering.

Author Response: In the main text we have removed the term “variance filtering”, using instead “filtering”. In the Supplementary Information document providing detailed information about analysis steps we have rephrased the part of the text referring to the Illumina EPIC data analysis to:

“Filtering for CpGs reduced the final dataset to 614977 probes. Briefly, the filtering was set so that, per CpG, there had to be an absolute difference in beta-value of at least 0.1 between the sample with the 5th lowest beta and the sample with the 5th highest beta in the 235 cases with DNA methylation data. In practice, this filter removes CpGs with a close to zero standard deviation in beta-value, that are uninformative in downstream supervised/unsupervised analyses.”

Reviewer #3 (Remarks to the Author):

This reviewer would like to thank the authors for the careful, thorough, and reasoned responses to the concerns of the three reviewers. The strength of this manuscript arises from the detailed analysis of multiple characteristics associated with BRCA1 methylation in early breast cancer patients. In one manner the case number is low compared to the overall prevalence of the disease and BRCA1 methylation. In another manner the depth of analysis of each case is a positive. Nevertheless, the new information and conclusions from this manuscript compared to the data in current literature remain limited. Thus the only major limitation of the manuscript is whether there is sufficient new information or understanding arising from the analysis to warrant publication.

Author Response: We thank the reviewer for acknowledging the value of the study.